# Histone functions as a cell-surface receptor for AGEs

Masanori Itakura [1], Kosuke Yamaguchi[1], Roma Kitazawa[1], Sei-Young Lim[1], Yusuke Anan[1], Jun Yoshitake[2], Takahiro Shibata [3], Lumi Negishi[4], Hikari Sugawa[5], Ryoji Nagai [5] & Koji Uchida[1,6 ✉]

Reducing sugars can covalently react with proteins to generate a heterogeneous and complex group of compounds called advanced glycation end products (AGEs). AGEs are generally considered as pathogenic molecules, mediating a pro-inflammatory response and contributing to the development of a number of human diseases. However, the intrinsic function of AGEs remains to be elucidated. We now provide multiple lines of evidence showing that AGEs can specifically bind histone localized on the cell surface as an AGE-binding protein, regulate the function of histone as a plasminogen receptor, and result in the regulation of monocytes/macrophage recruitment to the site of inflammation. Our finding of histone as a cell-surface receptor for AGEs suggests that, beside our common concept of AGEs as danger-associated molecular patterns mediating a pro-inflammatory response, they may also be involved in the homeostatic response via binding to histone.

[1] Graduate School of Agricultural and Life Sciences, The University of Tokyo, Tokyo 113-8657, Japan. [2] Institute of Nano-Life-Systems, Institutes of Innovation for Future Society, Nagoya University, Nagoya 464-8601, Japan. [3] Graduate School of Bioagricultural Sciences, Nagoya University, Nagoya 464-8601, Japan. [4] Central Laboratory, Institute for Quantitative Biosciences, The University of Tokyo, Tokyo 113-0032, Japan. [5] School of Agriculture, Tokai University, Kumamoto 862-8652, Japan. [6] Japan Agency for Medical Research and Development, CREST, Tokyo, Japan. ✉email: a-uchida@g.ecc.u-tokyo.ac.jp

Glycation, also known as the Maillard or amino-carbonyl reaction, is the non-enzymatic reaction between reducing sugars, such as glucose, and free amino groups of proteins and other biomolecules. During the initial stage of protein glycation, the amino groups are converted into Schiff bases upon the reaction with reducing sugars followed by rearrangement to relatively stable products called Amadori products. The Amadori products further undergo a series of dehydration and rearrangement reactions leading to the production of several carbonyl intermediates, such as deoxyglucosones. These intermediates further react with the amino groups of proteins to form a heterogeneous and complex group of compounds, irreversibly resulting in accumulation on long-lived proteins called advanced glycation end products (AGEs). The accumulation of AGEs has been implicated in the pathogenesis associated with a number of human diseases and aging[1,2]. AGEs have been shown to initiate a wide range of cell-mediated responses, such as vascular dysfunction, matrix expansion, and glomerulosclerosis. AGEs also function as neoantigens that enhance inflammatory cellular immune responses. AGEs are known to mediate these effects by binding to specific receptors, such as the receptor for AGEs (RAGE) and scavenger receptors, on a wide range of cells[3]. The binding of AGEs to the receptors triggers downstream signaling pathways and mediates the cellular activation or proliferation leading to inflammation and tissue destruction. Because of these pro-inflammatory effects, AGEs have been generally considered to be a pathogenic molecule contributing to the pathogenesis of human diseases, such as diabetic complications and atherosclerosis.

Vitamin C (L-ascorbic acid), an essential dietary nutrient required for multiple biological functions, exists as a reversible redox couple with its oxidized form, dehydroascorbic acid (DHA). Strikingly, unlike the reduced form of vitamin C, DHA and its oxidized products, such as 2,3-diketogulonic acid, 3-deoxythreosone, xylosone, and threosone[1], show their ability to covalently react with the positively charged amino acid side-chains of proteins to form a variety of covalent adduct species structurally similar to the traditional AGEs that originated from glucose[4–6]. Previous studies have indeed shown that there is a similarity between the modifications of proteins by ascorbate and those in aged human lenses and cataracts in vivo[5]. In addition, the overexpression of a vitamin C transporter results in a significant accumulation of AGEs[7]. Therefore, the oxidation products of vitamin C could mediate the chemical aging of proteins in vivo, suggesting a potential link between the antioxidant function of vitamin C and glycation.

On the other hand, it has been reported that AGEs, including DHA-modified proteins, can be commonly recognized by the natural IgM antibodies, accelerating the IgM production without a typical memory B cell response[8]. In addition, DHA-derived AGEs have been shown to interact with the complement component C1q in human serum, accelerating the C1q-dependent classical complement pathway[9]. These findings raise the possibility that AGEs could be an important trigger of innate immunity, thereby contributing to the homeostatic responses. This unprecedented view provides a rationale for further establishing a molecular mechanism and broad functional significance of the formation and regulation of AGEs. In this context, we speculated the presence of an unknown mechanism for the regulation of AGEs, by which AGEs may have a previously unrecognized innate ligand function that could contribute to the protection against exogenous invading pathogens and endogenous damage-associated molecules, independent of pro-inflammatory RAGE and scavenger receptors.

In the present study, we examined the presence of cell-surface protein(s) that can bind to AGEs and identified histone as a previously unrecognized binding proteins for AGEs. Furthermore, based on the finding that AGEs regulate the activation of plasminogen (Plg) and suppress the infiltration of monocyte/macrophage, we suggest a mechanism by which AGEs exert anti-inflammatory function through binding to histones.

## Results

**Identification of a cell-surface binding protein for the AGEs**. To identify cellular binding partners for AGEs, the biotin (Bt)-labeled AGEs prepared by incubating biotinylated bovine serum albumin (BSA) with DHA for 3 days (Supplementary Fig. S1) were used as ligands for the pull-down assay. Both the RIPA soluble fraction and detergent-resistant membrane fraction (lipid raft fraction) prepared from mouse splenocytes were incubated with beads coupled to either the unmodified Bt-BSA or Bt-AGEs, and then the bound proteins were eluted and separated by SDS-PAGE under reducing conditions. The AGEs pull-down of the lipid raft fraction detected several unique bands containing a protein (band a) with a molecular mass of approximately 15-kDa (Fig. 1a). To identify the protein, the band resolved by silver staining was excised and analyzed by LC-tandem mass spectrometry (Supplementary Table S1). Based on three independent proteomic experiments, the protein was identified as histone H2A and H2B, main histone proteins involved in the structure of chromatin. Other protein bands with the molecular mass about 25-kDa (band b) and 10-kDa (band c) were also identified as histone proteins, H1 and H4, respectively.

To demonstrate the presence of histones on the cell surface, we conducted cell-surface biotinylation assay using a membrane-impermeable labeling reagent. As a result, histone proteins were shown to be present on the cell surface (Supplementary Fig. S2). Unlike RAGE, which is completely localized in RIPA soluble fraction, histones were detected in both RIPA soluble and lipid raft fractions (Supplementary Fig. S2). To further confirm that AGEs directly interact with histone, the binding specificity of AGEs to recombinant histone proteins (H1, H2A/B heterodimer, and H3/4 heterotetramer) was examined in vitro. The pull-down assay showed that AGEs indeed interacted with these recombinant histones (Fig. 1b). The solid phase binding assay also revealed that AGEs were almost equally bound to the histones (Fig. 1c). Furthermore, we determined the kinetics of interaction between the recombinant histones and AGEs using surface plasmon resonance and observed that the AGEs had a high affinity toward H2B with a $K_D$ of 67.2 nM, followed by histone H2A at 85.5 nM, histone H3 at 96.5 nM, histone H4 at 151 nM, and histone H1 at 206 nM (Fig. 1d).

**Histone ubiquitously recognizes AGEs**. It was observed that (i) the binding activity of AGEs to the histone depends on the modification by DHA and (ii) proteins (human and mouse serum albumins and transferrin) treated with DHA are commonly recognized by histone (Supplementary Fig. S3). These results and the finding that the DHA-treated Bt-labeled N-pentylamine (DHA-Bt-PA) showed a significant binding to H2B (Fig. 2a) suggested that some specific structures of AGEs derived from the reaction between primary amines and DHA could be specifically recognized by histone. Hence, to gain a structural insight into DHA-derived adducts possessing a binding activity to H2B, we separated and fractionated DHA-Bt-PA by reverse-phase HPLC, and the binding activity of each fraction to H2B was evaluated. The solid-phase binding analysis demonstrated that the fractions eluted from 17–22 min had a binding activity to the histone protein (Fig. 2b, top and middle). In addition, these histone-binding fractions were recognized by anti-AGEs antibodies (Fig. 2b, bottom). To gain more insight into the structural properties of AGEs exhibiting H2B binding potential, we

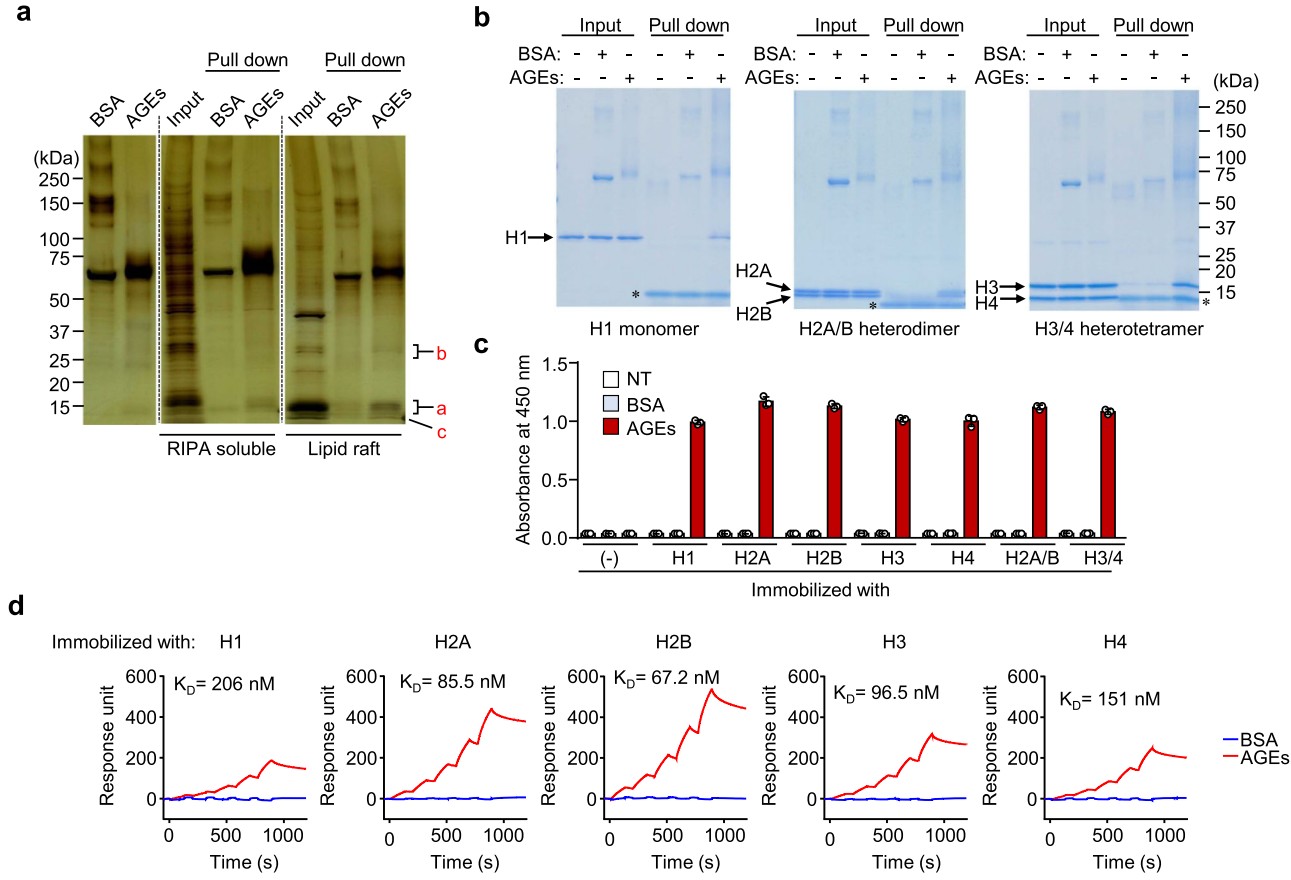

**Fig. 1 Identification of a cell-surface binding protein for AGEs. a** Pull-down assay for the detection of cell-surface AGEs-binding proteins. Both Bt-BSA or Bt-AGEs were incubated with proteins in RIPA-soluble or lipid raft fractions prepared from mouse splenocytes and subjected to pull-down assay with streptavidin-conjugated magnetic beads (Dynabeads). Binding proteins were separated by SDS-PAGE and detected by a silver staining method. Representative image of three independent experiments. **b** Pull-down assay for the binding of AGEs with the recombinant histone proteins. Recombinant histones (H1 monomer, H2A/B heterodimer, and H3/4 heterotetramer) incubated with Bt-BSA or Bt-AGEs were subjected to pull-down assay with streptavidin-conjugated magnetic beads. Pulled-down proteins were separated by SDS-PAGE, and gels were stained with CBB. The asterisk (*) represents nonspecific bands. **c** Binding of AGEs to the recombinant histones. Recombinant histones immobilized on ELISA plate were incubated with Bt-BSA or Bt-AGEs (5 µg/ml). The binding was detected using streptavidin-HRP. Data are mean ± S.D. of triplicate samples and are representative of three independent experiments. **d** Surface plasmon resonance measurements. The interaction of either BSA or AGEs (6.25, 12.5, 25, 50, and 100 µg/ml) with histones immobilized on a Biacore sensor chip NTA was monitored by the single-cycle kinetics method.

prepared the modified Bt-PAs with glucose and its metabolites and evaluated their binding potential to H2B. Solid-phase binding assays showed that modified Bt-PAs commonly bound to H2B, but to a greater or lesser extent, they did bind not to BSA (Fig. 2c). The histone protein also showed significant binding affinities for proteins modified with glucose and its metabolites (Fig. 2d). Furthermore, we confirmed the presence of histone-binding molecules in the sera of aged wild-type mice and diabetic ob/ob mice, for which elevated serum AGEs levels have been reported[10] (Supplementary Fig. S4).

Because histone is a highly basic protein, we speculated that electrostatic interactions might be involved in the recognition of AGEs by histone. Indeed, the binding of AGEs to histone H2B was inhibited by DNA (Supplementary Fig. S5) and NaCl (Fig. 2e). In addition, H2B recognized acylated proteins, such as acetylated BSA (Ac-BSA), succinylated BSA (Sc-BSA), and maleylated BSA (Ma-BSA) (Fig. 2f). These data support the hypothesis that the binding of the histone to AGEs may be driven by electrostatic interactions. Unlike the structured histone-fold domains, histone proteins have an N-terminal "tail" domain, which is known to be intrinsically disordered and contains highly basic amino acids. To test the involvement of the N-terminal domain in the binding of histone to AGEs, we conducted the

binding study using the recombinant maltose binding protein (MBP)-fusion fragments of H2B, namely H2B N-terminal (1-35) and H2B-ΔN35 (an H2B mutant lacking the N-terminal 35 amino acids) fragments. The H2B N-terminal (1-35) showed significant binding to AGEs to the same degree as the H2B-full length (FL), whereas no binding was observed with the H2B-ΔN35 fragment and with the acetylated H2B N-terminal (1-35) (Fig. 2g). In addition, AGEs were recognized by the scrambled fragments of the H2B N-terminal (scrambled 1, 2, and 3) comprising identical amino acids in different amino acid sequences (Fig. 2g). These data support our hypothesis that the electronegative potential of AGEs might be involved, at least in part, in the recognition by histone. However, lysozyme, a basic protein having an isoelectric point similar to that of histone proteins, did not show any recognition specificity to AGEs (Supplementary Fig. S6). The result may also suggest the involvement of some specific structures in the binding of histones to AGEs.

**Histone-dependent binding of AGEs to macrophages.** Histone H2B is known to have a unique function as a Plg receptor on the surface of human monocytes/macrophages[11,12]. Hence, we examined if there is a histone-dependent mechanism for the binding of

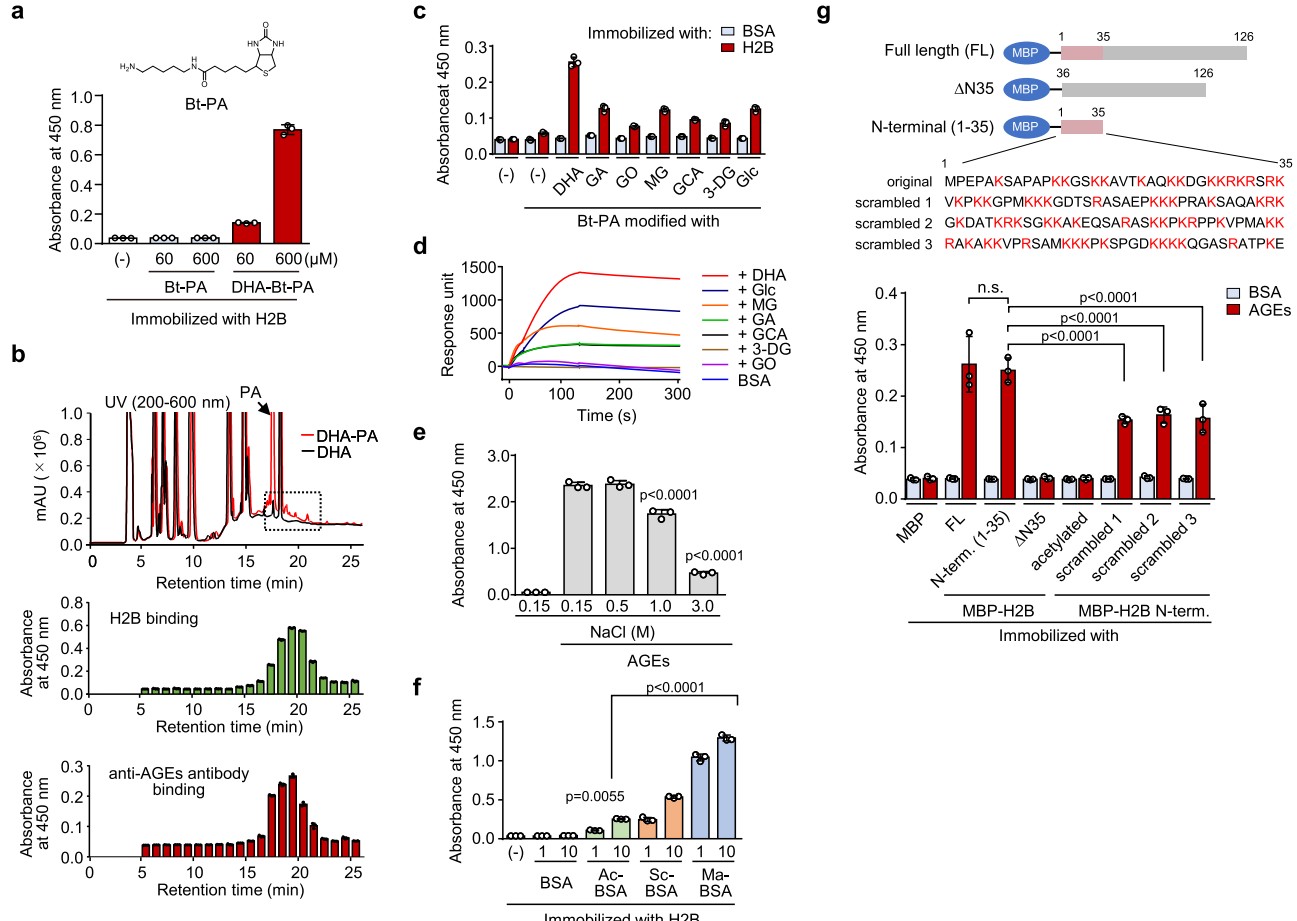

**Fig. 2 Ubiquitous recognition of AGEs by histones. a** Binding of DHA-modified Bt-PA to histone H2B. Recombinant histone H2B immobilized on ELISA plate was incubated with either Bt-PA or DHA-Bt-PA. The binding was detected using streptavidin-HRP. Data are mean ± S.D. of triplicate samples (representative of three independent experiments). **b** HPLC analysis of DHA-modified Bt-PA capable of binding to H2B. The reaction mixture of Bt-PA and DHA was separated by reverse-phase HPLC, and each fraction was measured for the binding to H2B (middle) and anti-AGEs antibodies (bottom). The reactions were monitored with absorbance at 200–600 nm (top). The dashed square indicates the region with the binding activity to H2B and anti-AGEs antibody. Data are mean ± S.D. of triplicate samples (representative of three independent experiments). **c** Binding of modified Bt-PAs to histone H2B. Bt-PA was incubated with DHA, glucose (Glc), and glucose metabolites, such as glycolaldehyde (GA), glyoxal (GO), methylglyoxal (MG), glyceraldehyde (GCA), and 3-deoxyglucosone (3-DG), and the binding to histone H2B was evaluated by solid-phase binding assay. Data are mean ± S.D. of triplicate samples (representative of three independent experiments). **d** Surface plasmon resonance measurements. The interaction of H2B and either BSA or AGEs (10 μg/ml) was monitored. **e** Effect of NaCl on binding of AGEs to H2B. H2B immobilized on ELISA plate was incubated with Bt-AGEs (5 μg/ml) under the indicated concentrations of NaCl. Data are mean ± S.D. of triplicate samples (representative of three independent experiments). Dunnett's test (two-sided), relative to AGEs treatment under 0.15 M NaCl. **f** Binding of acylated proteins to the recombinant histone H2B. Data are mean ± S.D. of triplicate samples (representative of three independent experiments). Dunnett's test (two-sided), relative to the no-treatment control. **g** Involvement of H2B N-terminal tail region and its electrostatic property in the binding to AGEs. MBP-fusion H2B full length (FL), N-terminal (N-term. (1-35)), N-terminal deletion mutant (ΔN35), acetylated Nterm. (1-35), or N-terminal scrambled fragments (scrambled 1, 2, and 3) were subjected to the solid phase binding assay. Data are mean ± S.D. of triplicate samples (representative of three independent experiments). Tukey–Kramer tests (two-sided), n.s. not significant (p > 0.05).

AGEs to cells. To determine the presence of AGE-binding cells, thioglycolate (TG)-induced peritoneal cells were treated with either Bt-BSA or Bt-AGEs and analyzed by flow cytometry using streptavidin-APC. Figure 3a (left panel) and Supplementary Fig. S7 show the presence of APC positive (APC (+)) cells in the Bt-AGEs-treated cells but not in the Bt-BSA-treated cells. APC positive cells were further gated on the population with APC positive profiles of the expression of both CD11b and F4/80, indicating that the cells involved in binding to AGEs in peritoneal cells were mainly macrophages (Fig. 3a right panel). Further examination of the cell-type specificity for the binding of AGEs revealed that significant binding of AGEs was observed only in macrophages and monocytes, not in dendritic cells, neutrophils, and eosinophils (Fig. 3b and Supplementary Fig. S8). The cellular binding of AGEs was also observed in macrophage cell lines, such

as J774A.1 and RAW264.7 (Fig. 3c, d), indicating the presence of AGEs-binding proteins on the cell surface. We further investigated whether the binding of AGEs to cells is a histone-dependent process. Consistent with the cellular binding of AGEs to macrophages, confocal microscopy showed colocalization of H2B and AGEs on the cell surface (Fig. 3e). In addition, to directly demonstrate the binding of AGEs to cell-surface H2B, we conducted the experiment using the membrane-impermeable cross-linker 3,3'-dithiobis (sulfusucciulmidyl propionate) (DTSSP). Cells were treated with Bt-BSA or Bt-AGEs and the protein interactions were cross-linked by DTSSP. The cells were then lysed and subjected to pull-down with streptavidin-coupled magnetic beads, followed by Western blotting of the pull-down and input fractions. As shown in Fig. 3f, significant signals were observed when treated with Bt-AGEs, which indicates the direct binding of AGEs to H2B

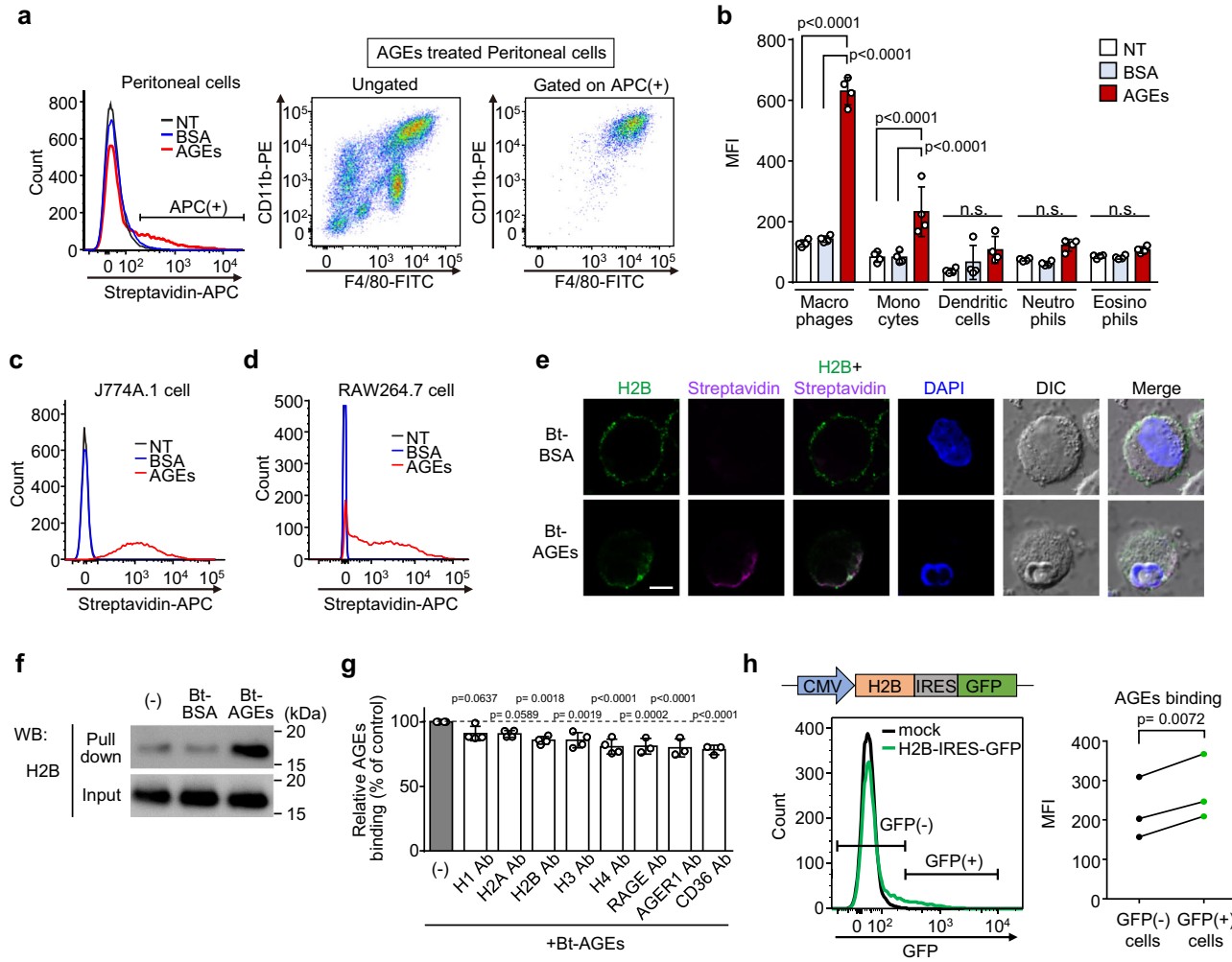

**Fig. 3 Histone-dependent binding of AGEs to macrophages. a** Binding of AGEs to mouse peritoneal cells. Left histogram, peritoneal cells were incubated alone (black line) or with Bt-BSA (blue line) or Bt-AGEs (red line), stained with streptavidin-APC, and analyzed by flow cytometry. Right two panels, Stained cells with CD11b and F4/80 were analyzed without or with gating on APC (+). **b** Cell-type specificity for AGEs binding. Peritoneal cells were incubated alone (NT) or with Bt-BSA (BSA) or Bt-AGEs (AGEs) and stained with streptavidin-BV421. Macrophages, Monocytes, Dendritic cells, Neutrophils, Eosinophils were gated and the bindings of Bt-BSA or Bt-AGEs were expressed as median fluorescent intensity (MFI) of BV421. Data are mean ± S.D. ($n = 4$, biologically independent experiments). Tukey–Kramer tests (two-sided), n.s. not significant ($p > 0.05$). **c, d** Binding of AGEs to J774A.1 (**c**) and RAW264.7 (**d**) cells. The cells were incubated alone (black line) or with either Bt-BSA (blue line) or Bt-AGEs (red line) and analyzed by flow cytometry. **e** Colocalization of H2B with AGEs. J774A.1 cells were treated with either Bt-BSA or Bt-AGEs and stained with anti-H2B antibody and streptavidin-alexa568. Scale bar, 10 µm. Representative image of three independent experiments. **f** Pull-down assay for the detection of H2B as a cell-surface AGEs-binding protein. J774A.1 cells were incubated with Bt-BSA or Bt-AGEs and then incubated with DTSSP. The detergent-resistant membrane fractions were subjected to pull-down, and the resulting precipitates were detected by western blotting using an anti-H2B antibody. Representative image of two independent experiments. **g** Inhibition of AGEs binding by anti-histones or anti-AGEs receptors antibodies. Cells were preincubated with antibody against histone H1, H2A, H2B, H3, H4, RAGE, AGER1, or CD36 prior to the treatment with Bt-AGEs. Relative AGEs bindings were determined by the MFI of APC. Data are mean ± S.D. ($n = 4$ for histone antibodies and $n = 3$ for known AGEs receptors antibodies, biologically independent experiments). Dunnett's test (two-sided), relative to the control. **h** Effect of H2B overexpression on AGEs binding. J774A.1 cells were transfected with a bicistronic construct encoding H2B followed by IRES driving the translation of GFP (upper left panel) and the binding of Bt-AGEs was analyzed ($n = 3$, biologically independent experiments). Paired $t$ test (two-sided).

localized in the membrane. We next assessed the contributions of histones and the other AGEs receptors to AGEs binding. Figure 3g shows that the treatment with anti-H2B, H3, or H4 antibodies significantly inhibited AGEs binding to macrophages to about the same extent as anti-AGE receptor (RAGE, AGER1, or CD36) antibodies. To study H2B-mediated AGEs binding to cells, J774A.1 cells were transiently transfected with a bicistronic construct encoding H2B, followed by an internal ribosomal entry site (IRES) that drives translation of GFP. As shown in Fig. 3h (lower left and right panels) and Supplementary Fig. S9, AGEs binding

was significantly enhanced in GFP(+) cells expressing exogenous H2B compared to GFP(−) cells.

H2B is known to localize on the cell surface by binding to phosphatidylserine[13]. To demonstrate the involvement of H2B in the binding of AGEs to cells more directly, we tested the effect of the addition of recombinant H2B treated extracellularly on the binding of AGEs to cells. The treatment of peritoneal cells with recombinant H2B enhanced the binding of H2B to cells, particularly macrophages, monocytes, and dendritic cells, resulting in significant increase in AGEs binding (Supplementary

Fig. S10). These data indicate that histone H2B contributes, at least in part, to the binding of AGEs to cells.

**AGEs inhibit Plg binding to macrophages**. The Plg/plasmin system plays important physiological and pathological roles in many biological processes. Data from several studies indicate that the interaction of Plg with cell-surface Plg receptors, including histone H2B, accelerates conversion of Plg to plasmin[14] and enhances the catalytic activity of plasmin[15]. In addition, plasmin formed on the cell surface is retained on the cell membrane and protected from inactivation by its inhibitor[16]. To assess the significance of the interaction between AGEs and H2B, we examined the effect of the AGEs on the binding of Plg to macrophages and the following macrophage response associated with the Plg activity in vitro. AGEs, but not unmodified BSA, indeed showed a significant inhibitory effect on the binding of the Bt-labeled Plg to the histone protein immobilized on microtiter plates (Fig. 4a). This inhibition was not due to the binding of AGEs to Plg, as Plg specifically bound to histone H2B, but not to BSA or AGEs (Supplementary Fig. S11). Similar inhibitory effects were observed for other DHA-modified proteins (Supplementary Fig. S12). In addition, we assessed the inhibitory effect of AGEs on Plg binding to H2B using surface plasmon resonance. The result showed that the pretreatment with AGEs was sufficient to block the binding of Plg to H2B, even after washing with a buffer (Fig. 4b).

We next sought to identify the putative AGE-binding site involved in the inhibition of the H2B-Plg binding. Full length H2B (H2B-FL) and a series of N-terminal deletion mutants of H2B (Fig. 4c upper panel) were used to evaluate the binding of the Bt-labeled Plg in the presence of BSA or AGEs as a competitor. Plg binding to the immobilized H2B-ΔN30 (an H2B mutant lacking the N-terminal 30 amino acids) was inhibited by the addition of AGEs, as was the case in H2B-FL. However, the AGEs failed to inhibit the Plg binding to H2B-ΔN40 and H2B-ΔN50 (Fig. 4c lower panel). The result indicates that sequence 30–40 of H2B, which are structurally close to the Plg binding site (carboxyl-terminal lysine) (Supplementary Fig. S13), were important for the regulation of H2B-Plg binding by AGEs. To address whether AGEs affect the binding of Plg to the cell surface, we carried out a Plg binding assay in the presence or absence of AGEs. As shown in Fig. 4d, e, the binding of Bt-labeled Plg to macrophages was significantly inhibited by the AGEs. Similarly, it was also found that the binding of Plg to macrophages was significantly suppressed by the pretreatment with AGEs, even after washing prior to the addition of Plg (Supplementary Fig. S14). This inhibitory effect of AGEs on Plg binding was not due to the release of H2B from cell surface (Supplementary Fig. S15). As a downstream biological effect of this binding, we evaluated Plg activation using recombinant H2B and cell line J774A.1. Plasmin formation was promoted in the presence of either recombinant H2B or cells and was suppressed by the treatment with AGEs (Fig. 4f, g). Thus, it is evident that AGEs act on the Plg/plasmin system.

**AGEs regulate monocytes/macrophage recruitment**. The binding of Plg to H2B and subsequent cell surface plasmin production is involved in the migration of monocytes and macrophages to the site of inflammation in vivo[12]. Hence, we examined the effect of the AGEs on the inflammatory response using an in vitro infiltration assay. Treatment of J774A.1 cells in the upper chamber with Plg increased the total number of cells across the Matrigel, referred to as invaded cells, by approximately 8-fold compared to the control (control: 20.47 ± 5.482 cells; Plg: 169.1 ± 15.28 cells). This increase was significantly attenuated by the co-incubation with AGEs (111.1 ± 13.98 cells), while there

was no significant reduction by the co-incubation with the unmodified BSA (171.7 ± 3.811 cells) (Fig. 5a). No cytotoxicity and proliferative changes were observed with BSA or AGEs in the concentration range of 30~300 μg/ml when the cells were incubated for 24–72 h (Supplementary Fig. S16). Because RAGE, among AGEs receptor, is known to be involved in the inflammatory response by AGEs, we examined whether the inhibitory effect of AGEs on macrophage infiltration could be affected by anti-H2B or RAGE antibodies. As shown in Fig. 5b, anti-H2B antibody suppressed the AGEs-dependent inhibition of macrophage infiltration, but anti-RAGE antibody showed no effect.

For in vivo infiltration assay, wild type C57BL/6 mice were intravenously injected with either BSA or AGEs (10 mg/kg/day) daily and TG was intraperitoneally injected to induce a sterile inflammatory response. Peritoneal lavages were collected at 0, 24, 48, and 72 h after TG injection and analyzed by flow cytometry. TG treatment resulted in an increase in the numbers of F4/80+ Gr-1 low macrophages at 48–72 h and the effect of AGEs was observed at 72 h, indicating a significant decrease in the number of macrophages in TG-treated animals (Fig. 5c). A similar trend was also observed in monocytes, where the number of the infiltrating monocytes are increased at 24–48 h in TG-treated animals, but was significantly suppressed by AGEs at 48 h. No significant difference in the numbers of peritoneal neutrophils and eosinophils was observed at any time point between BSA and AGE treatments. To determine whether AGEs suppress macrophage infiltration through inhibition of Plg, we conducted experiments using the Plg activation inhibitor, tranexamic acid (TXA). As shown in Fig. 5d, the inhibitory effect of AGEs on macrophage infiltration was completely canceled by treatment with TXA. The treatment with AGEs did not show any significant effects on cell death and in situ proliferation of peritoneal macrophages (Supplementary Fig. S17).

We also performed in vivo analysis of downstream factors of Plg activation. Plg is known to play a role in the activation of matrix metalloproteinases (MMPs), which is essential for monocytes/macrophages motility and infiltration during the inflammatory response[17]. Therefore, we investigated whether the treatment with AGEs could attenuate the activation of MMP-9 in the peritonitis model. Western blot analysis of peritoneal lavage fluid from TG-stimulated mice revealed that treatment with AGEs significantly suppressed the activation of MMP-9, as determined by the ratio of truncated (active) forms of MMP-9 to intact (pro) MMP-9 (Fig. 5e). In addition, daily treatment with AGEs reduced mRNA levels of pro-inflammatory cytokines (i.e., IL-1β and IL-6) but upregulated the expression of anti-inflammatory cytokine and M2 macrophage markers (i.e., Arg-1, IL-10, and CD206) in peritoneal macrophages isolated from mice 72 h after TG injection (Fig. 5f). Similarly, RNA-seq analysis showed that several M1-associated genes (i.e., IL-1b, Ccr7, Igtp, and Sell) were significantly downregulated, whereas M2-associated genes (i.e., GATA2, Clec10a, Snn, Pcdh7, Stxbp6, and Gar1) were upregulated in peritoneal macrophages from the mice treated with AGEs compared to those with BSA (Supplementary Fig. S18). Changes in the protein expression of phenotypic markers of M1 and M2 macrophages were negligible (Supplementary Fig. S19). Furthermore, we employed cecal ligation and puncture (CLP)-induced sepsis model to clarify the role of AGEs in more pathophysiological setting and found that mice treated with AGEs displayed a significantly prolonged survival time, but not with unmodified BSA (Fig. 5g). These data indicated that AGEs have a function of regulating monocytes/ macrophage recruitment to the site of inflammation through inhibition of Plg activation, which may contribute to protection against inflammatory disorders. In other words, beside our common concept of AGEs as danger-associated molecular

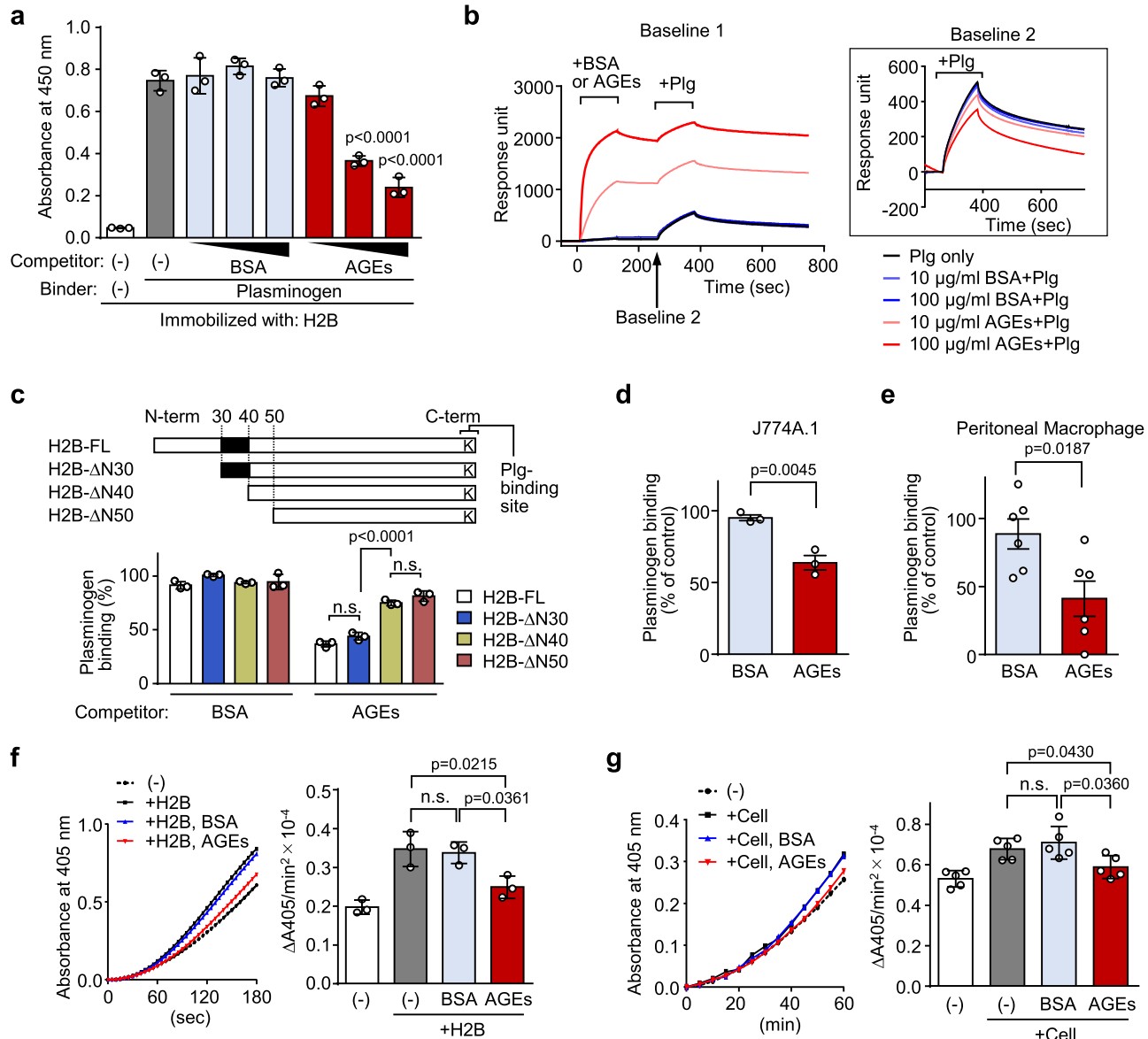

**Fig. 4 AGEs inhibit Plg binding to histone H2B and macrophages. a** Effect of AGEs on Plg binding to histone H2B in vitro. Recombinant H2B coated on ELISA plate was preincubated with either BSA or AGEs (1, 10, and 100 μg/ml), followed by treatment with Bt-Plg (1 μg/ml). Plg binding to H2B was detected with streptavidin-HRP. Data are mean ± S.D. of triplicate samples (representative of three independent experiments). Dunnett's test (two-sided), relative to the treatment with Plg alone. **b** Surface plasmon resonance measurements. The interactions of Plg (10 μg/ml) with H2B immobilized on a Biacore sensor chip NTA, following the addition of either BSA or AGEs (10 or 100 μg/ml) were monitored. **c** Schematic representation of full length and deletion mutants of H2B (upper panel) and effects of AGEs on Plg binding to deletion mutants of H2B (lower panel). H2B mutants coated on ELISA plate were preincubated with either BSA or AGEs (100 μg/ml), followed by treatment with Bt-Plg (1 μg/ml). Plg binding to H2B mutants was detected with streptavidin-HRP. The absorbance value in the absence of competitor was defined as 100% Plg binding. Data are mean ± S.D. of triplicate samples (representative of three independent experiments). Student's $t$ test (two-sided), n.s. not significant ($p > 0.05$). **d** Effect of BSA or AGEs on Plg binding to J774A.1 **e** Effect of BSA or AGEs on Plg binding to peritoneal macrophage. Cells were preincubated with either BSA, AGEs, or tranexamic acid (a Plg inhibitor). After further incubation with Bt-Plg, Plg binding to cells was analyzed by FACS with streptavidin-APC. Plg binding (% of control) was calculated as described in the Materials and Methods section. Data are mean ± S.D. ($n = 3$ and 6, respectively, biologically independent experiments). Student's $t$ test (two-sided). **f**, **g** Effect of AGEs on plasmin formation in the presence of recombinant H2B (**f**) or J774A.1 cells (**g**). Plasmin was measured by the absorbance at 405 nm using plasmin specific substrate S-2251. The rate of plasmin generation was calculated as described in the Materials and Methods section. Data are mean ± S.D. ($n = 3$ and 5, respectively, biologically independent experiments). Tukey–Kramer tests (two-sided), n.s. not significant ($p > 0.05$).

patterns mediating a pro-inflammatory response, AGEs may also function as an anti-inflammatory molecule.

## Discussion

AGEs are prevalent in the diabetic vasculature and contribute to the development of atherosclerosis and are generally believed to be endogenous damage-associated molecular patterns, mediating pro-inflammatory responses. Therefore, the general consensus is that the formation and accumulation of AGEs reflect the proportion of modified proteins with impaired function and may be at the root of disease and age-related functional losses. However, little is known about the physiological and pathophysiological roles of

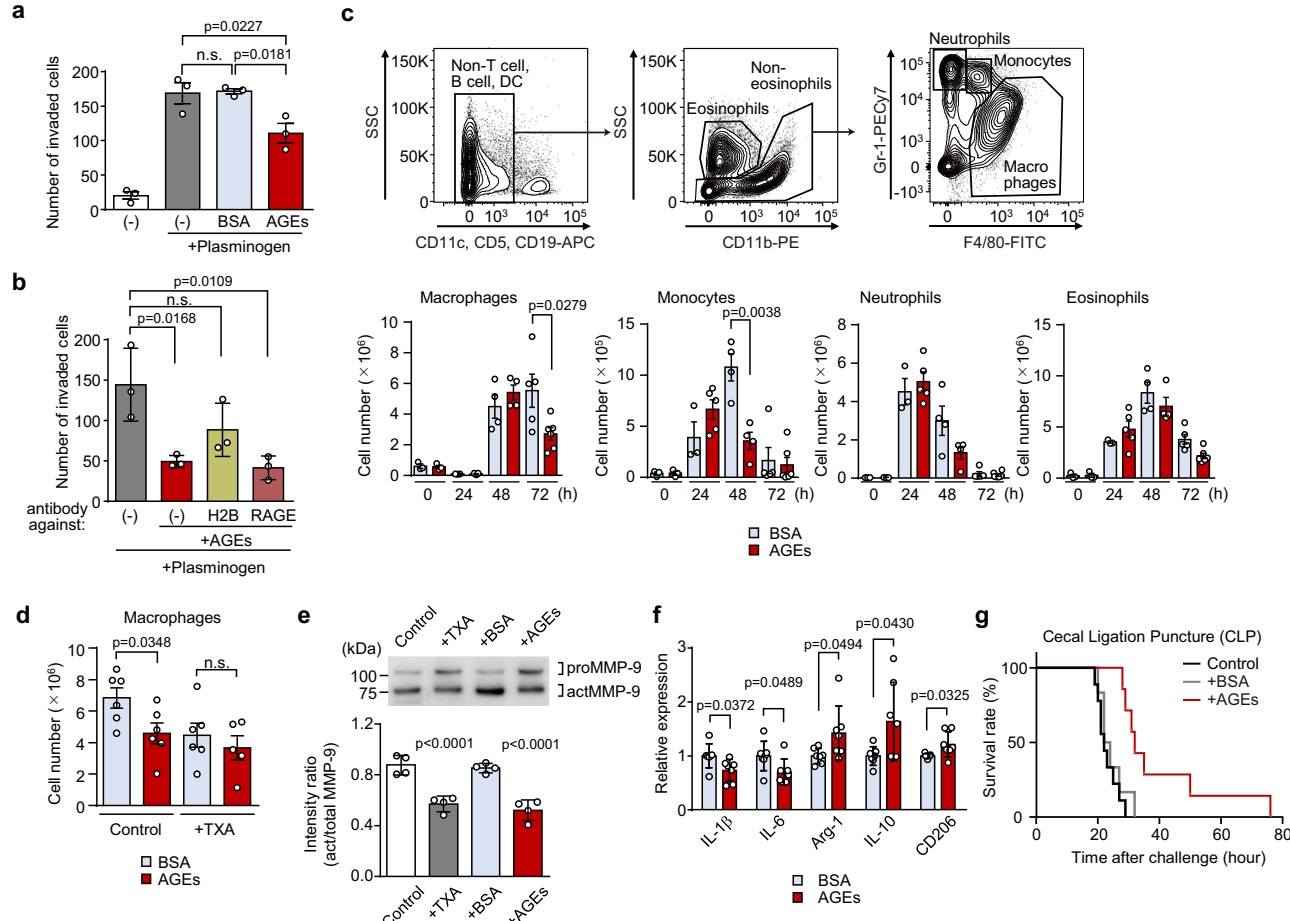

**Fig. 5 AGEs regulate monocyte/macrophage recruitment to the inflammatory sites. a** In vitro infiltration assay. J774A.1 cells were incubated alone or with BSA or AGEs in the presence of Plg. The infiltration activity was assessed by the number of cells across the Matrigel (mean ± S.D., $n = 3$, biologically independent experiments). Tukey–Kramer tests (two-sided), n.s. not significant ($p > 0.05$). **b** Effect of H2B or RAGE on AGEs-dependent suppression of infiltration. Cells were preincubated with anti-H2B or anti-RAGE antibody and the infiltration activity was assessed (mean ± S.D., $n = 3$, biologically independent experiments). Tukey–Kramer tests (two-sided), n.s., not significant ($p > 0.05$). **c** Effect of BSA or AGEs on the peritoneal infiltration. BSA or AGEs was given daily by intravenous administration and the number of peritoneal cells was determined 0, 24, 48, and 72 h after TG injection (mean ± S.D., $n = 3$ for BSA-0 h, AGEs-0 h, and BSA-24 h, $n = 4$ for BSA-48 h and AGEs-48 h, $n = 5$ for AGEs-24 h and BSA-72 h, $n = 6$ for AGEs-72 h, biologically independent experiments). Student's $t$ test (two-sided). **d** Inhibition of Plg activation abrogate the effect of AGEs on macrophage infiltration. Mice were administrated with TXA, and the number of peritoneal macrophages was determined 72 h after TG injection (mean ± S.D., $n = 5$ for AGEs+TXA group and $n = 6$ for other groups, biologically independent experiments). Student's $t$ test (two-sided), n.s. not significant ($p > 0.05$). **e** Effect of AGEs on MMP-9 activation. MMP-9 in peritoneal lavage was purified and detected with MMP-9 antibody (upper panel), and act/proMMP-9 ratio were quantified (lower panel; mean ± S.D., $n = 4$, biologically independent experiments). Dunnett's test (two-sided), relative to the control. **f** Expression levels of proinflammatory and anti-inflammatory macrophage markers in peritoneal macrophage. Data are relative expression to BSA-treated mice (mean ± S.D., $n = 7$, biologically independent experiments). Student's $t$ test (two-sided). **g** Effect of AGEs on sepsis model. Mice were treated with saline ($n = 9$, black line), BSA ($n = 6$, gray line), or AGEs ($n = 7$, brown line) 4 h after the CLP procedure. There were significant differences in the treatment with saline vs. AGEs and with BSA vs. AGEs, but not with saline vs. BSA ($p = 0.0004$, 0.0099, and 0.4154, respectively, log-rank test).

AGEs. In the current study, we examined the presence of cell surface proteins that are capable of binding AGEs and identified histone as a hitherto unrecognized AGE-binding protein. To the best of our knowledge, this is the first report to describe a previously unknown function of histone as a binding protein for damage-associated molecules, such as AGEs. We also found that the binding of AGEs to the membrane-associated histone resulted in suppression of Plg binding and subsequent inhibition of monocytes/macrophage recruitment to the site of inflammation (Supplementary Fig. S20). The finding of AGEs as a potential histone ligand suggests that, beside our common notion of AGEs as a pro-inflammatory mediator, AGEs may constitute a previously unrecognized, but important class of anti-inflammatory ligand. This study, therefore, represents a paradigm shift in glycation

biology that focuses on AGEs as pro-inflammatory, pathogenic molecules.

It was fully unexpected that histone proteins have a receptor function for damage-associated molecules, such as AGEs. The role of histones as AGEs receptors has been overlooked due to the fixed notion that histones are only present in nuclei. Histone proteins, even if detected in the lipid raft fraction, may have been considered contamination from nuclear fractions. Histones are the protein components of the nucleosome core particle and are composed of two copies of each of the histones H2A, H2B, H3, and H4. Based on the identification of histone as an AGE-binding protein, we investigated the binding specificity of histone to AGEs using the recombinant proteins (H1, H2A/B heterodimer, and H3/4 heterotetramer) in vitro. The pull-down and solid phase

binding assays revealed that the histone proteins had a similar binding affinity to AGEs. In addition, the specificity of the interaction between AGEs and the individual histone protein was also demonstrated by surface plasmon resonance. These data indicate that the proteins form a complex even in the fluid phase (Fig. 1). These data also suggest the presence of a common structural property in the histone proteins for the binding to the AGEs.

On the other hand, the finding that the DHA-modified Bt-PA showed significant binding affinity for H2B suggested that some specific structures of AGEs might be involved in histone recognition. We attempted to identify DHA adducts with H2B-binding activity and showed that the broad spectrum of the fractions eluted from 17–22 min showed binding specificity for histone proteins (Fig. 2b). In addition, these histone-binding potency was closely correlated with recognition by anti-AGEs antibodies. These data suggest that multiple products with an affinity for H2B are likely to be present in AGEs. Bt-PA modified with glucose and its metabolites also showed affinity for H2B (Fig. 2c), suggesting that histone binding activity might be a common property of AGEs. On the other hand, based on the facts that glycolaldehyde is an intermediate generated through oxidative degradation of vitamin C and sugar fragmentation during glycation[18,19] and shows a facile reactivity with the lysine amino groups of proteins to form AGEs, we tested the histone ligand activity of two authentic glycolaldehyde-lysine adducts, GA-pyridine[20] and $N^\varepsilon$-pyrrolelysine[21], and found that H2B had an affinity with GA-pyridine (Supplementary Fig. S21). However, we have not been able to detect GA-pyridine in the AGEs we used for this study. Thus, the structure of histone ligands generated during glycation of proteins remain largely unknown.

With respect to the binding mode of H2B and AGEs, previous studies have shown that the modification of proteins by DHA is associated with an increase in the net negative charges of the protein due to the modification of lysine residues, causing a structural similarity to dsDNA backbone composed of negatively charged phosphate groups[4]. Indeed, it has been suggested that the binding of AGEs to complement component C1q may be due to the electronegative potential of AGEs[9]. In addition, one of the major common features of known RAGE ligands regarding the recognition mechanism of AGEs receptors is the net negative charge at physiological pH[22]. These finding and the fact that histone proteins are highly basic proteins suggest that complex formation between histone and AGEs may be driven by electrostatic interactions. This may be similar to the proposed mechanism for the interaction of phosphatidylserine with the highly polar $N$-terminus of H2B[13]. To validate this speculation, we tested acylated proteins and confirmed that they indeed interacted with H2B (Fig. 2f). Furthermore, the possible contribution of electrostatic interaction was also demonstrated by the observations that the binding of AGEs to H2B was inhibited by 1 M NaCl (Fig. 2e) and by the deletion of the highly basic $N$-terminal region (ΔN35) from H2B (Fig. 2g). In addition, AGEs were recognized by scrambled fragments of H2B $N$-terminal containing the same amino acids in different amino acid sequences (Fig. 2g). These data strongly support our hypothesis that the electronegative potential of AGEs might be involved, at least in part, in the recognition by histone. The high affinities of histones for electronegative molecules could be brought by the combined electrostatic interactions between the $N$-terminal arginines plus lysines of histones and electronegative molecules in the same manner as reported for the binding of histone tail with DNA[23]. However, lysozyme, a basic protein with an isoelectric point similar to histone proteins, did not show any recognition specificity for AGEs (Supplementary Fig. S6). Similarly, given the reduced binding of AGEs to scrambled H2B $N$-terminal

fragments compared to original H2B $N$-terminal (Fig. 2g), some specific structures of histones, in addition to the electrostatic potential, may contribute to binding to AGEs.

Cell-surface expression of histone H2B has been reported to occur in a variety of cells, including human blood monocytes/macrophages, activated lymphocytes, and apoptotic cells[11,24,25]. Although H2B is normally assembled as core histones within the nucleosome, the membrane translocation of the histone occurs in association with changes in calcium-dependent protein trafficking[26], after which H2B tethers to the cell surface by interacting with phosphatidylserine and functions as a Plg receptor involved in the inflammatory response[13]. Previous reports have indicated that histones can bind not only to phosphatidylserine, but also to negatively charged cell surface molecules such as heparan sulfate proteoglycans[27]. Heparan sulfate-carrying core proteins, such as syndecan-4, have been shown to localize in lipid raft compartment[28], and their expression levels vary among cell types; activated macrophages, monocytes, and dendritic cells display high expression of syndecan-4, while neutrophils and eosinophils express low levels of syndecan-4[29,30]. These previous findings and our observations that the treatment with recombinant H2B enhanced the association of H2B to the cells (Supplementary Fig. S10) suggest that histones may also interact with highly sulfated glycosaminoglycans through charge interactions. Enhanced cell surface expression of H2B is associated with increased Plg binding and plasmin generation, which contributes to the recruitment of inflammatory cells[12]. Lysine analogues, such as ε-aminocaproic acid and TXA, have been shown to compete with Plg to bind histone, resulting in inhibition of macrophage migration[12]. In the present study, like these competitors, AGEs were shown to inhibit the binding of Plg to macrophages and the following macrophage response associated with Plg activity (Fig. 4d–g). In addition, in a mouse model of peritonitis, intravenous injection of AGEs significantly reduced the number of infiltrated monocytes after stimulation with TG (Fig. 5c). It was also shown that this inhibitory effect was brought about by the mechanism through inhibition of Plg activation by AGEs (Fig. 5d, e). Suppressing the infiltration of monocytes into tissues can lead to protection against inflammatory disorders, such as peritonitis[31], autoimmune encephalomyelitis[32], atherosclerosis[33], and multiple organ failure in sepsis[34]. Hence, we tested whether AGEs play a protective role in the disease progression using the CLP model, the animal model most relevant to clinical sepsis, and revealed that the AGEs treatment significantly prolonged survival time of septic mice (Fig. 5g). Therefore, AGEs may generally contribute to the regulation of the inflammatory response and disease severity by inhibiting the recruitment of monocytes/macrophage to the site of inflammation through inhibition of Plg activation.

Despite all these accumulated data, direct evidence of the contribution of the interaction between AGEs and histone H2B in physiological conditions is still limited. This limitation may be due to a lack of methodology for specifically inhibiting the interaction between AGEs and H2B. In this regard, we attempted to use antibodies to block this interaction. However, an anti-H2B antibody showed a significant but only modest decrease in AGEs binding (Fig. 3g). This is likely to be due to redundancy and functional compensation between the different types of histones in binding of AGEs. In addition, as an alternative approach, we attempted to inhibit the interaction between AGEs and H2B through genetic manipulations using truncated H2B. However, it was not feasible because the expression of the truncated H2B induced marked cell death (Supplementary Fig. S22). Another concern is the relationship between cell-surface histones and other AGEs receptors. In this regard, due to the high affinity between AGEs and histones, AGEs are expected to bind immediately to histones on the cell surface in addition to other AGEs receptors. Therefore, it can be expected that cell-surface histones

may compete with other AGEs receptor for binding to AGEs, resulting in the downregulation of AGEs-induced inflammatory responses via other AGEs receptors. On the other hand, there may be functional interactions between histones and AGEs receptors. An attractive hypothesis is that there may be an intracellular crosstalk of downstream signaling mediated by histone and other AGEs receptors. However, the presence of such intracellular signaling mechanisms induced by AGEs remains unexplored. Further study is under way in our laboratory to elucidate specific regulatory mechanisms of AGEs-histone interactions and to investigate functional interactions between histone and other AGE receptors.

In conclusion, the present study identifies histone as an AGE-binding protein and provides multiple lines of evidence that AGEs may contribute to biological events associated with the innate immune and inflammatory processes. These findings and the apparent presence of AGEs in vivo provide a possible link between glycation and innate immunity. However, the importance of this interaction in the pathogenesis of inflammatory diseases remains unclear. Therefore, future work is needed to understand the interaction of AGEs with histones in mediating the biological effects of AGEs and to determine whether the AGEs/histone complex contributes to vascular inflammation in vivo. Furthermore, the findings described here should also lead to new strategies or areas for therapeutic intervention in chronic inflammatory diseases in general.

## Methods

**Animals**. We used 8- to 12-week-old male C57BL/6 mice (SLC Japan) for the animal experiment.

All animal studies were approved by the Institutional Animal Care and Use Committee at the University of Tokyo (Permission No. P19-020).

**Modification of proteins and biotin-lebeled N-pentylamine**. Bt-labeled BSA (Bt-BSA) was prepared by incubating 5 mg/ml BSA (Iwai Chemicals) with a 10-fold molar Bt-PE-maleimide (Dojindo Laboratories) in PBS at 25 °C for 16 h. After incubation, the aliquots were dialyzed against PBS. Biotin-labeled Plg (Bt-Plg) was prepared with EZ-Link Sulfo-NHS-LC-Biotin (Thermo Fisher Scientific) following the manufacturer's instructions. The AGEs were prepared according to a previous report[9] by incubating 1.0 mg/ml BSA with 25 mM DHA (Sigma) in PBS at 37 °C under atmospheric oxygen. After 72 h, the reactants were collected and dialyzed with PBS. For the preparation of Bt-labeled acylated proteins (Ac-BSA, Sc-BSA, and Ma-BSA), 5 mg/ml Bt-BSA was added to equal volume of a saturated solution of sodium acetate with continuous stirring on ice, followed by the incubation at 4 °C for 1 h with 2 mM acetic anhydride, succinic anhydride, or maleic anhydride, respectively. After incubation, the aliquots were dialyzed against PBS. The protein concentrations were measured by a BCA assay (Nacalai Tesque Inc.). DHA-modified Bt-labeled N-pentylamine (DHA-Bt-PA) was prepared upon incubation of 6 mM EZ-Link pentylamine biotin (Thermo Fisher Scientific) with 25 mM DHA at 37 °C for 72 h.

**Mass spectrometry analysis of AGEs**. A mass spectrometry analysis of the unmodified BSA and AGEs was performed using a MALDI-TOF/TOF mass spectrometer (Autoflex Speed MALDI TOF/TOF system, Bruker Daltonics). Each protein (200 μg/ml) was mixed 1:1 with sinapinic acid MALDI matrix (saturated sinapinic acid in 30% acetonitrile and 0.07% trifluoroacetic acid) and spotted directly onto a ground steel MALDI target plate (Bruker Daltonics). MALDI-TOF mass data were collected in linear positive mode, and the spectra were recorded between 10,000 and 200,000 m/z. Mass calibration was performed using Bruker Protein Standard II (Bruker Daltonics).

**Preparation of RIPA soluble and detergent-resistant membrane (lipid raft) fractions from mouse splenocytes**. The mice were sacrificed, their spleens removed, homogenized in RPMI-1640 medium, and washed by centrifugation at $300 \times g$ for 5 min. The splenocyte suspensions were further treated with an ammonium chloride-potassium red blood cell (RBC) lysing solution and washed with RPMI-1640 to remove the lysed RBCs. The RIPA soluble and detergent-resistant membrane (lipid raft) fractions were isolated using the UltraRIPA kit according to the manufacturer's instructions (BioDynamics Laboratory). Briefly, the splenocytes were first homogenized in RIPA buffer containing 50 mM Tris-HCl (pH8.0), 150 mM NaCl, 1% NP-40, 0.1% SDS, and 0.5% sodium deoxycholate. The samples were sonicated, then centrifuged at $12,000 \times g$ for 5 min and the supernatants were used as the RIPA soluble fraction. The pellets were washed with RIPA

buffer, resuspended in buffer B (BioDynamics Laboratory), centrifuged at $12,000 \times g$ for 5 min and the supernatants were used as the lipid raft fraction. The protein concentrations were measured by a BCA assay.

**Identification of AGEs-binding proteins**. Both the Bt-BSA and Bt-AGEs were incubated with streptavidin-coupled magnetic beads (Dynabeads MyOne Strepta-vidin T1, Thermo Fisher Scientific). After washing three times with PBS and Tween 20 (PBST), the beads were incubated with proteins (100 μg) in RIPA soluble or lipid raft fraction in PBS containing a protease inhibitor cocktail (Roche Diagnostics) with rotation at 4 °C for 1 h. The beads-protein complex was washed three times with PBS, and the binding proteins were eluted with 50 mM glycine (pH 2.8). The proteins were separated by 5–20% SDS-PAGE and stained using a SilverQuest Silver Staining Kit (Invitrogen) according to the manufacture's protocol. After gel staining, the AGEs binding protein bands were subjected to in-gel trypsin digestion, and the peptides were desalted then analyzed by LC-MS/MS (LC and auto-sampler: Zaplous Advance nano UHPLC HTS-PAL xt System, AMR, MS:Orbitrap VELOS ETD, ThermoFisher Scientific). The proteins were identified with the Proteome Discoverer 2.1 searching algorithms using the database of Mus Musculus from Uniprot.

**Cell-surface biotinylation**. For cell-surface biotinylation, splenocytes were labeled with membrane impermeable EZ-Link Sulfo-NHS-LC-Biotin (Thermo Fischer Scientific) according to the manufacturers' instructions. Briefly, $3 \times 10^7$ cells were incubated with 2 mM EZ-Link Sulfo-NHS-LC-Biotin for 30 min on ice. Cells were washed three times with ice-cold PBS containing 100 mM glycine, separated into RIPA soluble or lipid raft fraction. The cell lysates were subjected to pull-down with streptavidin-coupled magnetic beads, and the resulting precipitates were subjected to western blotting.

**Expression and purification of recombinant histones and MBP-fusion peptides**. The pET15b vectors carrying the His-tagged human histone H2A, H2B, H3, and H4 and the pET21a vector carrying the His-tagged human histone H1.2 were kindly provided by Dr. Kurumizaka (The University of Tokyo, Japan). Expression and purification of the recombinant histones were performed according to previous reports[35–37]. Briefly, the E. coli BL21 (DE3) cells were freshly transformed with the vectors described above and grown on LB plates containing ampicillin (100 μg/ml) at 37 °C. After 16 h incubation, the colonies were inoculated into LB medium containing ampicillin. After overnight cultivation, the cells were harvested and resuspended in 50 mM Tris-HCl (pH 8.0), 500 mM NaCl, 1 mM phe-nylmethylsulfonyl fluoride (PMSF), and 5% glycerol. The cells were disrupted by two rounds of sonication for 15 min each. The cell lysates were centrifuged at $18,000 \times g$ for 20 min at 4 °C. The His-tagged histone H2A, 2B, H3, and H4 were recovered in the insoluble pellets, and H1 was in the supernatant. The pellets were dissolved in 50 mM Tris-HCl (pH 8.0), 500 mM NaCl, 1 mM PMSF, 5% glycerol, and 7 M guanidinium chloride by sonication and overnight incubation at 4 °C. After centrifugation, the supernatants containing the His-tagged histones were mixed with Ni-NTA agarose beads (FUJIFILM Wako Pure Chemical., Inc.), and the samples were rotated at 4 °C for 60 min. The beads were packed into Econo-columns (BioRad) and washed with 50 mM Tris-HCl (pH 8.0), 500 mM NaCl, 1 mM PMSF, 5% glycerol, 6 M urea, and 5 mM imidazole. The His-tagged histones were eluted with the same buffer except that the concentration of imidazole was 500 mM. The eluates were dialyzed against 5 mM Tris-HCl (pH 8.0) and 2 mM 2-mercaptoethanol at 4 °C overnight.

For the purification of the MBP-fusion peptides, peptide sequences of H2B fragments, H2B N-terminal (1-35), H2B-ΔN35 (an H2B mutant lacking the N-terminal 35 amino acids), and scrambled H2B N-terminal (scrambled 1, 2, and 3) (the sequences are shown in Supplementary Table S2), were amplified and cloned into the pMAL-p2X expression vector. E. coli BL21 (DE3) cells were transformed with vectors and isolated colonies cultured in LB medium containing ampicillin (100 μg/ml) at 37 °C. After overnight cultivation, cells were inoculated to 200 ml of LB medium and incubated at 37 °C until the OD600 reached 0.4–0.5 when expression was induced by the addition of iso-propyl-β-D-thiogalactopyranoside to a final concentration of 0.3 mM. After a 2 h incubation, cells were centrifuged and the bacterial pellets were resuspended with an osmotic shock solution (30 mM Tris-HCl, 20% sucrose, 1 mM EDTA, pH 8.0). The suspensions were incubated for 10 min with shaking followed by centrifugation at $8000 \times g$ for 20 min at 4 °C. The supernatants were removed, the pellets were resuspended in ice-cold 5 mM MgSO4, incubated for 10 min on ice, and centrifuged at $8000 \times g$ at 4 °C for 20 min. The supernatants were supplemented with 1 M Tris-HCl, pH 7.4 to a final concentration of 20 mM, applied to amylose resin columns, washed with column buffer (20 mM Tris-HCl, 200 mM NaCl, 1 mM EDTA, pH 7.4), and eluted with elution buffer (20 mM Tris-HCl, 200 mM NaCl, 1 mM EDTA, 10 mM maltose, pH 7.4). The fractions containing the MBP fusion peptides were pooled and dialyzed against PBS.

**Pull-down assay**. Bt-BSA or Bt-AGEs (10 μg) were incubated with recombinant histones (10 μg) in 100 μl of binding buffer containing 20 mM Tris-HCl (pH 7.5), 150 mM NaCl, 1 mM MgCl2, and 0.1% NP40 at 4 °C for 1 h, then streptavidin-coupled magnetic beads were added. After a 15 min incubation with rotation, beads

were washed three times with a washing buffer containing 20 mM Tris-HCl (pH 7.5), 500 mM NaCl, 1 mM MgCl₂, 0.1% NP40, 1% TritonX-100, and 0.1% SDS. Binding proteins were eluted with 50 mM glycine (pH 2.8), separated by 5–20% SDS-PAGE, and stained with CBB.

**Solid-phase binding assay.** Recombinant histones or MBP-fusion peptides (10 µg/ml) were immobilized onto 96-well MaxiSorp plates (Nunc) at 4 °C overnight. The plates were blocked with 2% skim milk in PBST for 1 h, washed three times with PBST, then either the Bt-BSA, modified Bt-BSA, Bt-Plg, Bt-PA, or modified Bt-PA was added to the wells. After 1 h incubation, streptavidin-HRP in PBST was added, incubated for 1 h and then developed using TMB Ultra substrate (Thermo Scientific). The binding of the biotinylated proteins to histones or MBP-fusion peptides was quantified by measuring the absorbance at 450 nm. For the competitive assay, the BSA or AGEs were preincubated for 30 min on H2B coated plates before the addition of Bt-Plg. To investigate whether DHA-modified Bt-PA could be recognized by anti-AGEs antibodies the culture supernatant of hybridoma (clone: BDM1)[8] was collected, and the monoclonal antibody was purified using HiTrap IGM purification column (Cytiva, Sweden,17-5110-01) according to the manufacturer's protocol, and then dialyzed against PBS. The purified antibody (10 ng/ml) was immobilized onto 96-well MaxiSorp plates at 4 °C overnight and the binding of DHA-modified Bt-PA was assessed as mentioned above.

**Surface plasmon resonance.** The surface plasmon resonance assays were performed using a Biacore T200 instrument (Cytiva). The His-tagged histone proteins were immobilized on a sensor chip NTA (Cytiva). All the solutions were freshly prepared and filtered through membranes with 0.22 µm pores. The running buffer was 10 mM HEPES buffer containing 150 mM NaCl and 0.05% Tween-20 for all the experiments. The interaction between the immobilized histones and either the unmodified BSA or modified BSA (AGEs) at the concentrations of 6.25, 12.5, 25, 50, and 100 µg/ml were examined at 25 °C with a flow rate of 30 µl/min. For the measurement of inhibitory effect of AGEs on plasminogen binding to H2B, BSA or AGEs (10 or 100 µg/ml) was injected on the H2B-immobilized sensor chip, and after washing with the running buffer, Plg (10 µg/ml) was injected with a flow rate of 30 µl/min. The response curves obtained from the control flow cell (without immobilized histones) and from injecting only buffer were subtracted from the histone-immobilized cell to correct for any nonspecific binding.

**HPLC.** For the fluorescence detection and fractionation using HPLC, the reaction mixture of DHA-Bt-PA (300 µl) was into the reverse-phase column (Sunniest C18, 5 um, 6 × 250 mm), equilibrated in a solution of 2% acetonitrile containing 0.1% trifluoroacetic acid (TFA), and eluted using 2–60% acetonitrile containing 0.1% TFA from 5–25 min and 100% acetonitrile containing 0.1% TFA from 25–30 min. All the analyses were performed at the flow rate of 0.8 ml/min, and effluents were monitored with absorbance at 200–600 nm and separated into 21 fractions from 5 to 25 min (1 min/fraction). Each fraction was lyophilized and reconstituted in 300 µl PBS and subjected to a solid-phase binding assay.

**Measurement of Nᵉ-(Carboxymethyl)Lysine (CML) in DHA-modified BSA using LC-MS/MS.** The CML content in sample was measured by LC-MS/MS using a TSQ Quantiva triple-stage quadrupole mass spectrometer (Thermo Fisher Scientific), as described previously[38]. Briefly, 5 µg of BSA or DHA-BSA were reduced with 100 mM sodium borohydride in 50 mM sodium borate buffer (pH 9.1) at 25 °C for 4 h. Standard [²H₂] CML (PolyPeptide Laboratories) and [¹³C₆] lysine (Cambridge Isotope Laboratories) were added to samples, which were then hydrolyzed with 1 mL of 6 M HCl at 100 °C for 18 h. The dried samples were resuspended in 1 mL of distilled water and passed over a Strata-X-C column (Phenomenex, Torrance), which was pre-washed with 1 mL of methanol and equilibrated with 1 mL of distilled water. The column was then washed with 3 mL of 2% formic acid and eluted with 2 mL 7% ammonia. The pooled elution fractions were dried and resuspended in 1 mL 20% acetonitrile containing 0.1% formic acid. The samples (10 µL) were subjected to LC-MS/MS. LC was conducted on a ZIC®-HILIC column (150 × 2.1 mm, 5 µm; Millipore). The mobile phase consisted of solvent A (distilled water containing 0.1% formic acid) and solvent B (acetonitrile containing 0.1% formic acid). The flow rate was 0.2 mL/min, and the column was maintained at 40 °C. The retention times for CML and lysine were approximately 12 and 14 min, respectively. CML, lysine, [²H₂] CML, and [¹³C₆] lysine were detected by electrospray ionization and positive ion mass spectrometric multiple reaction monitoring. The parent ions of CML, [²H₂] CML, lysine and [¹³C₆] lysine were 205 (m/z), 207 (m/z), 147 (m/z) and 153 (m/z), respectively. Fragment ions of 130 (m/z) from each parent ion were measured for the analysis of CML and [²H₂] CML in the samples. Lysine and [¹³C₆] lysine were measured as fragment ions of 84 (m/z) and 89 (m/z) from parent ion, respectively. The CML contents was normalized to the lysine content; thus, the data were expressed as mmol/mol lysine.

**Preparation of mouse peritoneal cells.** Mice were intraperitoneally administrated with 500 µl of a 4% TG solution (DIFCO Laboratories). After 72 h or the indicated time periods, ice-cold PBS (4 ml) was injected into the peritoneal cavity and 2.5 ml of lavage was collected. The peritoneal cells were washed once with ice-cold PBS and subjected to a flow cytometric analysis. For the analysis of the peritoneal cell

recruitment, either BSA or the AGEs (10 mg/kg/day) was daily provided by intravenous administration starting 24 h before the TG injection and continuing until the mice were sacrificed. For the experiments under the inhibitory condition of Plg activation, TXA (Sigma) was administrated in the drinking water at 20 mg/ml 48 h prior to thioglycollate injection and throughout the experiment. For isolation of the peritoneal macrophages, peritoneal exudate cells collected 72 h after the TG injection were incubated in RPMI-1640 for 2 h at 37 °C, wash twice with PBS, and the adherent cells (peritoneal macrophages) were collected using an enzyme-free cell dissociation solution (Sigma).

**Flow cytometry.** J774A.1 cells and RAW264.7 cells were obtained from the Japanese Cancer Research Resources Bank (JCRB, Cell Number: JCRB 9108) and American Type Culture Collection (ATCC, Cell Number: TIB-71), respectively. For the binding analysis, cell suspensions were preincubated with anti CD16/CD32 mAb (1:50 dilution, 93, Biolegend) to block the FcγRII/III receptors and incubated with either 100 µg/ml Bt-BSA or Bt-AGEs at 4 °C for 15 min in HBSS (Hanks balanced salt solution)-HEPES buffer (Invitrogen) containing 0.1% BSA and 0.1% sodium azide (HBSS-BSA). The cells were washed and stained at 4 °C for 15 min with the following fluorochrome-conjugated mAb: FITC-labeled anti-F4/80 (1:200 dilution, BM8, Biolegend); PE-labeled anti-CD11b (1:200 dilution, M1/70, BD Biosciences) and APC streptavidin (1:200 dilution, BD Biosciences) to reveal the Bt-conjugated proteins. After washing three times with HBSS-BSA, the stained cells were resuspended in 2.5 µg/ml 7-aminomycin D (7-AAD, BD Biosciences) to exclude the dead cells. For the analysis of the peritoneal cells, the cell suspensions were stained with FITC-labeled anti-F4/80 (BM8); PE-labeled anti-CD11b (M1/70); PECy7-labeled anti-Gr-1 (1:200 dilution, RB6-8C5, Biolegend) and APC-labeled anti-CD11c (1:200 dilution, HL3, BD Biosciences), CD5 (1:200 dilution, 53-7.3, Biolegend), or CD19 (1:200 dilution, 6D5, Biolegend). After washing three times with HBSS-BSA, the stained cells were analyzed by FACSVerse or FAC-SAriaII (BD Biosciences). For blocking experiment, cells were preincubated with one of the following antibodies (4 µg/ml) at 4 °C for 15 min against histone H1 (H-2, Santa Cruz), histone H2A (938CT5.1.1, Santa Cruz), histone H2B (A-6, Santa Cruz), histone H3 (1G1, Santa Cruz), histone H4 (F-9, Santa Cruz), RAGE (A-9, Santa Cruz), AGER1 (E-9, Santa Cruz), or CD36 (SMφ, Santa Cruz) prior to the treatment with Bt-AGEs. To assess cell viability, cells were stained with 7-AAD and PE-labeled Annexin V (MBL) in binding buffer (MBL), according to the manufacturer's instructions. For proliferation assay, peritoneal cells were stained with surface markers as described above, fixed, and permeabilized using the Foxp3/transcription factor buffer set (eBioscience). After washing, the cells were incubated with PE-labeled anti-Ki-67 antibody (1:200 dilution, 16A8, Biolegend) at 4 °C for 60 min and analyzed. For BrdU incorporation assay, J774A.1 cells were cultured in the presence of BSA or AGEs (100 or 300 µg/ml) for the indicated time periods and labeled with BrdU for 45 min before being harvested. Cells were fixed and stained with APC-labeled anti-BrdU antibody, following the protocol provided by the manufacturer (BD Biosciences, APC-BrdU Flow Kit) and detected by FACSVerse. The data were analyzed by FlowJo v10 software (Tree Star).

**Immunocytochemistry.** J774A.1 cells, cultured on glass-based dishes, were preincubated with anti CD16/CD32 mAb (93, Biolegend) to block the FcγRII/III receptors and incubated with anti-histone H2B mAb (5HH2-2A8, Millipore, 1:200 dilution) and either 100 µg/ml Bt-BSA or Bt-AGEs at 4 °C for 15 min in HBSS-BSA. After washing three times with PBS (−), the cells were stained with streptavidin-Alexa Fluor 568 conjugate (1:1,000 dilution, Invitrogen) and anti-mouse IgG-Alexa Fluor 488 conjugate (1:1,000 dilution, Invitrogen) at 4 °C for 15 min in HBSS-BSA. After washing, the stained cells were fixed with 4% paraformaldehyde in PBS (−) for 10 min at room temperature and further stained with 1 µg/ml 4′,6-Diamidino-2-phenylindole (DAPI, Dojindo). Fluorescent images were obtained using a confocal microscope (FV1200, Olympus) and analyzed by Olympus software (FV10-ASW, Olympus).

**Detection of AGEs-binding histone H2B on plasma membrane.** J774A.1 cells were incubated with either Bt-BSA or Bt-AGEs (400 µg/ml) at 4 °C for 10 min in HBSS-BSA, washed three times with PBS and then incubated with 2 mM 3,3′-dithiobis(sulfosuccinimidyl propionate) (DTSSP, Dojindo Laboratories) at room temperature for 15 min. After three times wash with HBSS-BSA containing 100 mM glycine, cells were lysed by buffer B of UltraRIPA kit as described above. Streptavidin-coupled magnetic beads were added to cell lysates and incubated at 4 °C for 1 h with rotation. After washing three times with PBST, the binding proteins were eluted by adding sample buffer and heating (95 °C, 5 min) and subjected to western blotting to detect histone H2B.

**Western blotting.** The samples were separated by 10% SDS-PAGE and transferred to PVDF membrane (Millipore). The membrane was incubated for 1 h with Blocking One (Nacalai Tesque) to block nonspecific binding. The membrane was then incubated overnight at 4 °C with antibodies against histone H2B (5HH2-2A8, Millipore), histone H1.2 (19649-1-AP, Proteintech), histone H2A (938CT5.1.1, Santa Cruz), histone H3 (1G1, Santa Cruz), histone H4 (L64C1, Cell signaling), RAGE (A-9, Santa Cruz), and MMP-9 (AB19016, Millipore) in 10% Blocking One-PBST, followed by the incubation for 1 h at room temperature with HRP-

conjugated secondary antibodies (1:5,000 dilution, anti-mouse IgG, Cell signaling, #7076; 1:5,000 dilution, anti-rabbit IgG, Cell signaling, #7074). Detection was performed using SuperSignal West Pico Chemiluminescent Substrate (Thermo Fisher Scientific) and LAS4000 (Cytiva). The data were analyzed by ImageJ Fiji software version 1.52n. For the detection of total proteins, PVDF membrane was stained with 0.1% Ponceau S solution containing 5% acetate for 5 min. The stained membrane was scanned with LAS4000, destained with PBST, and then used for subsequent blocking and immunodetection for histone H2B.

**Generation of construct expressing H2B-IRES-GFP and transfection.** For construction of pcDNA4-H2B-IRES-GFP, the gene coding H2B (amplified from pET15b vectors carrying the His-tagged human histone H2B as described above) and IRES-GFP (amplified from IRES-GFP Co-Expression HR Targeting Vector, SBI System Biosciences) were inserted into the pcDNA4/TO/myc-HisA vector (Invitrogen) using NEBuilder HiFi DNA Assembly (New England Biolabs). J774A.1 cells were transfected with either pcDNA4-H2B-IRES-GFP vector or empty pcDNA4 vector (mock) using an ELEPO21 electroporator (Nepa Gene), incubated for 24 h, and subjected to flow cytometry analysis.

**Plg binding assay.** Plg binding assay was performed according to a previous report with minor modifications[12]. Briefly, freshly-isolated peritoneal cells or J774A.1 cells were preincubated with BSA (100 μg/ml), AGEs (100 μg/ml) or TXA (1 mM) in HBSS-BSA buffer for 15 min at 4 °C, then incubated with 200 nM Bt-Plg for 15 min. The cells were washed with HBSS-BSA and Plg binding was quantified by the mean fluorescence intensity (MFI) of the streptavidin-conjugated APC by FACS. Plg binding (% of control) was calculated according to the following equation: $(S - T)/(C - T) \times 100$, in which S was the MFI of BSA or the AGEs-treated cells, T was the MFI of the tranexamic acid-treated cells, and C was the MFI of the non-treated cells.

**Plg activity assay.** The kinetics of the Plg activation were determined by measuring the amidolytic activity of the plasmin generated during activation of Plg as previously described with some modifications[39]. Recombinant histone H2B (10 μg/ml) was mixed with the unmodified BSA or AGEs (100 μg/ml) in 100 μl of HBSS-BSA, followed by incubation for 5 min, then Glu-Plg (50 nM, Millipore) was added. The reaction was initiated by the addition of the urokinase Plg activator (10 nM, R&D Systems) and the chromogenic substrate H-D-Val-Leu-Lys-p-nitroaniline dihydrochloride (S-2251) (500 μM, Sekisui Medical), and the reaction was measured at 25 °C, monitoring the absorbance at 405 nm. The initial rates of the plasmin generation were calculated by a linear regression analysis of the plots of the absorbance at 405 nm versus time[2] utilizing the data points at a low extent of substrate conversion as outlined in[39]. For the plasmin activity assay using cell line J774A.1, cells were seeded onto 96-well plate ($10^5$ cells/well). Next day, culture media were changed to HBSS-HEPES buffer containing 3 mM $CaCl_2$ and 1 mM $MgCl_2$. Cells were treated with the unmodified BSA or AGEs (100 μg/ml), followed by incubation at 4 °C for 15 min, then Glu-Plg (300 nM) was added. After incubation at 4 °C for 1 h, the reaction was initiated by the addition of the urokinase Plg activator (3 nM) and S-2251 (500 μM), and the reaction was measured at 25 °C, monitoring the absorbance at 405 nm.

**In vitro infiltration assay.** The macrophage infiltration activities were assessed using transwell chambers. The upper surface of the transwell (8 μm pore size, Corning Costar) was coated with Matrigel (300 μg/ml, BD Biosciences) at 37 °C for 16 h. J774A.1 cells ($2.5 \times 10^4$ cells) in serum-free DMEM were seeded on the upper chamber, and DMEM containing serum and the macrophage chemoattractant, MCP-1 (10 nM, R&D Systems), was placed into the lower chamber. Plg (200 nM), unmodified BSA (100 μg/ml), AGEs (100 μg/ml), anti-H2B antibody (A-6, 4 μg/ml), and/or anti-RAGE antibody (A-9, 4 μg/ml) were added into the upper chamber, then the cells were incubated for 24 h. After incubation, cells across the Matrigel were fixed with 4% paraformaldehyde, permeabilized with methanol and stained with DAPI. The number of DAPI-stained cells on the transwell membrane was counted using a confocal microscope (FV1200, Olympus) and ImageJ software.

**Determination of MMP-9 activation.** To detect both pro and cleaved MMP-9, gelatinases were concentrated from peritoneal lavage fluid from TG-stimulated mice by binding to gelatin-agarose beads (Sigma). Peritoneal lavages were collected at 72 h after TG injection, centrifuged to remove peritoneal cells, and then the resulting supernatants (0.5 ml from 2.5 ml of the total supernatants) were incubated with 15 μl of 50% gelatin-agarose beads at 4 °C for 1 h with rotation. After washing three times with PBST containing 5 mM $CaCl_2$, the binding proteins were eluted by adding sample buffer and heating (95 °C, 5 min) and subjected to western blotting to detect MMP-9, as described above.

**RT-PCR.** The total RNA was extracted from peritoneal macrophages using the FastGene RNA Basic Kit (NIPPON Genetics) and reverse transcribed using ReverTra Ace (Toyobo) according to the supplier's instructions. The PCR amplification was performed using the THUNDERBIRD SYBR qPCR Mix (Toyobo) and LightCycler 96 system (Roche). The expression levels were normalized using

β-actin as an endogenous control gene. The primers sequences used are shown in Supplementary Table S3.

**RNA-seq analysis.** Mice were daily administered with BSA or AGEs (10 mg/kg/day) starting 24 h before TG injection and continuing until the mice were sacrificed. Peritoneal macrophages were isolated from mice 72 h after intraperitoneal TG injection, and total RNA was extracted as described above. The mRNA library preparation, mRNA sequencing, and gene expression analysis were performed by Macrogen (Korea). Briefly, the removal of ribosomal RNA and library construction of biological duplicate samples were performed with TruSeq Stranded Total RNA with Ribo-Zero H/M/R_Gold (Illumina), and 100 bp paired-end sequencing was done with a NovaSeq6000 instrument (Illumina). The reads were mapped to the reference genome (mm10) using the HISAT2 version 2.1.0. After the read alignment, fragments per kilobase million (FPKM) and transcripts per kilobase million (TPM) were calculated with StringTie version 2.1.3b, and differentially expressed genes (DEG) were determined by DESeq2. Genes that satisfied |fold change|>1.5 and nbinomWaldTest raw $p$-value < 0.05 were taken as significantly differentially expressed and compared with previously published data on gene signatures in M1 and M2 macrophages[40].

**ATP assay.** ATP contents were determined using ATPlite (PerkinElmer) according to manufacturer's instructions. Briefly, J774A.1 cells were seeded in 96-well plate and then treated with either BSA or AGEs at indicated concentrations and time periods. After incubation, cells were added with mammalian cell lysis solution. The cell lysates were transferred into white 96-well plate, mixed with substrate solution, and incubated for 5 min with shaking. The luminescence was measured using a 2030 Multilabel Reader ARVO X3 (PerkinElmer).

**Cecal ligation and puncture (CLP) model.** The CLP procedure was performed as described previously[41]. Briefly, mice were anaesthetized and subjected to midline laparotomy. The cecum was exposed and ligated with 4-0 silk suture on about 50% of the caecum. The cecal wall was punctured with a 21-gauge needle, avoiding puncture the blood vessels, and a small amount of stool was extruded. After that, the cecum is returned to the peritoneal cavity, and the incision was closed layer by layer. At the end of the operation, the pre-warmed saline at 37 °C was subcutaneously injected. Mice were treated with either saline, BSA (50 mg/kg), or AGEs (50 mg/kg) by intravenous injection 4 h after the CLP procedure was performed, and the survival rates of the mice were monitored over time.

**Statistical analyses.** Data are expressed as means ± S.D. Statistical analyses were performed using Prism Software v6.07 (GraphPad), with a $P$ value of <0.05 being considered significant. Differences between two groups were analyzed by either unpaired or paired, two-tailed Student's $t$ test. Multiple comparisons between groups were made using Tukey–Kramer tests or Dunnett's test.

**Reporting summary.** Further information on research design is available in the Nature Research Reporting Summary linked to this article.

## Data availability
The RNA-Seq data have been deposited into the Gene Expression Omnibus under accession code "GSE195558". The structure of histone H2B used in this study is available in the Protein Data Bank (PDB) under accession code "2RVQ". The remaining data are available within the Article, Supplementary Information, or from the corresponding author upon reasonable request. Source data are provided with this paper.

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

## Acknowledgements

This work was supported by JSPS KAKENHI Grant Numbers JP17H06170, JP26111011, JP20K15467, and by AMED under Grant Number JP19gm0910013h0003. This research was partially supported by Platform Project for Supporting Drug Discovery and Life Science Research (Basis for Supporting Innovative Drug Discovery and Life Science Research (BINDS)) from AMED under Grant Number JP19am0101076 (support number 1496).

## Author contributions

M.I., K.Y., R.K., S.-Y.L., J.Y., and T.S. performed the experiments. L.N. performed LC-MS analysis. H.S., and R.N. performed the measurement of AGEs. M.I., Y.A., J.Y., T.S., and K.U. designed and analyzed the experiments. M.I. and K.U. wrote the manuscript and contributed funding acquisition. K.U. supervised and coordinated research.

## Competing interests

The authors declare no competing interests.
