## [Peer Review File · Nature Communications]

Reviewers' comments:

Reviewer #1 (Remarks to the Author):

Glycation is a common non-enzymatic carbohydrate addition to Lys residues of extracellular proteins, typically accelerated by hyperglycemia, and can lead to advanced glycation end products (AGEs). This is an interesting manuscript that addresses the functions of advanced AGEs and presents the novel observation that histone H2A/B and H4 are binding partners for dehydroascorbic acid (DHA)-modified albumin, which is used as a surrogate for proteomic studies in protein-AGEs interactions to bind such binding partners. Based on their analyses, the authors propose that AGEs bind to cell surface histones and mediate cytotoxicity. Thus, the key element of discovery here is proposed to be the unique binding of AGEs to cell-surface histone and the functional consequences for regulating plasminogen (Plg) activation, as Plg also binds histones, resulting in the suppression of monocyte/macrophage invasion. Thus, in essence the mechanism proposed is that Plg binds surface histone, but AGEs inhibit or modulate such interactions. Thus, the findings raise the possibility that AGEs have a role in regulating homeostatic processes and not just act as proinflammatory agents.

1. It is unclear mechanistically how binding to histone at the surface of cells leads to regulating Plg activation. They suggest that cell surface histone is a surface Plg binding protein. Is the binding of AGEs independent of binding to Plg? The evidence for AGE binding blocking Plg binding is for the most part shown in Fig. 4 for example. But in that Fig. 4A, the concentration of AGE used to inhibit is 10 micromolar and above. Do AGEs cause release of histone from the cell surface, thus resulting in decreased Plg binding? Does pretreatment with AGE and then removal also block the binding?
2. There are only modest effects on plasmin formation shown in Fig. 4e in vitro, but no data are provided for effects on intact cells?
3. The mechanism of histone and Plg interactions at the cell surface are not clear in this study. From the references cited by the authors, histones are proposed to tether to surface membranes by electrostatic binding to phosphatidylserine (PS); such binding is salt-dependent, and it was shown that the same agents that block binding of Plg to H2B inhibit binding to PS. But the authors here show that the binding of H2B to AGEs is different as it is not sensitive to 500 mM NaCl and must not occur solely through electrostatic interactions. The authors should comment on this phenomenon of interaction.
4. The DHA AGE product is quite different in structure from glucose or methylglyoxal products, but the authors data indicates that all such AGEs are bound by histones. This is unusual, and suggest that a unique motif of the AGE product shared by all the different glycation products is recognized. Do the authors have an idea about this novelty? While the authors have taken steps to understand this phenomenon, as described in Fig. 2, they can only conclude that "The data support the idea that fluorescent products generated in the AGEs might be involved in the interaction with histones.", but this is not particularly informative in regard to the molecular recognition shared by all the glycation products.
5. A key aspect of the paper is in regard to surface expression of histones on activated macrophages and their binding to AGE, depicted for the most part in Fig. 3. But there is only indirect evidence here that AGE binding is to histones, as the direct interaction to surface histone is not demonstrated, only correlated. Other AGE receptors on the surface of macrophages, such as scavenger receptor-A on

macrophages, are likely to also bind AGE and may be the principal receptors. Direct evidence of binding of AGE to surface histone should be demonstrated.

6. In Fig. 1A, there are many proteins that are pulled down by the AGE, but the authors focused on the histones? Is there a rationale for this? What about the identification of other proteins in that pull-down?

7. Is there direct evidence that Histones 2A/B occur at the surface of cells, and are they localized in lipid rafts, as the data from Fig. 1A suggests?

8. Does the binding of AGEs to known AGE receptors, which are generally universally expressed, influence these pathways? It would appear from the pull-down evidence that histones are more prominent receptors for AGEs than the known family of AGE receptors. Could the authors provide some context as to how over the years such AGE receptors as histones were not identified in the many studies on AGE receptors?

9. From the results in Fig. 5 the authors conclude that "These data indicated that AGEs have a function in regulating monocytes/macrophage recruitment to the inflammatory sites via the inhibition of the Plg activation". However, there is no direct evidence that the mechanism is through inhibition of Plg activation.

10. The rather artificially in vitro produced AGE product with VitC and albumin is very heavily glycosylated, but this degree of glycation is perhaps not physiological. Do lower levels of glycosylated products produce similar effects or does it require dozens of glycosylated residues, as suggested by Suppl. Fig. 1, to produce the observed effects?

Reviewer #2 (Remarks to the Author):

The scientific importance of this manuscript is that the authors identified for the first time histone as a novel AGE binding cellular protein. Then, the authors tried to explain homeostatic role of AGEs in regarding with their binding activity to histone using variety of advanced instrumental and molecular biological assays both in vitro and in vivo. However, following comments should be considered for publication of this work in corresponding peer reviewed journal.

Major comments:

1. Ascorbic acid is also known for their pro-oxidant activity. Accordingly, how did you recognize that these products are whether the AGEs or protein oxidation products. Please specify how did you control the formation of intermediate products of glucose metabolites, such as glycoaldehyde, GA-pyridine, and N-pyrrolylsine in your designated experiment conditions?

2. Please demonstrate at least one result, where produced AGEs were quantitatively measured after incubating BSA with DHA in this experiment.

3. Please prove pull down assay and solid phase binding assays by using AGEs - specific antibodies to demonstrate that the fractions are exactly matched to the AGEs.

4. It seems like that experimental design is systematically not well organized, for example, the authors used mouse peritoneal cells to determine histone dependent binding of AGEs to macrophages, whereas J774A.1 cell line was applied for determination of regulating activity of AGEs against the monocytes or macrophages. And cytotoxic effects of AGEs are tested on human endothelial EA. hy926 cell lines. Such unlinked arrangement is making the results uncomprehensive. Similarly, experiments in first 5 results

have been obtained for histone H2B, while last experiment was conducted using histone H4.

5. It is known that AGEs largely bound to the cell surface through their receptor – RAGE. Please demonstrate a proof that in this experiment, AGEs specifically bound to histone proteins, not with RAGE.

Secondary comments:

1. There are several minor technical errors in this manuscripts, including:

a. Please use same abbreviations for all reagents. For example, glucose was written in different ways in figures and their corresponding below explanation (Glc and Glu) as shown in Figure 2C and 2D.

b. Please include concentrations for each reagents and experiment conditions in Figure 2C. It is curious whether the concentrations of DHA and other glucose metabolites were same and if so please explain why the absorbance of DHA is higher than others.

c. In Figure 1A, please explain what is represented by a, b, and c.

d. Please check spelling errors (i.e., biding affinity in Page 10).

2. Do the concentrations of the fractions which are obtained from approximately 5 to 15 minutes and those found in 15 to 25 minutes by HPLC analysis were comparable? Please make it clear.

3. In materials and methods, please provide statistical analysis information.

4. It would be more precise, if the title of the manuscripts changed as “AGEs mediate homeostatic response by binding to histone H2B”, since most of the results are obtained for this type and each of them may have distinguished role during the cell cycle.

Reviewer #3 (Remarks to the Author):

In this manuscript, Uchida and colleagues report that advanced glycation end products (AGEs) might mediate homeostatic responses by binding to histone proteins. The authors utilize dehydroascorbic acid (DHA)-modified serum albumin as an AGE probe to screen for binding proteins in the lipid raft fraction from mouse splenocytes and identify histone as an AGE-binding protein. The authors then demonstrate the binding capacity of different types of AGEs (such as glucose, methylglyoxal, and glycolaldehyde glycation products) to histones. The authors also identify that AGEs regulate monocytes/macrophage recruitment and inhibit histone toxicity through the formation of aggregates with histone H4. The general concept of this paper is interesting however, the data is very preliminary and I have major concerns regarding this manuscript:

1) In a strictly chemical sense, DHA is not a sugar molecule and the structures of DHA-modified amino acid residues are quite different from AGEs induced by reducing sugars, despite the similar linkage. Thus, there is little comparability between the DHA-modified proteins and classic AGEs, making them a poor mimic (see schematics). To utilize a DHA-modified BSA as a pull-down probe also does not have any physiological relevance due to the low concentration of DHA in human blood serum.

2) The binding mode of AGEs to histone is not clear. Since there are hundreds of different glycation product structures induced by multiple sugar molecules or their metabolites, it is extremely difficult to

imagine a uniform binding mode where the positively charged AGEs could tightly bind histone proteins, which are rich in Lys and Arg. The authors have tested the binding ability of AGEs formed by distinct sugar molecules, such as Glc, MG, GA, GCA, 3-DG, and GO. However, it is meaningless to have this comparison due to their entirely different chemical structures. Finally, the authors do not provide an adequate negative control of highly positively charged protein and/or aggregation-prone protein that does NOT bind these AGEs.

3) The formed AGE structures are highly dependent on the microenvironment of the glycosylated proteins because of the rearrangements that form crosslinks (e.g., glucosepane). Even though the authors have tested two AGE examples with exact chemical structures (GA-pyridine and N ϵ -pyrrolylysine), these two free small molecule mimics do not represent the complexity and specificity of AGEs formed on different proteins. More structural and biochemical insights should be provided in order to understand the interactions between histones and AGEs.

4) The pull-down analysis presented at the basis of the manuscript (Figure 1a) is poorly designed and executed. It relies on a visual identification of Coomassie bands (which selection seems random). Instead, authors should apply a quantitative proteomic analysis that will provide an accurate and comprehensive determination of the AGE binding proteins.

5) The logic and flow of the manuscript is confusing. On one hand, the authors first identify that the cell-surface plasminogen receptor, H2B, is a binding target of DHA-modified BSA and AGEs. On the other hand, they next claim that AGEs could form aggregates with other histones to inhibit histone toxicity.

6) The downstream biological effects of this binding activity are not clearly elucidated. More biochemical, in cellulo and in vivo experiments are required to uncover the function of the interactions between AGEs and histones. For example, the experiments performed in Fig.5 should also be performed with truncated and mutant histones.

Reviewer #4 (Remarks to the Author):

The manuscript by Itakura et al investigates how sugar-protein compounds called glycation end products (AGEs) function and attempts to discover their binding partners in cells. They use DHA modified serum albumin as an example of AGE to screen for cellular binding partners in lipid raft fractions prepared from mouse spleen cells. Ultimately, histones are shown to bind DHA (AGE). More specifically, AGE bound histone component H2B which serves as a cell surface plasminogen receptor. Although, AGE species are known to be pro-inflammatory and are implicated in inflammatory diseases the authors propose that AGE in fact can be immunosuppressive by preventing inflammatory monocyte/macrophage recruitment and activation.

Advanced glycation end products or AGEs form when reduced sugars covalently react w/ proteins. That AGEs are associated with diseases as pathogenic molecules is known, but "how" is a black box. In order to characterize the AGE-associated mechanisms, authors used a 3-step approach:

- 1) Find AGE-binders
- 2) Gain / Loss of function experiments to validate AGE binding
- 3) Evaluate the dynamic efforts of AGE during disease/models

Major concerns:

The biochemical studies related to the discovery of binding partners for DHA modified serum albumin (AGE) are by and large convincing. However, the physiological significance of this particular AGE species as a bona fide immunoregulatory complex is less convincing. This is important because the authors are proposing a relatively new function of AGE species as anti-inflammatory instead of what has been well established that AGE serve as proinflammatory entities.

With respect to AGE binding to monocytes/macrophages can the authors use another myeloid cell type to show the specificity to monocytes? Perhaps myeloid dendritic cell?

AGEs bind to macrophages via histones. In order to show that histones are really required for this binding, can histones be blocked somehow to do a loss of function expt to make data more convincing?

The studies described in Fig 5 and 6 where the authors attempt to demonstrate the physiological significance of AGEs as immunosuppressive molecules is not as convincing. The in vitro invasion assay needs to be described better. The TG treatment shows reduction in F480+ GR1 low macrophages suddenly at 72 hrs after treatment, however, the trend at 48 hrs shows as actual increase in macrophage number in TG treated animals. Similar trend can be observed with monocytes where they are actually increased at 24 followed by a precipitous decline at 48 hrs. Considering the model of peritonitis this looks less like an issue with recruitment but it appears that monocytes and macrophages in AGE treated mice are undergoing cell death. This needs to be clarified. Furthermore, the authors show that in addition to the putative recruitment defect the macrophages exhibit a M2 phenotype. However, the relative gene expression differences are minor at best. The authors need to assess protein expression and also perform assays that show the physiological significance of these proposed intrinsic immunoregulatory nature AGE treated macrophages – perhaps even some in vitro assays may suffice.

In Fig.6 the authors perform a survival experiment where they show that histones+ BSA has a notable effect. This effect is not trivial and needs to be explored further.

The authors at the very least show a protective effect of AGE in a more robust inflammatory model. Perhaps to begin with they should use a sepsis model – LPS induced sepsis, and cecal slurry model.

Point-by-point reply: Itakura et al. NCOMMS-20-36860

Dear Reviewers:

We thank the reviewers for their helpful comments. We carefully studied their comments and made the necessary corrections that we hope meet with their approval. Please find below a detailed point-by-point response to all comments (reviewers' comments in blue, our replies in black).

Response to Reviewer #1:

Glycation is a common non-enzymatic carbohydrate addition to Lys residues of extracellular proteins, typically accelerated by hyperglycemia, and can lead to advanced glycation end products (AGEs). This is an interesting manuscript that addresses the functions of advanced AGEs and presents the novel observation that histone H2A/B and H4 are binding partners for dehydroascorbic acid (DHA)-modified albumin, which is used as a surrogate for proteomic studies in protein-AGEs interactions to bind such binding partners. Based on their analyses, the authors propose that AGEs bind to cell surface histones and mediate cytotoxicity. Thus, the key element of discovery here is proposed to be the unique binding of AGEs to cell-surface histone and the functional consequences for regulating plasminogen (Plg) activation, as Plg also binds histones, resulting in the suppression of monocyte/macrophage invasion. Thus, in essence the mechanism proposed is that Plg binds surface histone, but AGEs inhibit or modulate such interactions. Thus, the findings raise the possibility that AGEs have a role in regulating homeostatic processes and not just act as proinflammatory agents.

1. It is unclear mechanistically how binding to histone at the surface of cells leads to regulating Plg activation. They suggest that cell surface histone is a surface Plg binding protein. Is the binding of AGEs independent of binding to Plg? The evidence for AGE binding blocking Plg binding is for the most part shown in Fig. 4 for example. But in that Fig. 4A, the concentration of AGE used to inhibit is 10 micromolar and above. Do AGEs cause release of histone from the cell surface, thus resulting in decreased Plg binding? Does pretreatment with AGE and then removal also block the binding?

Response: In our experiment using recombinant histone proteins (**Figures 1b, c, and d**), AGEs can bind to histone proteins in the absence of Plg, indicating that Plg is dispensable for the binding of AGEs to histones. This interpretation is supported by our new data showing that Plg specifically binds to histone H2B, but not to AGEs (**Supplementary Figure 6**). To clarify in more detail the relationship among H2B, AGEs and Plg, we assessed the inhibitory effect of AGEs on Plg binding to H2B by surface plasmon resonance (**new Figure 4b**). The result revealed that the pretreatment with AGEs was sufficient to block the Plg binding to H2B, even after washing with a buffer. Similarly, the binding of Plg to macrophages was significantly suppressed by the pretreatment with AGEs (**Supplementary Figure 9**). Furthermore, we added new data showing that the level of cell surface histone H2B was not significantly decreased by the treatment with AGEs (**Supplementary Figure 10a**) and that extracellular histone H2B was not detected even in the case of AGEs treatment

(**Supplementary Figure 10b**). These new supporting data indicated that the inhibition of Plg binding to macrophage by AGEs was not due to H2B release from the cell surface.

2. There are only modest effects on plasmin formation shown in Fig. 4e *in vitro*, but no data are provided for effects on intact cells?

Response: We thought the reviewer's comment made sense. Therefore, we carried out the plasmin activity assay using cell line J774A.1 (**new Figure 4g**). Consistent with the result obtained from *in vitro* experiment using recombinant H2B (**Figure 4f**), plasmin formation was promoted in the presence of cells and suppressed by the treatment with AGEs. Although these effects appear small when focused on the plasmin formation promoted by H2B (or alternatively by the addition of cells), AGEs showed almost 60% inhibitory activity. This is reasonable and consistent with the results of the plasminogen binding assay (**Figure 4a and new Figure 4b**).

3. The mechanism of histone and Plg interactions at the cell surface are not clear in this study. From the references cited by the authors, histones are proposed to tether to surface membranes by electrostatic binding to phosphatidylserine (PS); such binding is salt-dependent, and it was shown that the same agents that block binding of Plg to H2B inhibit binding to PS. But the authors here show that the binding of H2B to AGEs is different as it is not sensitive to 500 mM NaCl and must not occur solely through electrostatic interactions. The authors should comment on this phenomenon of interaction.

Response: As the reviewer mentioned, histone H2B interacts with phosphatidylserine by electrostatic binding to phosphatidylserine. It is also known that histone H2B has a carboxy-terminal lysine, which interacts with the kringle domain of plasminogen (Herren *et al.*, *Biochemistry* 45, 9463–9474 (2006)). With regards to the binding mode of H2B and AGEs, we speculated that electrostatic interaction might be involved in the interaction of AGEs with histone, based on the facts that (i) histone is a highly basic protein and (ii) AGEs are electronegative molecules (Chikazawa *et al.*, *Biochemistry* 55, 435–446 (2016)). To validate this speculation, we tested whether histone H2B could recognize electronegative molecules and confirmed that acylated proteins (acetylated BSA, succinylated BSA, and maleylated BSA) indeed interacted with H2B (**new Figure 2e**). In addition, the possible contribution of electrostatic interaction was also demonstrated by the observations that the binding of the AGEs to H2B was inhibited by 1 M NaCl (**new Figure 2f**) and by the deletion of a highly basic N-terminal region (Δ N35) from H2B (**new Figure 2g**). Furthermore, the AGEs were recognized by the scrambled fragments of the H2B N-terminal (scrambled 1, 2, and 3) comprising identical amino acids in different amino acid sequences (**new Figure 2g**). These data support our hypothesis that the electronegative potential of AGEs might be involved, at least in part, in the recognition by histone. These results are also analogous to a previous report showing that histone could form a tight complex with nuclear protein including acidic domain, such as heterochromatin protein 1, even in the presence of 500 mM NaCl, and this interaction could be dissociated with 1 M NaCl (Hara *et al.*, *EMBO reports*, 2, 920-925 (2001)). The high affinities of histones for electronegative molecules could be brought by the combined electrostatic interactions between the N-terminal arginines plus lysines of histones and electronegative molecules in the same manner as reported for the

binding of histone tail with DNA (Hong *et al.*, *J Biol Chem*, 268, 305–314 (1993)). Meanwhile, we observed that lysozyme, a basic protein having an isoelectric point similar to histone proteins, did not show recognition specificity to AGEs (**Supplementary Figure 5**). This led us to speculate that, in addition to the electrostatic potential, some specific structures of H2B may contribute to the histone binding to the AGEs. We explained these reasons in the revised manuscript (page 11, line 17-page 12 line 9).

4. The DHA AGE product is quite different in structure from glucose or methylglyoxal products, but the authors data indicates that all such AGEs are bound by histones. This is unusual, and suggest that a unique motif of the AGE product shared by all the different glycation products is recognized. Do the authors have an idea about this novelty? While the authors have taken steps to understand this phenomenon, as described in Fig. 2, they can only conclude that “The data support the idea that fluorescent products generated in the AGEs might be involved in the interaction with histones.”, but this is not particularly informative in regard to the molecular recognition shared by all the glycation products.

Response: It has been reported that DHA and its degradation products could form adduct species, such as CML, pentosidine, GA-pyridine, and LM-1, structurally similar to those obtained upon incubation of proteins with other reducing sugars (Dunn *et al.*, *Biochemistry*, 29, 10964–10970 (1990); Nagaraj *et al.*, *Proc. National Acad. Sci.*, 88, 10257–10261 (1991); Argirov *et al.*, *Biochim. Biophys. Acta - Gen. Subj.*, 1620, 235–244 (2003); Tessier *et al.*, *J. Biol. Chem.*, 274, 20796–20804 (1999).). In addition, as far as we know, no DHA-specific AGEs have been identified so far. Therefore, we believe that DHA gives AGEs similar to glucose, although there may be structural differences in the details.

Because lysine residues in proteins are major targets of glycation, we speculate that common feature of AGEs caused by all the different glycation products may be electronegative potential, not a unique motif or chemical structure of the AGE product shared by all the different glycation products. Consistent with this view, this manuscript shows that DHA-derived AGEs and other sugars-derived AGEs generally bind to histones (**Figures 2c and d**). Our speculation is also supported by the literature regarding the recognition mechanism of AGEs receptor, showing that one of the major common characteristics of known RAGE ligands is the net negative charge at physiological pH (Fritz *et al.* *Trends Biochem. Sci.*, 36, 625–32 (2011)). The new sentences concerning electrostatic interaction were added in the revised manuscript (page 11, line 17-page 12 line 9).

As for our conclusion that “The data support the idea that fluorescent products generated in the AGEs might be involved in the interaction with histones.”, we agree with the reviewer’s comment that this is not particularly informative in regard to the molecular recognition shared by all the glycation products. Therefore, the related sentences were deleted from the manuscript.

5. A key aspect of the paper is in regard to surface expression of histones on activated macrophages and their binding to AGE, depicted for the most part in Fig. 3. But there is only indirect evidence here that AGE binding is to histones, as the direct interaction to surface histone is not demonstrated, only correlated. Other AGE receptors on the surface of macrophages, such as scavenger receptor-A on macrophages, are likely to also bind AGE and

may be the principal receptors. Direct evidence of binding of AGE to surface histone should be demonstrated.

Response: To directly demonstrate the binding of AGEs to histone H2B on the cell membrane, we conducted the experiment using the membrane-impermeable cross-linker (DTSSP). Cells were treated with biotin-labeled (Bt-) BSA or Bt-AGEs and interacting proteins were cross-linked by DTSSP. Cells were then lysed and subjected to pull-down followed by Western blotting. As shown in **new Figure 3f**, significant signals were observed when treated with Bt-AGEs. This indicates that AGEs are directly bound to H2B localized in the membrane. We also investigated the involvement of other AGEs receptors on macrophages using anti-histone or anti-AGEs receptor antibodies. As shown in **new Figure 3g**, the treatment with anti-H2B, H3, or H4 antibodies significantly inhibited AGEs binding to macrophages to about the same extent as anti-AGEs receptors (RAGE, AGER1, or CD36) antibodies. These new data demonstrate that AGEs bind to macrophages through interactions with histones, in addition to the known family of AGE receptors.

6. In Fig. 1A, there are many proteins that are pulled down by the AGE, but the authors focused on the histones? Is there a rationale for this? What about the identification of other proteins in that pull-down?

Response: Based on repeated experiments, the protein bands uniquely detected by the AGEs pull-down were histone proteins (bands a-c). Others were BSA or AGEs that were used for pull-down. That's why we focused on histone proteins.

7. Is there direct evidence that Histones 2A/B occur at the surface of cells, and are they localized in lipid rafts, as the data from Fig. 1A suggests?

Response: To directly show the presence of histones on the cell surface, a cell-surface biotinylation assay was conducted. Cells were treated with sulfo-NHS-biotin, a membrane-impermeable labeling reagent, and cell surface proteins were labeled with biotin. Biotinylated membrane proteins in RIPA soluble and lipid raft fractions were affinity purified and subjected to Western blotting. As a result, it was shown that some histone proteins are present on the cell surface (**new Supplementary Figure 2**). Unlike RAGE, which is completely localized in the RIPA-soluble fraction, histones were detected in both the RIPA-soluble and the lipid raft fractions (**new Supplementary Figure 2**).

8. Does the binding of AGEs to known AGE receptors, which are generally universally expressed, influence these pathways? It would appear from the pull-down evidence that histones are more prominent receptors for AGEs than the known family of AGE receptors. Could the authors provide some context as to how over the years such AGE receptors as histones were not identified in the many studies on AGE receptors?

Response: As mentioned earlier in the response to the reviewer's comment #5, the new data (**new Figure 3g**) show that AGEs bind to macrophages through the interaction with histones, in addition to the known family of AGE receptors. Among them, RAGE is particularly known for its involvement in inflammatory responses. Therefore, we investigated whether the inhibitory effect of AGEs on macrophage infiltration could be affected by RAGE. Our new result from in vitro invasion assay (**new Figure 5b**) showed that anti-H2B antibody,

but not anti-RAGE antibody, could suppress the AGEs-dependent inhibition of macrophage infiltration. Thus, histones are more prominent receptors for AGEs than the known family of AGE receptors, RAGE.

There is a general concept that histones are nuclear proteins, and the presence of histone proteins in lipid raft fractions has been attributed to contamination from the nuclear fraction (Poston *et al.*, *Biochem. Biophys. Res. Commun.*, 415:355–360 (2011)). This may be the reason why histones have not been identified as AGE receptors until our current study. Thus, the role of histones as AGEs receptors have been overlooked.

9. From the results in Fig. 5 the authors conclude that “These data indicated that AGEs have a function in regulating monocytes/macrophage recruitment to the inflammatory sites via the inhibition of the Plg activation”. However, there is no direct evidence that the mechanism is through inhibition of Plg activation.

Response: For the direct evidence that the mechanism is related to the inhibition of Plg activation, we performed a new *in vivo* experiment using a Plg activation inhibitor, tranexamic acid. As shown in **new Figure 5d**, the inhibitory effect of AGEs on macrophage infiltration was completely canceled by treatment with tranexamic acid. We also analyzed downstream factors of Plg activation *in vivo*. Plg is known to play a role in the activation of matrix metalloproteinases (MMPs), which is essential for monocytes/macrophages motility and invasion during the inflammatory response (Gong *et al.*, *J. Clin. Invest.* 118, 3012–3024 (2008).). Hence, we investigated whether treatment with AGEs could attenuate the activation of MMP-9 in the peritonitis model. Western blotting with peritoneal lavage fluid from thioglycollate-stimulated mice showed that treatment with AGEs significantly suppressed MMP-9 activation compared to treatment with unmodified BSA (**new Figure 5e**). These results support our conclusion that AGEs have the ability to regulate monocytes/macrophage recruitment to inflammatory sites through inhibition of Plg activation.

10. The rather artificially *in vitro* produced AGE product with VitC and albumin is very heavily glycosylated, but this degree of glycation is perhaps not physiological. Do lower levels of glycosylated products produce similar effects or does it require dozens of glycosylated residues, as suggested by Suppl. Fig. 1, to produce the observed effects?

Response: As the reviewer pointed out, DHA-AGEs prepared *in vitro* may be more glycosylated than physiological AGEs. However, the same is true for *in vitro*-prepared glucose-derived AGEs. To assess the validity of the amounts of AGEs used in this study, we measured the concentrations of histone-bound AGEs in sera from wild-type mice and diabetic *ob/ob* mice. Elevated serum levels of AGEs have previously been reported in *ob/ob* mice (Kim *et al.*, *Exp. Dermatol.*, 27, 815–823 (2018).). Quantitative analysis using a competitive solid-phase binding assay revealed that serum levels of histone-binding proteins were significantly elevated in *ob/ob* mice compared to WT mice (**new Supplementary Figure 4a**). The concentrations of AGEs equivalent to DHA-BSA in WT mice and *ob/ob* mice were 191 $\mu\text{g/ml}$ and 318 $\mu\text{g/ml}$, respectively (**new Supplementary Figure 4b**). Therefore, the amount of AGEs used in this study is considered to be within the range that occurs under physiological or pathophysiological conditions.

Response to Reviewer #2:

The scientific importance of this manuscript is that the authors identified for the first time histone as a novel AGE binding cellular protein. Then, the authors tried to explain homeostatic role of AGEs in regarding with their binding activity to histone using variety of advanced instrumental and molecular biological assays both in vitro and in vivo. However, following comments should be considered for publication of this work in corresponding peer reviewed journal.

Major comments:

1. Ascorbic acid is also known for their pro-oxidant activity. Accordingly, how did you recognize that these products are whether the AGEs or protein oxidation products. Please specify how did you control the formation of intermediate products of glucose metabolites, such as glycoaldehyde, GA-pyridine, and N-pyrrolylsine in your designated experiment conditions?

Response: As the reviewer pointed out, ascorbate shows a pro-oxidant activity, so do other reducing sugars such as glucose. Therefore, as long as reducing sugars are used as a source of AGEs, the formation of protein oxidation products is unavoidable. This means that AGEs in general are a mixture of both glycation and oxidation products. It is almost impossible to control the formation of intermediate products, including GA-pyridine and N-pyrrolylsine, and we did not do in our study. Data using these small products (**Figure 2e, f**) have been deleted from the manuscript because we were unable to detect them in the AGEs used in this study.

2. Please demonstrate at least one result, where produced AGEs were quantitatively measured after incubating BSA with DHA in this experiment.

Response: Following the reviewer's advice, we quantitatively measured N^ε-carboxymethyllysine (CML), a well-known AGE, generated in the DHA-modified BSA. The concentration of CML was significantly increased in DHA-modified BSA (AGEs, 0.64±0.30 mmol/mol Lys), compared to unmodified BSA (0.04±0.02 mmol/mol Lys). The results were included in the revised manuscript (**Supplementary Figure 1c**).

3. Please prove pull down assay and solid phase binding assays by using AGEs - specific antibodies to demonstrate that the fractions are exactly matched to the AGEs.

Response: To demonstrate the presence of DHA-derived AGEs in the fractions with histone binding activity, we attempted to detect the AGEs using an anti-AGEs monoclonal antibody established in our laboratory (Chikazawa *et al.*, *J. Biol. Chem.*, 288, 13204–13214 (2013)). **Figure 2b** shows that H2B-binding fractions were recognized by the antibody. The data support our finding that AGEs bind to histone H2B.

4. It seems like that experimental design is systematically not well organized, for example, the authors used mouse peritoneal cells to determine histone dependent binding of AGEs to

macrophages, whereas J774A.1 cell line was applied for determination of regulating activity of AGEs against the monocytes or macrophages. And cytotoxic effects of AGEs are tested on human endothelial EA. hy926 cell lines. Such unlinked arrangement is making the results uncomprehensive. Similarly, experiments in first 5 results have been obtained for histone H2B, while last experiment was conducted using histone H4.

Response: As the reviewer pointed out, this paper consisted of two main parts with multiple cells used and was therefore confusing. In this study, to clarify AGEs-binding cells, we first used mouse peritoneal cells and identified macrophages as target cells (**Figures 3a and b**). Based on this result, we evaluated AGEs binding to macrophage/monocyte cell lines (J774A.1 and RAW264.7) to examine the universality of AGEs binding to macrophages (**Figures 3c and d**). These two cell lines, similar to mouse primary macrophages, have been reported to express membrane-bound H2B, which contributes to plasminogen binding (Das *et al*, *Blood*, 110, 3763–72 (2007)). However, in our preliminary experiments, RAW264.7 showed marked non-specific plasminogen binding (which could not be blocked by the inhibitor of plasminogen receptors, tranexamic acid) compared to J774A.1 (data not shown). Therefore, we used J774A.1 cells for subsequent experiments on the binding of AGEs (**Figures 3e, f, g, and h**), the inhibition of plasminogen binding (**Figure 4d**), and the inhibition of plasminogen activation (**Figure 4g**), except that the inhibition of plasminogen binding to macrophages by AGEs was also confirmed in peritoneal macrophages (**Figure 4e**).

Human endothelial EA. hy926 cell lines were used for the effect of AGEs on histone cytotoxicity because endothelial cells are known to be the major target of histone proteins, especially histone H4 (Xu *et al.*, *Nat. Med.*, 15, 1318–1321 (2009)). However, to avoid unnecessary confusion, the last part regarding histone cytotoxicity has been removed from the manuscript.

5. It is known that AGEs largely bound to the cell surface through their receptor – RAGE. Please demonstrate a proof that in this experiment, AGEs specifically bound to histone proteins, not with RAGE.

Response: To answer the reviewer's remarks, we investigated the involvement of other AGE receptors on macrophages using anti-histone or AGE receptor antibodies. As shown in **new Figure 3g**, treatment with anti-H2B, H3, or H4 antibodies significantly inhibited the binding of AGEs to macrophages, similar to treatment with antibodies to the known family of AGE receptors (RAGE, AGER1, or CD36). Given previous findings regarding the binding of AGEs to these receptors, the results are very reasonable. However, the new data also establish that the binding of AGEs is not specific to these AGEs receptor, but they can also bind histones.

Response to Reviewer #3

In this manuscript, Uchida and colleagues report that advanced glycation end products (AGEs) might mediate homeostatic responses by binding to histone proteins. The authors utilize dehydroascorbic acid (DHA)-modified serum albumin as an AGE probe to screen for binding proteins in the lipid raft fraction from mouse splenocytes and identify histone as an

AGE-binding protein. The authors then demonstrate the binding capacity of different types of AGEs (such as glucose, methylglyoxal, and glycolaldehyde glycation products) to histones. The authors also identify that AGEs regulate monocytes/macrophage recruitment and inhibit histone toxicity through the formation of aggregates with histone H4. The general concept of this paper is interesting however, the data is very preliminary and I have major concerns regarding this manuscript:

1) In a strictly chemical sense, DHA is not a sugar molecule and the structures of DHA-modified amino acid residues are quite different from AGEs induced by reducing sugars, despite the similar linkage. Thus, there is little comparability between the DHA-modified proteins and classic AGEs, making them a poor mimic (see schematics). To utilize a DHA-modified BSA as a pull-down probe also does not have any physiological relevance due to the low concentration of DHA in human blood serum.

Response: Several previous studies reported that DHA and its degradation products form the same adduct species, such as CML, pentosidine, GA-pyridine, and LM-1, that are formed when proteins are incubated with other reducing sugars (Dunn *et al.*, *Biochemistry*, 29, 10964–10970 (1990); Nagaraj *et al.*, *Proc. National Acad. Sci.*, 88, 10257–10261 (1991); Argirov *et al.*, *Biochim. Biophys. Acta - Gen. Subj.*, 1620, 235–244 (2003); Tessier *et al.*, *J. Biol. Chem.*, 274, 20796–20804 (1999)). Other studies have also shown that DHA and its degradation products are causally involved in the formation of AGEs *in vivo* (Nagaraj *et al.*, *Proc. National Acad. Sci.*, 88, 10257–10261 (1991)). In fact, following the advice of other reviewer (reviewer #2), we measured *N*^ε-carboxymethyllysine (CML), a well-known AGE, generated in the DHA-modified BSA and observed that the concentration of CML was significantly increased in DHA-modified BSA (AGEs, 0.64±0.30 mmol/mol Lys), compared to unmodified BSA (0.04±0.02 mmol/mol Lys) (**Supplementary Figure 1c**). Thus, it is evident that there is comparability between the DHA-derived AGEs and classic glucose-derived AGEs. Consistent with this view, no DHA-specific AGEs have been identified so far. In addition, this manuscript shows that sugars-derived AGEs, including glucose-derived AGEs, also bind to histones as well as DHA-derived AGEs (**Figures 2c and d**).

On the other hand, DHA is present in the blood at about 10 - 20 μM in blood (Sato *et al.*, *Biological Pharm. Bulletin*, 33, 364–369 (2010)). In addition, as mentioned above, DHA and its degradation products are causally related to the formation of AGEs *in vivo*. Therefore, we believe that the use of a DHA-modified BSA as a pull-down probe have physiological relevance.

2) The binding mode of AGEs to histone is not clear. Since there are hundreds of different glycation product structures induced by multiple sugar molecules or their metabolites, it is extremely difficult to imagine a uniform binding mode where the positively charged AGEs could tightly bind histone proteins, which are rich in Lys and Arg. The authors have tested the binding ability of AGEs formed by distinct sugar molecules, such as Glc, MG, GA, GCA, 3-DG, and GO. However, it is meaningless to have this comparison due to their entirely different chemical structures. Finally, the authors do not provide an adequate negative control

of highly positively charged protein and/or aggregation-prone protein that does NOT bind these AGEs.

Response: As the reviewer pointed out, AGEs are a mixture of numerous different types of glycation product structures induced by multiple sugar molecules or their metabolites. Therefore, it is reasonable to imagine that a uniform binding mode may not be involved in the binding of AGEs to histone. That is why we speculate that electrostatic interaction may be involved in the interaction of AGEs with positively charged histone proteins. To validate this speculation, we tested whether histone H2B could recognize electronegative molecules and confirmed that acylated proteins (acetylated BSA, succinylated BSA, and maleylated BSA) indeed interacted with H2B (**new Figure 2e**). Furthermore, the possible contribution of electrostatic interaction was also demonstrated by the observations that the binding of the AGEs to H2B was inhibited by 1 M NaCl (**new Figure 2f**) and by the deletion of a highly basic N-terminal region (Δ N35) from H2B (**new Figure 2g**). In addition, AGEs were recognized by the scrambled fragments of the H2B N-terminal (scrambled 1, 2, and 3) comprising identical amino acids in different amino acid sequences (**new Figure 2g**). These data support our hypothesis that the electronegative potential of AGEs might be involved, at least in part, in the recognition by histone. These results are also analogous to a previous report showing that histone could form a tight complex with nuclear protein including acidic domain, such as heterochromatin protein 1, even in the presence of 500 mM NaCl, and this interaction could be dissociated with 1 M NaCl (Hara *et al.*, *EMBO reports*, 2, 920-925 (2001)). The high affinities of histones for electronegative molecules could be brought by the combined electrostatic interactions between the N-terminal arginines plus lysines of histones and electronegative molecules in the same manner as reported for the binding of histone tail with DNA (Hong *et al.*, *J Biol Chem*, 268, 305–314 (1993)). Moreover, we observed that lysozyme, a basic protein having an isoelectric point similar to histone proteins, did not show any recognition specificity to AGEs (**Supplementary Figure 5**). This observation can be a negative control of highly positively charged protein requested by the reviewer. These are the reasons for speculating that, in addition to the electrostatic potential, some specific structures of H2B may contribute to the histone binding to the AGEs. These have been described in the revised manuscript (page 11, line 17-page 12 line 9).

3) The formed AGE structures are highly dependent on the microenvironment of the glycosylated proteins because of the rearrangements that form crosslinks (e.g., glucosepane). Even though the authors have tested two AGE examples with exact chemical structures (GA-pyridine and N ϵ -pyrrolylysine), these two free small molecule mimics do not represent the complexity and specificity of AGEs formed on different proteins. More structural and biochemical insights should be provided in order to understand the interactions between histones and AGEs.

Response: We agree with the reviewer's comment that the modified small molecules, such as GA-pyridine and N ϵ -pyrrolylysine, do not represent the complexity and specificity of AGEs. In addition, we were unable to detect these small molecules in the AGEs used in this study. Therefore, data using these small products have been deleted from the manuscript. On the other hand, as stated above in response to the reviewer's comment #2, we speculated that electrostatic interaction can be involved in the binding of AGEs with positively charged

histone proteins and revealed that the electronegative potential of AGEs might be involved, at least in part, in the recognition by histone.

4) The pull-down analysis presented at the basis of the manuscript (Figure 1a) is poorly designed and executed. It relies on a visual identification of Coomassie bands (which selection seems random). Instead, authors should apply a quantitative proteomic analysis that will provide an accurate and comprehensive determination of the AGE binding proteins.

Response: We understand that our pull-down experiment is a classic “comparative” approach for identifying target binding proteins: comparison of native protein with AGEs. Using the same approach, however, we have successfully identified several binding proteins, including C1q as a serum AGEs-binding protein (Chikazawa et al., *Biochemistry* 55, 435-446, 2016) and apolipoprotein E as a pyrrole-binding protein (Hirose et al., *J. Biol. Chem.* 294, 11035-11045, 2019). It is not a state-of-the-art method, but it still works. In addition, as commented by the reviewer, it basically relies on a visual identification of CBB- or silver-stained bands. However, at least in this experiment, the selection was not random, as most of the proteins present in the AGEs-pulldown rather than the BSA-pull-down were histone proteins. Others are AGEs that were used for pull-down. Quantitative proteomic analysis may be the best approach, but it is not readily available in our laboratory.

5) The logic and flow of the manuscript is confusing. On one hand, the authors first identify that the cell-surface plasminogen receptor, H2B, is a binding target of DHA-modified BSA and AGEs. On the other hand, they next claim that AGEs could form aggregates with other histones to inhibit histone toxicity.

Response: This has also been pointed out by other reviewers. As the reviewer mentioned, the structure of this paper was confusing due to the existence of two different streams. Therefore, we decided to remove the data on cytoprotective effect of AGEs via binding to histone H4 from the manuscript and decided to focus on AGEs-dependent regulatory effect on monocytes and macrophages via binding to histone H2B in the revised manuscript.

6) The downstream biological effects of this binding activity are not clearly elucidated. More biochemical, in cellulo and in vivo experiments are required to uncover the function of the interactions between AGEs and histones. For example, the experiments performed in Fig.5 should also be performed with truncated and mutant histones.

Response: As a downstream biological effect of the interaction between AGEs and histone H2B, monocytes/macrophages cell line J774A.1 was used to assess plasminogen activation (**new Figure 4g**). Consistent with the result obtained from experiments with recombinant H2B (**Figure 4d**), plasmin formation was promoted by treatment of cells with plasminogen, which was suppressed by the treatment with AGEs. In addition, to investigate whether the inhibitory effect of AGEs on macrophage infiltration was due to a similar mechanism, we conducted a new *in vivo* experiment using a plasminogen activation inhibitor, tranexamic acid. As shown in **new Figure 5d**, the inhibitory effect of AGEs on macrophage infiltration was completely canceled by treatment with tranexamic acid. We also added a new *in vivo* analysis for downstream factors of plasminogen activation. Plasminogen is known to

play a role in the activation of matrix metalloproteinases (MMPs), and activation of MMP-9 is essential for monocytes/macrophages motility and invasion during the inflammatory response (Gong *et al.*, *J. Clin. Invest.* 118, 3012–3024 (2008)). Hence, we investigated whether treatment with AGEs could reduce the activation of MMP-9 in a model of peritonitis. Western blotting with peritoneal lavage fluid in thioglycollate-stimulated mice showed that AGEs significantly suppressed MMP-9 activation compared to untreated and unmodified BSA (**new Figure 5e**). Following the reviewer's suggestion, we further conducted the infiltration assay using mutant H2B-expressing cells. Unfortunately, however, the expression of truncated histone induced marked cell death, and therefore it was not possible to assess its impact on the infiltration.

Reviewer #4 (Remarks to the Author):

The manuscript by Itakura et al investigates how sugar-protein compounds called glycation end products (AGEs) function and attempts to discover their binding partners in cells. They use DHA modified serum albumin as an example of AGE to screen for cellular binding partners in lipid raft fractions prepared from mouse spleen cells. Ultimately, histones are shown to bind DHA (AGE). More specifically, AGE bound histone component H2B which serves as a cell surface plasminogen receptor. Although, AGE species are known to be pro-inflammatory and are implicated in inflammatory diseases the authors propose that AGE in fact can be immunosuppressive by preventing inflammatory monocyte/macrophage recruitment and activation.

Advanced glycation end products or AGEs form when reduced sugars covalently react w/ proteins. That AGEs are associated with diseases as pathogenic molecules is known, but “how” is a black box. In order to characterize the AGE-associated mechanisms, authors used a 3-step approach:

- 1) Find AGE-binders
- 2) Gain / Loss of function experiments to validate AGE binding
- 3) Evaluate the dynamic efforts of AGE during disease/models

Major concerns:

The biochemical studies related to the discovery of binding partners for DHA modified serum albumin (AGE) are by and large convincing. However, the physiological significance of this particular AGE species as a bona fide immunoregulatory complex is less convincing. This is important because the authors are proposing a relatively new function of AGE species as anti-inflammatory instead of what has been well established that AGE serve as proinflammatory entities.

Response: We agree with the reviewer's comment that the physiological significance of this particular AGE species as a bona fide immunoregulatory complex was not very convincing. We therefore conducted further experiments to establish our proposal. Especially,

as a downstream biological effect of the interaction between AGEs and histone H2B, monocytes/macrophages cell line J774A.1 was used to assess plasminogen activation (**new Figure 4g**). Consistent with the result obtained from experiments with recombinant H2B (**Figure 4d**), plasmin formation was promoted by treatment of cells with plasminogen, which was suppressed by the treatment with AGEs. In addition, to investigate whether the inhibitory effect of AGEs on macrophage infiltration was due to a similar mechanism, we conducted a new *in vivo* experiment using a plasminogen activation inhibitor, tranexamic acid. As shown in **new Figure 5d**, the inhibitory effect of AGEs on macrophage infiltration was completely canceled by treatment with tranexamic acid. This result indicates that AGEs regulate monocytes/macrophage recruitment via the inhibition of the Plg activation, but not via its cytotoxicity. This notion is also supported by our new data (**Supplementary Figure 11**), showing that AGEs do not have any cytotoxic effect on macrophages even in the concentration range up to 300 µg/ml for 72 h. We also added a new *in vivo* data on downstream factors of plasminogen activation. Western blotting with peritoneal lavage fluid in thioglycollate-stimulated mice showed that AGEs significantly suppressed MMP-9 activation compared to untreated and unmodified BSA (**new Figure 5e**). These new data further strengthen our findings that AGEs can serve as anti-inflammatory entities.

With respect to AGE binding to monocytes/macrophages can the authors use another myeloid cell type to show the specificity to monocytes? Perhaps myeloid dendritic cell?

Response: In accordance with the reviewer's comment, we tested other cell-types (dendritic cells, neutrophils, and eosinophils) in addition to monocytes and macrophages and observed that AGEs bind primarily to macrophages and monocytes (**New Figure 3b**).

AGEs bind to macrophages via histones. In order to show that histones are really required for this binding, can histones be blocked somehow to do a loss of function expt to make data more convincing?

Response: To answer the reviewer's comment, we conducted a new experiment using anti-histone and anti-AGE receptor antibodies. As shown in **new Figure 3g**, treatment with anti-H2B, H3, or H4 antibody significantly inhibited AGEs binding to macrophages to about the same extent as anti-AGE receptors (RAGE, AGER1, or CD36) antibodies. These data indicate that AGEs bind to macrophages via histones, in addition to the well-known AGE receptors.

The studies described in Fig 5 and 6 where the authors attempt to demonstrate the physiological significance of AGEs as immunosuppressive molecules is not as convincing. The *in vitro* invasion assay needs to be described better. The TG treatment shows reduction in F480+ GR1 low macrophages suddenly at 72 hrs after treatment, however, the trend at 48 hrs shows as actual increase in macrophage number in TG treated animals. Similar trend can be observed with monocytes where they are actually increased at 24 followed by a precipitous decline at 48 hrs. Considering the model of peritonitis this looks less like an issue with recruitment but it appears that monocytes and macrophages in AGE treated mice are undergoing cell death. This needs to be clarified.

Response: As mentioned above in response to the reviewer's major concern, we conducted further experiments to establish our proposal and added some new data to the revised manuscript. We believe that these new data further strengthen our findings that AGEs can serve as anti-inflammatory molecules. As for a description on the *in vitro* invasion assay, the related text was rewritten with reference to the reviewer's comments (page 8, lines 22-30). To clarify the mechanism underlying the inhibitory effect of AGEs on monocytes/macrophage recruitment into peritoneal cavity in more detail, we carried out a new *in vivo* experiment using a plasminogen activation inhibitor, tranexamic acid. As shown in **new Figure 5d**, the inhibitory effect of AGEs on macrophage recruitment was completely canceled by the treatment with tranexamic acid. This result indicates that AGEs regulate monocytes/macrophage recruitment via the inhibition of the Plg activation, but not via its cytotoxicity. This notion is supported by our new data (**Supplementary Figure 11**) showing that AGEs do not have any cytotoxic effect on macrophages even in the concentration range up to 300 µg/ml for 72 h.

Furthermore, the authors show that in addition to the putative recruitment defect the macrophages exhibit a M2 phenotype. However, the relative gene expression differences are minor at best. The authors need to assess protein expression and also perform assays that show the physiological significance of these proposed intrinsic immunoregulatory nature AGE treated macrophages – perhaps even some *in vitro* assays may suffice.

Response: Based on our findings that AGEs bind to histone H2B on the surface of macrophage and regulate plasminogen activation, we speculated that predominant action of AGEs is the suppression of proinflammatory cell infiltration through binding to H2B. This means that the changes in gene expression shown in **Figure 5e** are due to a decrease in infiltrated inflammatory cells, not a promotion of macrophage polarization. This may be the reason why the differences in the gene expression were relatively small. In accordance with the reviewer's suggestion, we examined the expression of protein markers for M1 (CD80) and M2 macrophages (CD206) by flow cytometry. However, the differences were negligible, and significant increase of M2 macrophage was not observed (**new Supplementary Figure 12**).

In Fig.6 the authors perform a survival experiment where they show that histones+ BSA has a notable effect. This effect is not trivial and needs to be explored further.

The authors at the very least show a protective effect of AGE in a more robust inflammatory model. Perhaps to begin with they should use a sepsis model – LPS induced sepsis, and cecal slurry model.

Response: Based on the reviewer's suggestion, we carried out new experiments using cecal ligation and puncture (CLP)-induced sepsis model for a more pathophysiological approach. As a result, both prolonged survival and decreased levels of inflammatory cytokines in the blood were observed in mice treated with AGEs. However, other reviewers pointed out that the structure of this paper was confusing due to the existence of two different streams. Therefore, we decided to remove the data on cytoprotective effect of AGEs via binding to histone H4 from the manuscript.

The comments offered by the four reviewers have been very helpful in formulating what we believe is a stronger paper. We appreciate these thoughtful comments and hope that our responses and in particular our revisions have allowed this paper to achieve a priority sufficient for publication in Nature Communications.

Koji Uchida, Ph.D.
Professor
Graduate School of Agricultural and Life Sciences,
The University of Tokyo, Tokyo 113-8657, Japan
Tel: 81-3-5841-5127
Fax: 81-3-5841-8026
E-mail: a-uchida@mail.ecc.u-tokyo.ac.jp

REVIEWER COMMENTS

Reviewer #1 (Remarks to the Author):

Here the authors present evidence that histone H2A/B and H4 are binding partners for dehydroascorbic acid (DHA)-modified albumin. The presence of histones on cell surfaces has been observed by others, but the key finding of this paper is that such displayed histones can interact with AGEs. The histone interaction involves electrostatic mechanisms, as high salt (1-3M) disrupts such interactions, but the authors demonstrate than simply being a basic protein, such as lysozyme, does not confer binding to the AGEs.

The authors observed that AGEs bound only to macrophages and monocytes but not dendritic cells and neutrophils for example. While the evidence indicates that histones can bind AGEs it is not clear as to what approximate percentage of binding of AGEs to cells is attributable to histone interactions. According to Fig. 3g, binding is slightly, albeit statistically significantly, inhibited by antibodies to histones, implying that histone binding is contributing to only a small fraction of AGE binding. An intriguing experiment would be add histones to cells, including monocytes and even dendritic cells and neutrophils, to observe whether that would enhance surface histone expression and confer binding of AGEs to the cells.

Other studies have reported that plasminogen binding to cell surfaces may be to histones. Here the authors examined that AGEs inhibited plasminogen binding to histones, implying the binding sites for both are similar. Interestingly, the authors show that AGEs (Fig. 4) can inhibit binding of plasminogen to macrophages. The consequences of such data implies that AGEs can functionally interfere with macrophage movements and involvements in inflammation, also consistent with evidence of inhibition of MMP9 activation by AGEs, independently of RAGE recognition. However, it is a bit unclear as to whether plasminogen binding can displace histones on their surface through their binding to the surface through unknown mechanisms.

An obvious question is whether DNA might inhibit histone binding to AGEs, but this was not directly address.

Altogether, this study provides strong evidence that cell surface histones in macrophages can function as a receptor for AGEs, with functional consequence on interactions with plasminogen and other signaling pathways previously associated with only AGE interactions with RAGEs. An unsolved riddle at present is the mechanism of histone localization in lipid rafts or membrane microdomains. This reviewer would propose the possibility that histones may interact through charge interactions with highly sulfated glycosaminoglycans, e.g. heparan sulfate, and thereby associate with the membrane. In any case, the potential anti-inflammatory nature of histone/AGE interactions proposed here is intriguing and potentially of biological significance.

Reviewer #3 (Remarks to the Author):

In this resubmission, the authors made efforts to add additional data and reorganize the figures. There are more details in this new version in comparison to the original one. However, the overall

content of the manuscript and findings did not change. The authors still fail to address the major concerns raised by the reviewers, including more mechanistic insights and physiological relevance.

I still stick to the original point of view that this paper is not suitable for publication in Nat. Commun. It could be transferred to another journal which requires less mechanistic insights (see attached).

Reviewer #5 (Remarks to the Author):

In this present manuscript, authors identified for the first time that histones can play as a novel AGE binding cellular protein, and its manifestation in monocyte/macrophage recruitment. Authors have reasonably justified majority of comments from the first reviewer and others. The author gave the explanation that as they are using reducing sugars it is unavoidable to formation of these products and they have deleted data (which is causing the concern) of these products from the study because they were not able to detect them in the AGEs used in the study.

I think overall work is novel and thought provoking, the manuscript structure is well organized and well written and should be consider for the publication.

Reviewer #6 (Remarks to the Author):

Itakura et al provide an improved version of the manuscript adding novel pieces of data to support their conclusions but some major concerns remain. Firstly, the new evidence supporting a direct binding of AGE to histones using assays with blocking antibodies is useful, although the effects shown are mild, raising the question whether there is significant redundancy between the different types of histones bound by AGE. This limitation should be at least discussed in the manuscript. In addition, there are two major limitations in the current version of the manuscript that should be addressed:

Major concerns:

1. The evidence that AGE directly inhibits migration of monocytes/macrophages to the peritoneum is rather weak and other possible explanations such as cell death or in situ proliferation differences are not properly addressed. This should be done at least with simple flow cytometric analysis of apoptotic (using ie Annexin V staining) and proliferation markers (ie Ki-67, even better using BrdU/EdU incorporation experiments). Excluding these two options would greatly strengthen the migration hypothesis. This is particularly important as the in vitro viability assay seems to be problematic with some values above 100%, which is impossible. I would recommend repeating these experiments using a viability dye and flow cytometry.

2. There is no evidence provided on where the inhibition of AGE on macrophage/monocytes might play a role in a disease or more complex in vivo setting. As previously suggested a sepsis model would be a useful model here to evaluate the relevance of the findings of the study. Indeed, assessing the functional impact of AGE binding to histones would greatly increase the value of the study. Rather than the superficial analysis of the macrophage phenotype currently provided, doing RNAseq of

sorted peritoneal macrophages with and without AGE treatment would be very informative and help better understand the biological impact of AGE binding.

Minor concerns:

- In figure 3a, the dot plots for BSA vs AGE condition should be shown, not just the histograms
- In Figure 3b histograms for each cell type including gating strategy should be shown, not just an MFI quantification
- In figure 3h dot plots should be shown in addition

**** See Nature Research's author and referees' website at www.nature.com/authors for information about policies, services and author benefits.**

Dear Reviewers:

We thank the reviewers for their constructive comments and suggestions on the manuscript. They were very helpful to provide further evidence to support our claim and to emphasize the physiological and pathophysiological relevance of our findings. We carefully revised the manuscript according to the instructions by the editor and the reviewers. We sincerely hope that our responses and revisions allow this paper to achieve a priority sufficient for publication in Nature Communications.

Below is a detailed point-by-point response to all comments by the reviewers (reviewer's comment in blue and our response in black).

Response to Reviewer #1:

Here the authors present evidence that histone H2A/B and H4 are binding partners for dehydroascorbic acid (DHA)-modified albumin. The presence of histones on cell surfaces has been observed by others, but the key finding of this paper is that such displayed histones can interact with AGEs. The histone interaction involves electrostatic mechanisms, as high salt (1-3M) disrupts such interactions, but the authors demonstrate than simply being a basic protein, such as lysozyme, does not confer binding to the AGEs.

The authors observed that AGEs bound only to macrophages and monocytes but not dendritic cells and neutrophils for example. While the evidence indicates that histones can bind AGEs it is not clear as to what approximate percentage of binding of AGEs to cells is attributable to histone interactions. According to Fig. 3g, binding is slightly, albeit statistically significantly, inhibited by antibodies to histones, implying that histone binding is contributing to only a small fraction of AGE binding. An intriguing experiment would be add histones to cells, including monocytes and even dendritic cells and neutrophils, to observe whether that would enhance surface histone expression and confer binding of AGEs to the cells.

Response: We agree with the reviewer's comment on the effects of antibodies on the binding of AGEs to cells (**Fig. 3g**). Following the reviewer's suggestion, we carried out an additional experiment using recombinant histone H2B to demonstrate the involvement of H2B in the binding of AGEs to cells. New data (**new Supplementary Figure 10**) show that the treatment of peritoneal cells with recombinant H2B significantly enhanced the association of H2B to the cell surface of macrophages, monocytes, and dendritic cells, resulting in remarkable increase in the binding of AGEs. Therefore, the addition of an extracellular histone to cells further enhanced surface expression of histones and conferred binding of AGEs to the cells. The data support our view that the binding of AGEs to cells is attributed to H2B. We appreciate the reviewer for suggesting this experiment. These new findings are included in the Result section (**page 7, lines 23-29**) and the Discussion section (**page 13, lines 4-12**) of the revised manuscript.

Other studies have reported that plasminogen binding to cell surfaces may be to histones. Here the authors examined that AGEs inhibited plasminogen binding to histones, implying the binding sites for both are similar. Interestingly, the authors show that AGEs (Fig. 4) can inhibit binding of plasminogen to macrophages. The consequences of such data implies that

AGEs can functionally interfere with macrophage movements and involvements in inflammation, also consistent with evidence of inhibition of MMP9 activation by AGEs, independently of RAGE recognition. However, it is a bit unclear as to whether plasminogen binding can displace histones on their surface through their binding to the surface through unknown mechanisms.

Response: To answer the concerns of the reviewer, we examined changes in the cell surface and extracellular levels of histone H2B before and after treatment with plasminogen (Plg) and observed that there was no significant effect of Plg treatment for H2B levels (Refer to

Supporting data for review on the right). Data from several studies indicate that the interaction of Plg with cell-surface Plg receptors, including histone H2B, accelerates conversion of Plg to plasmin (Plm) (Longstaff *et al.*, *Blood*, 93, 3839-3846 (1999)) and enhances the catalytic activity of Plm (Gonzalez-Gronow *et al.*, *Arch. Biochem. Biophys.*, 286, 625-628 (1991)). In addition, Plm formed on the cell surface is retained on the cell membrane and protected from inactivation by its inhibitor (Plow *et al.*, *J. Cell Biology*, 103, 2411-20 (1986)). Considering these previous reports and our findings (Supporting data for review), the regulation of macrophage dynamics by AGEs is thought to result from the inhibition of Plg activity on cell surface. These statements related to plasminogen activation on cell surface are now included in the Result section (**page 7, line 32-page 8, line 2**) and the graphical summary of this study was added in **new Supplementary Figure 20**.

An obvious question is whether DNA might inhibit histone binding to AGEs, but this was not directly address.

Response: To clarify the effect of DNA on histone binding to AGEs, we carried out a competitive solid phase binding assay and found that DNA inhibits histone binding to AGEs. This finding is included in the Result section (**page 6, lines 2-3**) as **new Supplementary Figure 5** in the revised manuscript.

Altogether, this study provides strong evidence that cell surface histones in macrophages can function as a receptor for AGEs, with functional consequence on interactions with plasminogen and other signaling pathways previously associated with only AGE interactions with RAGEs. An unsolved riddle at present is the mechanism of histone localization in lipid rafts or membrane microdomains. This reviewer would propose the possibility that histones may interact through charge interactions with highly sulfated glycosaminoglycans, e.g. heparan sulfate, and thereby associate with the membrane.

Supporting data for review

Plg treatment does not cause H2B release from cell surface. J774A.1 cells were labeled with membrane impermeable EZ-Link Sulfo-NHS-LC-Biotin for 30 min on ice and then treated with either 200 nM or 1000 nM Plg. After incubation for 15 min, cells were centrifuged, and supernatants were collected as “Extracellular fraction”. The cell lysates were subjected to pull-down with streptavidin-coupled magnetic beads, and the resulting precipitates were collected as “Membrane fraction”. Proteins in Membrane fraction (a) and Extracellular fraction (b) were transferred to PVDF membrane, stained with Ponceau S to visualize total proteins, and subjected to western blotting for H2B.

Response: Previous reports have indicated that histone proteins can bind not only to phosphatidylserine (PS), as referenced in the original manuscript, but also to negatively charged cell surface molecules such as heparan sulfate proteoglycans (Watson *et al.*, *J. Biol. Chem.*, 274, 21707–21713 (1999)). In addition, heparan sulfate-carrying core proteins, such as syndecan-4, have been shown to localize in lipid raft compartment (Tkachenko *et al.*, *J. Biol. Chem.*, 277, 19946–19951 (2002)), and their expression levels vary among cell types; activated macrophages, monocytes, and dendritic cells display high expression of syndecan-4, while neutrophils and eosinophils express low levels of syndecan-4 (Parish *et al.*, *Nat. Rev. Immunol.*, 6, 633–643 (2006); Averbek *et al. Exp. Dermatol.*, 16, 580–589 (2007)). These previous studies suggest that, as proposed by the reviewer, histones bound to the highly sulfated glycosaminoglycans may also be involved in the binding to AGEs. Therefore, new statements related to the mechanism of histone localization have been added to the revised manuscript (**page 13, lines 4-12**).

In any case, the potential anti-inflammatory nature of histone/AGE interactions proposed here is intriguing and potentially of biological significance.

Response: We are very pleased to receive favorable reviews from the reviewer.

Response to Reviewer #3:

In this resubmission, the authors made efforts to add additional data and reorganize the figures. There are more details in this new version in comparison to the original one. However, the overall content of the manuscript and findings did not change. The authors still fail to address the major concerns raised by the reviewers, including more mechanistic insights and physiological relevance.

I still stick to the original point of view that this paper is not suitable for publication in *Nat. Commun.* It could be transferred to another journal which requires less mechanistic insights (see attached).

Response: In response to our first submission, the reviewer raised multiple concerns regarding mechanistic insights and physiological relevance. The issues included (i) comparability between the DHA-derived AGEs and classic AGEs, (ii) binding mode of AGEs to histone, (iii) structure of AGEs involved in the binding to histone, (iv) methodology (pull-down assay), (v) structure of manuscript, and (vi) downstream biological effects of AGEs-histone binding. We thought that our experiments and explanations in the response to the reviewer were sufficient to answer all these concerns and to show mechanistic insights and physiological relevance of our study. However, the reviewer commented that we should have addressed more mechanistic insights and physiological relevance. Other reviewers, who are very positive about this paper, have also suggested doing some experiments related to these issues.

Based on the reviewers' comments and suggestions, we conducted the additional new experiments in this submission as follows: (i) To gain insight into the involvement of histones in the binding of AGEs to the cells, we investigated the effects of the addition of recombinant histone H2B and observed that the treatment of peritoneal cells with recombinant H2B significantly enhanced the association of H2B to the cell surface of macrophages, monocytes, and dendritic cells, resulting in remarkable increase in the binding of AGEs (**new Supplementary Figure 10**). Therefore, the addition of an extracellular histone to cells further enhanced surface expression of histones and conferred binding of AGEs to the cells. The data support our view that the binding of AGEs to cells is

attributed, at least in part, to H2B. These new findings are included in the Result section (**page 7, lines 23-29**) and the Discussion section (**page 13, lines 4-12**) of the revised manuscript. (ii) Because histone is a highly basic protein, we speculated that electrostatic interactions might be involved in the recognition of AGEs by histone. Hence, we investigated the effect of DNA on histone binding to AGEs using a competitive solid phase binding assay and observed that histone binding to AGEs was significantly inhibited by DNA (**new Supplementary Figure 5**). (iii) To provide further evidence that AGEs directly inhibit the migration of monocytes/macrophages to the peritoneum, we examined the effects of AGEs on cell death and *in situ* proliferation of peritoneal macrophages in *in vivo* peritonitis model and revealed that the treatment with AGEs had no significant effect on cell death and proliferation (**new Supplementary Figures 16 and 17**). The new data exclude the possibility of AGEs to be involved in cell death and cell proliferation, which supports our migration hypothesis. These results are included in the Result section of the revised manuscript (**page 9, lines 8-10 and page 9, lines 28-30**). (iv) To better understand the biological impact of AGE binding, we performed RNA-Seq analysis of macrophages isolated from the BSA- or AGEs-treated mice. The results showed that several pro-inflammatory M1-associated genes, such as IL-1b, Ccr7, Igtp, and Sell, were significantly downregulated in macrophages from the mice treated with AGEs compared to those treated with BSA (**new Supplementary Figure 18**). Conversely, AGEs-treatment increased the expressions of anti-inflammatory M2-associated genes, including GATA2, Clec10a, Snn, Pcdh7, Stxbp6, and Gar1 (**new Supplementary Figure 18**). Differential expression analysis data further strengthened our conclusion that AGEs could suppress the infiltration of proinflammatory monocytes/macrophages. These findings are included in the Result section (**page 10, lines 8-11**). The RNA-Seq data have been deposited into the Gene Expression Omnibus (accession code: GSE195558, secure token: upexsueupjylhwf). (v) We conducted *in vivo* experiments using cecal ligation and puncture (CLP)-induced sepsis model and observed that mice treated with AGEs rather than unmodified BSA displayed a significantly prolonged survival time (**new Figure 5g**). The data suggest that AGEs have a function of regulating monocytes/macrophage recruitment to the site of inflammation through inhibition of Plg activation, which may contribute to protection against inflammatory disorders. These *in vivo* data are included in the Result section (**page 10, lines 13-16**) and the Discussion section (**page 13, lines 25-30**) of the revised manuscript. In addition, regarding physiological relevance, the sentences on the possible involvement of some specific structures of H2B in the histone binding to the AGEs, which was addressed in the previous submission, have been retained in the current version of the manuscript (**page 12, lines 26-31**). We believe that these new findings have made this paper much more compelling and solid.

Despite these new data, we also admit that direct evidence of the contribution of the interaction between AGEs and histone H2B in physiological and pathophysiological settings is still limited. This limitation may be due, at least in part, to a lack of methodology for specifically inhibiting the interaction between AGEs and H2B. In this regard, we attempted to use antibodies to block this interaction; however, an anti-H2B antibody showed a significant but only modest decrease in AGEs binding (**Fig. 3g**). We speculate that this may be due to redundancy and functional compensation between the different types of histones in binding of AGEs. In addition, as an alternative approach, we also attempted to inhibit the interaction between AGEs and H2B using both gene silencing of extrachromosomal H2B and dominant negative mutants of H2B (data not shown). However, neither method worked for unknown reasons. Further study is under way in our laboratory to elucidate specific regulatory mechanisms of AGEs-H2B interactions and to establish

their physiological relevance. According to the editor's suggestion, these explanation and limitation of this study have been included in the Discussion section of the revised manuscript (**page 13, line 31-page 14, line 8**).

We sincerely hope that the reviewer finds the revised paper suitable for publication in Nature Communications.

Response to Reviewer #5:

In this present manuscript, authors identified for the first time that histones can play as a novel AGE binding cellular protein, and its manifestation in monocyte/macrophage recruitment. Authors have reasonably justified majority of comments from the first reviewer and others. The author gave the explanation that as they are using reducing sugars it is unavoidable to formation of these products and they have deleted data (which is causing the concern) of these products from the study because they were not able to detect them in the AGEs used in the study.

I think overall work is novel and thought provoking, the manuscript structure is well organized and well written and should be consider for the publication.

Response: We are very pleased to receive favorable reviews form the reviewer.

Response to Reviewer #6:

Itakura et al provide an improved version of the manuscript adding novel pieces of data to support their conclusions but some major concerns remain. Firstly, the new evidence supporting a direct binding of AGE to histones using assays with blocking antibodies is useful, although the effects shown are mild, raising the question whether there is significant redundancy between the different types of histones bound by AGE. This limitation should be at least discussed in the manuscript.

Response: As the reviewer pointed out, the blocking antibody for H2B showed a significant but only modest decrease in AGEs binding. Given that all types of histones are localized on the cell membrane of macrophages (**Fig. S2**) and have capacities for binding to AGEs (**Figs. 1b, c, and d**), this is presumed to be due to redundancy and functional compensation between the different types of histones in binding of AGEs, as suggested by reviewer. Therefore, we added new sentences to the revised manuscript to describe this methodological limitation (**page 13, line 31-page 14, line 3**).

To demonstrate the direct binding of AGEs to H2B on the cell surface in another way, we carried out the experiments using recombinant H2B based on the suggestion by other reviewer (reviewer #1). The treatment of peritoneal cells with recombinant H2B enhanced the association of H2B to the cell surface of macrophages, monocytes, and dendritic cells, which led to the remarkable increase in the binding of AGEs (**new Supplementary Figure 10**). These new data support our claim that the binding of AGEs to cells is attributed, at least in part, to H2B.

In addition, there are two major limitations in the current version of the manuscript that should be addressed:

Major concerns:

1. The evidence that AGE directly inhibits migration of monocytes/macrophages to the peritoneum is rather weak and other possible explanations such as cell death or in situ proliferation differences are not properly addressed. This should be done at least with

simple flow cytometric analysis of apoptotic (using ie Annexin V staining) and proliferation markers (ie Ki-67, even better using BrdU/EdU incorporation experiments). Excluding these two options would greatly strengthen the migration hypothesis. This is particularly important as the *in vitro* viability assay seems to be problematic with some values above 100%, which is impossible. I would recommend repeating these experiments using a viability dye and flow cytometry.

Response: Following the reviewer's suggestion, to provide further evidence that AGEs directly inhibit the migration of monocytes/macrophages to the peritoneum, we examined the effects of AGEs on cell death and *in situ* proliferation of peritoneal macrophages in *in vivo* peritonitis model. Flow cytometric analysis using Annexin V and anti-Ki-67 antibody revealed that the treatment with AGEs had no significant effect on cell death and proliferation (**new Supplementary Figure 17**). These findings were also confirmed by both *in vitro* Annexin V staining and *in vitro* BrdU incorporation assay (**new Supplementary Figures 16a and b**). These new supplemental data exclude the possibility of AGEs to be involved in cell death and cell proliferation, which supports our migration hypothesis. These results are included in the Result section of the revised manuscript (**page 9, lines 8-10 and page 9, lines 28-30**).

On the other hand, in the original manuscript, cell viability was determined by measuring ATP contents and expressed as relative values to control. The values reflect the combined effects of cell viability and proliferative activity, which might be the reason why some values exceeded 100%. Therefore, in addition to the data on cell viability (Annexin V assay) and cell proliferation (BrdU incorporation assay), this data has been revised to be displayed as "ATP contents" (**Supplementary Figure 16c**).

2. There is no evidence provided on where the inhibition of AGE on macrophage/monocytes might play a role in a disease or more complex *in vivo* setting. As previously suggested a sepsis model would be a useful model here to evaluate the relevance of the findings of the study. Indeed, assessing the functional impact of AGE binding to histones would greatly increase the value of the study.

Response: Based on the reviewer's suggestion, new data have been added for *in vivo* experiment using cecal ligation and puncture (CLP)-induced sepsis model to clarify the role of AGEs in more pathophysiological settings (**new Figure 5g**). The data showed that mice treated with AGEs rather than unmodified BSA displayed a significantly prolonged survival time, providing evidence that AGEs might play a role in the context of disease. This result is included in the Result section (**page 10, lines 13-16**) and the Discussion section (**page 13, lines 25-27**) of the revised manuscript.

Rather than the superficial analysis of the macrophage phenotype currently provided, doing RNAseq of sorted peritoneal macrophages with and without AGE treatment would be very informative and help better understand the biological impact of AGE binding.

Response: Following the reviewer's suggestion, RNA-Seq analysis of isolated macrophages from the BSA- or AGEs-treated mice was performed for more comprehensive information. Hierarchical clustering analysis revealed that peritoneal macrophages derived from BSA- and AGEs-treated mice displayed different gene expression profiles (**New Supplementary Figure 18a**). By comparing our data with previously published data on gene signatures in M1 and M2 macrophages (Orecchioni *et al.*, *Front. Immunol.*, 10, 1084 (2019)), we found that several M1-associated genes, such as IL-1b, Ccr7, Igtp, and Sell, were significantly downregulated in macrophages from the mice treated with AGEs compared to those treated with BSA (**New Supplementary Figure 18b**). Conversely,

AGEs-treatment increased the expressions of M2-associated genes, including GATA2, Clec10a, Snn, Pcdh7, Stxbp6, and Gar1 (**New Supplementary Figure 18b**). These results further strengthened our conclusion that AGEs could suppress the infiltration of pro-inflammatory monocytes/macrophages. Quantitative PCR (qPCR) analysis revealed that treatment with AGEs affected the gene expression of pro-inflammatory M1 macrophage markers (IL-1 β and IL-6) and anti-inflammatory M2 macrophage markers (Arg-1, IL-10, and CD206) in infiltrated macrophages (**Fig. 5f**), whereas no significant differences in their expression except for IL-1 β between treatments were found in the RNA-Seq analysis. This discrepancy between qPCR and RNA-Seq analysis may be due to differences in methodologies between the two technologies in quantifying transcript abundances. Given that qPCR generally has higher sensitivity than RNA-Seq (Everaert et al. *Sci. Rep.*, 7, 1559 (2017)), the relatively small differences (~0.6-fold decrease and ~1.5-fold increase) in gene expression analysis by qPCR shown in **Fig. 5f** might not be detectable by RNA-Seq analysis. The results of these findings are described in detail in the Result section (**page 10, lines 8-11**). The RNA-Seq data have been deposited into the Gene Expression Omnibus (accession code: GSE195558, secure token: upexsueupjylhwf).

Minor concerns:

- In figure 3a, the dot plots for BSA vs AGE condition should be shown, not just the histograms

Response: Following the reviewer's comment, we have added the dot plots of Figure 3a to the **new Supplementary Figure 7**.

- In Figure 3b histograms for each cell type including gating strategy should be shown, not just an MFI quantification

Response: Following the reviewer's comment, we have added the gating strategy and histograms of Figure 3b to the **new Supplementary Figure 8**.

- In figure 3h dot plots should be shown in addition

Response: We have added contour plots of **Figure 3h** instead of dot plots to the **new Supplementary Figure 9**. This is because the numbers of cells in GFP(-) and GFP(+) were quite different and we need to display these data as the relative frequency of the populations, regardless of the number of events collected. We are very grateful to the reviewer for pointing this out.

REVIEWERS' COMMENTS

Reviewer #1 (Remarks to the Author):

The authors have responded robustly, in the opinion of this reviewer, to the many suggestions and criticisms raised in the prior reviews. The new results here, of which there are many, further support the discovery and interpretation of results that the authors report here a novel discovery that histones, and especially cell surface histones, play a role as novel receptors for AGE. These interactions may be particularly important to the recruitment of monocyte/macrophage recruitment. This reviewer has no more specific suggestions for improvement beyond those already mentioned.

A key limitation is the physiological significance of this discovery, specifically in regard to cell surface histone involvement in binding AGE. The data largely relies on in vitro models, but overall the results are well presented and the conclusions are strongly supported by these data. Thus, while the concept presented in this manuscript is provocative, it is very interesting and could spur the field to consider the relationship of histones as AGE receptors in comparison to the many other AGE receptors known. But to this reviewer, the big picture of how such histone-specific AGE interactions function in the context of the many other AGE receptors, at least in a physiological setting, is completely lacking. So overall, one has to weigh the content of the paper in terms of its discovery and its potential implications, and on this score the paper is sound and credible.

Reviewer #6 (Remarks to the Author):

The authors have thoroughly addressed all concerns in their revised manuscript, providing both new evidence strengthening their original conclusions as well as improving several minor aspects of the article. The article is in my view fit for publication in its current form

** See Nature Portfolio's author and referees' website at www.nature.com/authors for information about policies, services and author benefits

Dear Reviewers:

We thank the reviewers for their constructive comments on the manuscript. Following the suggestion by Reviewer #1, we have added text to the revised manuscript on how histone-specific AGE interactions function in the context of the many other AGE receptors. We sincerely hope that our responses and revisions allow this paper to achieve a priority sufficient for publication in Nature Communications.

Below is a detailed point-by-point response to all comments by the reviewers (reviewer's comment in blue and our response in black).

Response to Reviewer #1:

The authors have responded robustly, in the opinion of this reviewer, to the many suggestions and criticisms raised in the prior reviews. The new results here, of which there are many, further support the discovery and interpretation of results that the authors report here a novel discovery that histones, and especially cell surface histones, play a role as novel receptors for AGE. These interactions may be particularly important to the recruitment of monocyte/macrophage recruitment. This reviewer has no more specific suggestions for improvement beyond those already mentioned. A key limitation is the physiological significance of this discovery, specifically in regard to cell surface histone involvement in binding AGE. The data largely relies on in vitro models, but overall the results are well presented and the conclusions are strongly supported by these data. Thus, while the concept presented in this manuscript is provocative, it is very interesting and could spur the field to consider the relationship of histones as AGE receptors in comparison to the many other AGE receptors known.

Response: We are very pleased to receive favorable reviews from the reviewer. We agree with the reviewer's comment on the limitation of this discovery, specifically with respect to the involvement of cell surface histone in the binding AGEs. This is partially due to a lack of methodology for specifically inhibiting the interaction between AGEs and histone as addressed in the last submission of the revised manuscript (**page 13, last paragraph**). With that in mind, further study is under way in our laboratory to elucidate specific regulatory mechanisms of AGEs-histone interactions and to establish their physiological relevance.

But to this reviewer, the big picture of how such histone-specific AGE interactions function in the context of the many other AGE receptors, at least in a physiological setting, is completely lacking.

Response: The reviewer's comment reminded us that we should have addressed how histone-specific AGE interactions function in the context of the many other AGE receptors. In this regard, binding of AGEs to the many other AGE receptors has been shown to trigger downstream signaling pathways and mediate the cellular activation or proliferation leading to inflammation and tissue destruction. On the other hand, the discovery of histone as a cell-surface receptor for AGEs in this study suggests that AGEs may also be involved in the homeostatic response via binding to histone. Therefore, there should be some functional interaction between histone and other AGEs receptors. To answer the reviewer's concern, the following statement has been added to the Discussion section (**page 14, lines 6-18**).

“Another concern is the relationship between cell-surface histones and other AGEs receptors. In this regard, due to the high affinity between AGEs and histones, AGEs are

expected to bind immediately to histones on the cell surface in addition to other AGEs receptors. Therefore, it can be expected that cell-surface histones may compete with other AGEs receptor for binding to AGEs, resulting in the downregulation of AGEs-induced inflammatory responses via other AGEs receptors. On the other hand, there may be functional interactions between histones and AGEs receptors. An attractive hypothesis is that there may be an intracellular crosstalk of downstream signaling mediated by histone and other AGEs receptors. However, the presence of such intracellular signaling mechanisms induced by AGEs remains unexplored. Further study is under way in our laboratory to elucidate specific regulatory mechanisms of AGEs-histone interactions and to investigate functional interactions between histone and other AGE receptors.”

So overall, one has to weigh the content of the paper in terms of its discovery and its potential implications, and on this score the paper is sound and credible.

Response: We sincerely thank the reviewer for a positive evaluation.

Response to Reviewer #6:

The authors have thoroughly addressed all concerns in their revised manuscript, providing both new evidence strengthening their original conclusions as well as improving several minor aspects of the article. The article is in my view fit for publication in its current form.

Response: We sincerely thank the reviewer for a positive evaluation.

Below is a detailed point-by-point response to all comments by the reviewers (reviewer's comment in blue and our response in black).

Response to Reviewer #1:

Glycation is a common non-enzymatic carbohydrate addition to Lys residues of extracellular proteins, typically accelerated by hyperglycemia, and can lead to advanced glycation end products (AGEs). This is an interesting manuscript that addresses the functions of advanced AGEs and presents the novel observation that histone H2A/B and H4 are binding partners for dehydroascorbic acid (DHA)-modified albumin, which is used as a surrogate for proteomic studies in protein-AGEs interactions to bind such binding partners. Based on their analyses, the authors propose that AGEs bind to cell surface histones and mediate cytotoxicity. Thus, the key element of discovery here is proposed to be the unique binding of AGEs to cell-surface histone and the functional consequences for regulating plasminogen (Plg) activation, as Plg also binds histones, resulting in the suppression of monocyte/macrophage invasion. Thus, in essence the mechanism proposed is that Plg binds surface histone, but AGEs inhibit or modulate such interactions. Thus, the findings raise the possibility that AGEs have a role in regulating homeostatic processes and not just act as proinflammatory agents.

1. It is unclear mechanistically how binding to histone at the surface of cells leads to regulating Plg activation. They suggest that cell surface histone is a surface Plg binding protein. Is the binding of AGEs independent of binding to Plg? The evidence for AGE binding blocking Plg binding is for the most part shown in Fig. 4 for example. But in that Fig. 4A, the concentration of AGE used to inhibit is 10 micromolar and above. Do AGEs cause release of histone from the cell surface, thus resulting in decreased Plg binding? Does pretreatment with AGE and then removal also block the binding?

Response: In our experiment using recombinant histone proteins (**Figures 1b, c, and d**), AGEs can bind to histone proteins in the absence of Plg, indicating that Plg is dispensable for the binding of AGEs to histones. This interpretation is supported by our new data showing that Plg specifically binds to histone H2B, but not to AGEs (**Supplementary Figure 6**). To clarify in more detail the relationship among H2B, AGEs and Plg, we assessed the inhibitory effect of AGEs on Plg binding to H2B by surface plasmon resonance (**new Figure 4b**). The result revealed that the pretreatment with AGEs was sufficient to block the Plg binding to H2B, even after washing with a buffer. Similarly, the binding of Plg to macrophages was significantly suppressed by the pretreatment with AGEs (**Supplementary Figure 9**). Furthermore, we added new data showing that the level of cell surface histone H2B was not significantly decreased by the treatment with AGEs (**Supplementary Figure 10a**) and that extracellular histone H2B was not detected even in the case of AGEs treatment (**Supplementary Figure 10b**). These new supporting data indicated that the inhibition of Plg binding to macrophage by AGEs was not due to H2B release from the cell surface.

2. There are only modest effects on plasmin formation shown in Fig. 4e *in vitro*, but no data are provided for effects on intact cells?

Response: We thought the reviewer's comment made sense. Therefore, we carried out the plasmin activity assay using cell line J774A.1 (**new Figure 4g**). Consistent with the result obtained from *in vitro* experiment using recombinant H2B (**Figure 4f**), plasmin formation was promoted in the presence of cells and suppressed by the treatment with

AGEs. Although these effects appear small when focused on the plasmin formation promoted by H2B (or alternatively by the addition of cells), AGEs showed almost 60% inhibitory activity. This is reasonable and consistent with the results of the plasminogen binding assay (**Figure 4a and new Figure 4b**).

3. The mechanism of histone and Plg interactions at the cell surface are not clear in this study. From the references cited by the authors, histones are proposed to tether to surface membranes by electrostatic binding to phosphatidylserine (PS); such binding is salt-dependent, and it was shown that the same agents that block binding of Plg to H2B inhibit binding to PS. But the authors here show that the binding of H2B to AGEs is different as it is not sensitive to 500 mM NaCl and must not occur solely through electrostatic interactions. The authors should comment on this phenomenon of interaction.

Response: As the reviewer mentioned, histone H2B interacts with phosphatidylserine by electrostatic binding to phosphatidylserine. It is also known that histone H2B has a carboxy-terminal lysine, which interacts with the kringle domain of plasminogen (Herren *et al.*, *Biochemistry* 45, 9463–9474 (2006)). With regards to the binding mode of H2B and AGEs, we speculated that electrostatic interaction might be involved in the interaction of AGEs with histone, based on the facts that (i) histone is a highly basic protein and (ii) AGEs are electronegative molecules (Chikazawa *et al.*, *Biochemistry* 55, 435–446 (2016)). To validate this speculation, we tested whether histone H2B could recognize electronegative molecules and confirmed that acylated proteins (acetylated BSA, succinylated BSA, and maleylated BSA) indeed interacted with H2B (**new Figure 2e**). In addition, the possible contribution of electrostatic interaction was also demonstrated by the observations that the binding of the AGEs to H2B was inhibited by 1 M NaCl (**new Figure 2f**) and by the deletion of a highly basic N-terminal region (Δ N35) from H2B (**new Figure 2g**). Furthermore, the AGEs were recognized by the scrambled fragments of the H2B N-terminal (scrambled 1, 2, and 3) comprising identical amino acids in different amino acid sequences (**new Figure 2g**). These data support our hypothesis that the electronegative potential of AGEs might be involved, at least in part, in the recognition by histone. These results are also analogous to a previous report showing that histone could form a tight complex with nuclear protein including acidic domain, such as heterochromatin protein 1, even in the presence of 500 mM NaCl, and this interaction could be dissociated with 1 M NaCl (Hara *et al.*, *EMBO reports*, 2, 920-925 (2001)). The high affinities of histones for electronegative molecules could be brought by the combined electrostatic interactions between the N-terminal arginines plus lysines of histones and electronegative molecules in the same manner as reported for the binding of histone tail with DNA (Hong *et al.*, *J Biol Chem*, 268, 305–314 (1993)). Meanwhile, we observed that lysozyme, a basic protein having an isoelectric point similar to histone proteins, did not show recognition specificity to AGEs (**Supplementary Figure 5**). This led us to speculate that, in addition to the electrostatic potential, some specific structures of H2B may contribute to the histone binding to the AGEs. We explained these reasons in the revised manuscript (page 11, line 17-page 12 line 9).

4. The DHA AGE product is quite different in structure from glucose or methylglyoxal products, but the authors data indicates that all such AGEs are bound by histones. This is unusual, and suggest that a unique motif of the AGE product shared by all the different glycation products is recognized. Do the authors have an idea about this novelty? While the authors have taken steps to understand this phenomenon, as described in Fig. 2, they can only conclude that “The data support the idea that fluorescent products generated in the

AGEs might be involved in the interaction with histones.”, but this is not particularly informative in regard to the molecular recognition shared by all the glycation products.

Response: It has been reported that DHA and its degradation products could form adduct species, such as CML, pentosidine, GA-pyridine, and LM-1, structurally similar to those obtained upon incubation of proteins with other reducing sugars (Dunn *et al.*, *Biochemistry*, 29, 10964–10970 (1990); Nagaraj *et al.*, *Proc. National Acad. Sci.*, 88, 10257–10261 (1991); Argirov *et al.*, *Biochim. Biophys. Acta - Gen. Subj.*, 1620, 235–244 (2003); Tessier *et al.*, *J. Biol. Chem.*, 274, 20796–20804 (1999)). In addition, as far as we know, no DHA-specific AGEs have been identified so far. Therefore, we believe that DHA gives AGEs similar to glucose, although there may be structural differences in the details.

Because lysine residues in proteins are major targets of glycation, we speculate that common feature of AGEs caused by all the different glycation products may be electronegative potential, not a unique motif or chemical structure of the AGE product shared by all the different glycation products. Consistent with this view, this manuscript shows that DHA-derived AGEs and other sugars-derived AGEs generally bind to histones (**Figures 2c and d**). Our speculation is also supported by the literature regarding the recognition mechanism of AGEs receptor, showing that one of the major common characteristics of known RAGE ligands is the net negative charge at physiological pH (Fritz *et al. Trends Biochem. Sci.*, 36, 625–32 (2011)). The new sentences concerning electrostatic interaction were added in the revised manuscript (page 11, line 17-page 12 line 9).

As for our conclusion that “The data support the idea that fluorescent products generated in the AGEs might be involved in the interaction with histones.”, we agree with the reviewer’s comment that this is not particularly informative in regard to the molecular recognition shared by all the glycation products. Therefore, the related sentences were deleted from the manuscript.

5. A key aspect of the paper is in regard to surface expression of histones on activated macrophages and their binding to AGE, depicted for the most part in Fig. 3. But there is only indirect evidence here that AGE binding is to histones, as the direct interaction to surface histone is not demonstrated, only correlated. Other AGE receptors on the surface of macrophages, such as scavenger receptor-A on macrophages, are likely to also bind AGE and may be the principal receptors. Direct evidence of binding of AGE to surface histone should be demonstrated.

Response: To directly demonstrate the binding of AGEs to histone H2B on the cell membrane, we conducted the experiment using the membrane-impermeable cross-linker (DTSSP). Cells were treated with biotin-labeled (Bt-) BSA or Bt-AGEs and interacting proteins were cross-linked by DTSSP. Cells were then lysed and subjected to pull-down followed by Western blotting. As shown in **new Figure 3f**, significant signals were observed when treated with Bt-AGEs. This indicates that AGEs are directly bound to H2B localized in the membrane. We also investigated the involvement of other AGEs receptors on macrophages using anti-histone or anti-AGEs receptor antibodies. As shown in **new Figure 3g**, the treatment with anti-H2B, H3, or H4 antibodies significantly inhibited AGEs binding to macrophages to about the same extent as anti-AGEs receptors (RAGE, AGER1, or CD36) antibodies. These new data demonstrate that AGEs bind to macrophages through interactions with histones, in addition to the known family of AGE receptors.

6. In Fig. 1A, there are many proteins that are pulled down by the AGE, but the authors focused on the histones? Is there a rationale for this? What about the identification of other proteins in that pull-down?

Response: Based on repeated experiments, the protein bands uniquely detected by the AGEs pull-down were histone proteins (bands a-c). Others were BSA or AGEs that were used for pull-down. That's why we focused on histone proteins.

7. Is there direct evidence that Histones 2A/B occur at the surface of cells, and are they localized in lipid rafts, as the data from Fig. 1A suggests?

Response: To directly show the presence of histones on the cell surface, a cell-surface biotinylation assay was conducted. Cells were treated with sulfo-NHS-biotin, a membrane-impermeable labeling reagent, and cell surface proteins were labeled with biotin. Biotinylated membrane proteins in RIPA soluble and lipid raft fractions were affinity purified and subjected to Western blotting. As a result, it was shown that some histone proteins are present on the cell surface (**new Supplementary Figure 2**). Unlike RAGE, which is completely localized in the RIPA-soluble fraction, histones were detected in both the RIPA-soluble and the lipid raft fractions (**new Supplementary Figure 2**).

8. Does the binding of AGEs to known AGE receptors, which are generally universally expressed, influence these pathways? It would appear from the pull-down evidence that histones are more prominent receptors for AGEs than the known family of AGE receptors. Could the authors provide some context as to how over the years such AGE receptors as histones were not identified in the many studies on AGE receptors?

Response: As mentioned earlier in the response to the reviewer's comment #5, the new data (**new Figure 3g**) show that AGEs bind to macrophages through the interaction with histones, in addition to the known family of AGE receptors. Among them, RAGE is particularly known for its involvement in inflammatory responses. Therefore, we investigated whether the inhibitory effect of AGEs on macrophage infiltration could be affected by RAGE. Our new result from *in vitro* invasion assay (**new Figure 5b**) showed that anti-H2B antibody, but not anti-RAGE antibody, could suppress the AGEs-dependent inhibition of macrophage infiltration. Thus, histones are more prominent receptors for AGEs than the known family of AGE receptors, RAGE.

There is a general concept that histones are nuclear proteins, and the presence of histone proteins in lipid raft fractions has been attributed to contamination from the nuclear fraction (Poston *et al.*, *Biochem. Biophys. Res. Commun.*, 415:355–360 (2011)). This may be the reason why histones have not been identified as AGE receptors until our current study. Thus, the role of histones as AGE receptors have been overlooked.

9. From the results in Fig. 5 the authors conclude that “These data indicated that AGEs have a function in regulating monocytes/macrophage recruitment to the inflammatory sites via the inhibition of the Plg activation”. However, there is no direct evidence that the mechanism is through inhibition of Plg activation.

Response: For the direct evidence that the mechanism is related to the inhibition of Plg activation, we performed a new *in vivo* experiment using a Plg activation inhibitor, tranexamic acid. As shown in **new Figure 5d**, the inhibitory effect of AGEs on macrophage infiltration was completely canceled by treatment with tranexamic acid. We also analyzed downstream factors of Plg activation *in vivo*. Plg is known to play a role in the activation of matrix metalloproteinases (MMPs), which is essential for monocytes/macrophages motility and invasion during the inflammatory response (Gong *et al.*, *J. Clin. Invest.* 118, 3012–

3024 (2008).). Hence, we investigated whether treatment with AGEs could attenuate the activation of MMP-9 in the peritonitis model. Western blotting with peritoneal lavage fluid from thioglycollate-stimulated mice showed that treatment with AGEs significantly suppressed MMP-9 activation compared to treatment with unmodified BSA (**new Figure 5e**). These results support our conclusion that AGEs have the ability to regulate monocytes/macrophage recruitment to inflammatory sites through inhibition of Plg activation.

10. The rather artificially in vitro produced AGE product with VitC and albumin is very heavily glycated, but this degree of glycation is perhaps not physiological. Do lower levels of glycated products produce similar effects or does it require dozens of glycated residues, as suggested by Suppl. Fig. 1, to produce the observed effects?

Response: As the reviewer pointed out, DHA-AGEs prepared *in vitro* may be more glycated than physiological AGEs. However, the same is true for in vitro-prepared glucose-derived AGEs. To assess the validity of the amounts of AGEs used in this study, we measured the concentrations of histone-bound AGEs in sera from wild-type mice and diabetic ob/ob mice. Elevated serum levels of AGEs have previously been reported in ob/ob mice (Kim *et al.*, *Exp. Dermatol.*, 27, 815–823 (2018).). Quantitative analysis using a competitive solid-phase binding assay revealed that serum levels of histone-binding proteins were significantly elevated in ob/ob mice compared to WT mice (**new Supplementary Figure 4a**). The concentrations of AGEs equivalent to DHA-BSA in WT mice and ob/ob mice were 191 $\mu\text{g/ml}$ and 318 $\mu\text{g/ml}$, respectively (**new Supplementary Figure 4b**). Therefore, the amount of AGEs used in this study is considered to be within the range that occurs under physiological or pathophysiological conditions.

Response to Reviewer #2:

The scientific importance of this manuscript is that the authors identified for the first time histone as a novel AGE binding cellular protein. Then, the authors tried to explain homeostatic role of AGEs in regarding with their binding activity to histone using variety of advanced instrumental and molecular biological assays both in vitro and in vivo. However, following comments should be considered for publication of this work in corresponding peer reviewed journal.

Major comments:

1. Ascorbic acid is also known for their pro-oxidant activity. Accordingly, how did you recognize that these products are whether the AGEs or protein oxidation products. Please specify how did you control the formation of intermediate products of glucose metabolites, such as glycoaldehyde, GA-pyridine, and N-pyrrolylsine in your designated experiment conditions?

Response: As the reviewer pointed out, ascorbate shows a pro-oxidant activity, so do other reducing sugars such as glucose. Therefore, as long as reducing sugars are used as a source of AGEs, the formation of protein oxidation products is unavoidable. This means that AGEs in general are a mixture of both glycation and oxidation products. It is almost impossible to control the formation of intermediate products, including GA-pyridine and N-pyrrolylsine, and we did not do in our study. Data using these small products (**Figure 2e, f**) have been deleted from the manuscript because we were unable to detect them in the AGEs used in this study.

2. Please demonstrate at least one result, where produced AGEs were quantitatively measured after incubating BSA with DHA in this experiment.

Response: Following the reviewer's advice, we quantitatively measured *N* ϵ -carboxymethyllysine (CML), a well-known AGE, generated in the DHA-modified BSA. The concentration of CML was significantly increased in DHA-modified BSA (AGEs, 0.64 ± 0.30 mmol/mol Lys), compared to unmodified BSA (0.04 ± 0.02 mmol/mol Lys). The results were included in the revised manuscript (**Supplementary Figure 1c**).

3. Please prove pull down assay and solid phase binding assays by using AGEs - specific antibodies to demonstrate that the fractions are exactly matched to the AGEs.

Response: To demonstrate the presence of DHA-derived AGEs in the fractions with histone binding activity, we attempted to detect the AGEs using an anti-AGEs monoclonal antibody established in our laboratory (Chikazawa *et al.*, *J. Biol. Chem.*, 288, 13204–13214 (2013)). **Figure 2b** shows that H2B-binding fractions were recognized by the antibody. The data support our finding that AGEs bind to histone H2B.

4. It seems like that experimental design is systematically not well organized, for example, the authors used mouse peritoneal cells to determine histone dependent binding of AGEs to macrophages, whereas J774A.1 cell line was applied for determination of regulating activity of AGEs against the monocytes or macrophages. And cytotoxic effects of AGEs are tested on human endothelial EA. hy926 cell lines. Such unlinked arrangement is making the results uncomprehensive. Similarly, experiments in first 5 results have been obtained for histone H2B, while last experiment was conducted using histone H4.

Response: As the reviewer pointed out, this paper consisted of two main parts with multiple cells used and was therefore confusing. In this study, to clarify AGEs-binding cells, we first used mouse peritoneal cells and identified macrophages as target cells (**Figures 3a and b**). Based on this result, we evaluated AGEs binding to macrophage/monocyte cell lines (J774A.1 and RAW264.7) to examine the universality of AGEs binding to macrophages (**Figures 3c and d**). These two cell lines, similar to mouse primary macrophages, have been reported to express membrane-bound H2B, which contributes to plasminogen binding (Das *et al*, *Blood*, 110, 3763–72 (2007)). However, in our preliminary experiments, RAW264.7 showed marked non-specific plasminogen binding (which could not be blocked by the inhibitor of plasminogen receptors, tranexamic acid) compared to J774A.1 (data not shown). Therefore, we used J774A.1 cells for subsequent experiments on the binding of AGEs (**Figures 3e, f, g, and h**), the inhibition of plasminogen binding (**Figure 4d**), and the inhibition of plasminogen activation (**Figure 4g**), except that the inhibition of plasminogen binding to macrophages by AGEs was also confirmed in peritoneal macrophages (**Figure 4e**).

Human endothelial EA. hy926 cell lines were used for the effect of AGEs on histone cytotoxicity because endothelial cells are known to be the major target of histone proteins, especially histone H4 (Xu *et al.*, *Nat. Med.*, 15, 1318–1321 (2009)). However, to avoid unnecessary confusion, the last part regarding histone cytotoxicity has been removed from the manuscript.

5. It is known that AGEs largely bound to the cell surface through their receptor – RAGE. Please demonstrate a proof that in this experiment, AGEs specifically bound to histone proteins, not with RAGE.

Response: To answer the reviewer's remarks, we investigated the involvement of other AGE receptors on macrophages using anti-histone or AGE receptor antibodies. As shown in **new Figure 3g**, treatment with anti-H2B, H3, or H4 antibodies significantly inhibited the binding of AGEs to macrophages, similar to treatment with antibodies to the known family of AGE receptors (RAGE, AGER1, or CD36). Given previous findings regarding the binding of AGEs to these receptors, the results are very reasonable. However, the new data also establish that the binding of AGEs is not specific to these AGEs receptor, but they can also bind histones.

Response to Reviewer #3

In this manuscript, Uchida and colleagues report that advanced glycation end products (AGEs) might mediate homeostatic responses by binding to histone proteins. The authors utilize dehydroascorbic acid (DHA)-modified serum albumin as an AGE probe to screen for binding proteins in the lipid raft fraction from mouse splenocytes and identify histone as an AGE-binding protein. The authors then demonstrate the binding capacity of different types of AGEs (such as glucose, methylglyoxal, and glycolaldehyde glycation products) to histones. The authors also identify that AGEs regulate monocytes/macrophage recruitment and inhibit histone toxicity through the formation of aggregates with histone H4. The general concept of this paper is interesting however, the data is very preliminary and I have major concerns regarding this manuscript:

1) In a strictly chemical sense, DHA is not a sugar molecule and the structures of DHA-modified amino acid residues are quite different from AGEs induced by reducing sugars, despite the similar linkage. Thus, there is little comparability between the DHA-modified proteins and classic AGEs, making them a poor mimic (see schematics). To utilize a DHA-modified BSA as a pull-down probe also does not have any physiological relevance due to the low concentration of DHA in human blood serum.

Response: Several previous studies reported that DHA and its degradation products form the same adduct species, such as CML, pentosidine, GA-pyridine, and LM-1, that are formed when proteins are incubated with other reducing sugars (Dunn *et al.*, *Biochemistry*, 29, 10964–10970 (1990); Nagaraj *et al.*, *Proc. National Acad. Sci.*, 88, 10257–10261 (1991); Argirov *et al.*, *Biochim. Biophys. Acta - Gen. Subj.*, 1620, 235–244 (2003); Tessier *et al.*, *J. Biol. Chem.*, 274, 20796–20804 (1999)). Other studies have also shown that DHA and its degradation products are causally involved in the formation of AGEs *in vivo* (Nagaraj *et al.*, *Proc. National Acad. Sci.*, 88, 10257–10261 (1991)). In fact, following the advice of other reviewer (reviewer #2), we measured *N* ϵ -carboxymethyllysine (CML), a well-known AGE, generated in the DHA-modified BSA and observed that the concentration of CML was significantly increased in DHA-modified BSA (AGEs, 0.64 \pm 0.30 mmol/mol Lys), compared to unmodified BSA (0.04 \pm 0.02 mmol/mol Lys) (**Supplementary Figure 1c**). Thus, it is evident that there is comparability between the DHA-derived AGEs and classic glucose-derived AGEs. Consistent with this view, no DHA-specific AGEs have been identified so far. In addition, this manuscript shows that sugars-derived AGEs, including glucose-derived AGEs, also bind to histones as well as DHA-derived AGEs (**Figures 2c and d**).

On the other hand, DHA is present in the blood at about 10 - 20 μ M in blood (Sato *et al.*, *Biological Pharm. Bulletin*, 33, 364–369 (2010)). In addition, as mentioned above, DHA and its degradation products are causally related to the formation of AGEs *in vivo*.

Therefore, we believe that the use of a DHA-modified BSA as a pull-down probe have physiological relevance.

2) The binding mode of AGEs to histone is not clear. Since there are hundreds of different glycation product structures induced by multiple sugar molecules or their metabolites, it is extremely difficult to imagine a uniform binding mode where the positively charged AGEs could tightly bind histone proteins, which are rich in Lys and Arg. The authors have tested the binding ability of AGEs formed by distinct sugar molecules, such as Glc, MG, GA, GCA, 3-DG, and GO. However, it is meaningless to have this comparison due to their entirely different chemical structures. Finally, the authors do not provide an adequate negative control of highly positively charged protein and/or aggregation-prone protein that does NOT bind these AGEs.

Response: As the reviewer pointed out, AGEs are a mixture of numerous different types of glycation product structures induced by multiple sugar molecules or their metabolites. Therefore, it is reasonable to imagine that a uniform binding mode may not be involved in the binding of AGEs to histone. That is why we speculate that electrostatic interaction may be involved in the interaction of AGEs with positively charged histone proteins. To validate this speculation, we tested whether histone H2B could recognize electronegative molecules and confirmed that acylated proteins (acetylated BSA, succinylated BSA, and maleylated BSA) indeed interacted with H2B (**new Figure 2e**). Furthermore, the possible contribution of electrostatic interaction was also demonstrated by the observations that the binding of the AGEs to H2B was inhibited by 1 M NaCl (**new Figure 2f**) and by the deletion of a highly basic N-terminal region (Δ N35) from H2B (**new Figure 2g**). In addition, AGEs were recognized by the scrambled fragments of the H2B N-terminal (scrambled 1, 2, and 3) comprising identical amino acids in different amino acid sequences (**new Figure 2g**). These data support our hypothesis that the electronegative potential of AGEs might be involved, at least in part, in the recognition by histone. These results are also analogous to a previous report showing that histone could form a tight complex with nuclear protein including acidic domain, such as heterochromatin protein 1, even in the presence of 500 mM NaCl, and this interaction could be dissociated with 1 M NaCl (Hara *et al.*, *EMBO reports*, 2, 920-925 (2001)). The high affinities of histones for electronegative molecules could be brought by the combined electrostatic interactions between the N-terminal arginines plus lysines of histones and electronegative molecules in the same manner as reported for the binding of histone tail with DNA (Hong *et al.*, *J Biol Chem*, 268, 305–314 (1993)). Moreover, we observed that lysozyme, a basic protein having an isoelectric point similar to histone proteins, did not show any recognition specificity to AGEs (**Supplementary Figure 5**). This observation can be a negative control of highly positively charged protein requested by the reviewer. These are the reasons for speculating that, in addition to the electrostatic potential, some specific structures of H2B may contribute to the histone binding to the AGEs. These have been described in the revised manuscript (page 11, line 17-page 12 line 9).

3) The formed AGE structures are highly dependent on the microenvironment of the glycated proteins because of the rearrangements that form crosslinks (e.g., glucosepane). Even though the authors have tested two AGE examples with exact chemical structures (GA-pyridine and N ϵ -pyrrolylsine), these two free small molecule mimics do not represent the complexity and specificity of AGEs formed on different proteins. More structural and biochemical insights should be provided in order to understand the interactions between histones and AGEs.

Response: We agree with the reviewer's comment that the modified small molecules, such as GA-pyridine and N ϵ -pyrrolylsine, do not represent the complexity and specificity of AGEs. In addition, we were unable to detect these small molecules in the AGEs used in this study. Therefore, data using these small products have been deleted from the manuscript. On the other hand, as stated above in response to the reviewer's comment #2, we speculated that electrostatic interaction can be involved in the binding of AGEs with positively charged histone proteins and revealed that the electronegative potential of AGEs might be involved, at least in part, in the recognition by histone.

4) The pull-down analysis presented at the basis of the manuscript (Figure 1a) is poorly designed and executed. It relies on a visual identification of Coomassie bands (which selection seems random). Instead, authors should apply a quantitative proteomic analysis that will provide an accurate and comprehensive determination of the AGE binding proteins.

Response: We understand that our pull-down experiment is a classic "comparative" approach for identifying target binding proteins: comparison of native protein with AGEs. Using the same approach, however, we have successfully identified several binding proteins, including C1q as a serum AGEs-binding protein (Chikazawa et al., *Biochemistry* 55, 435-446, 2016) and apolipoprotein E as a pyrrole-binding protein (Hirose et al., *J. Biol. Chem.* 294, 11035-11045, 2019). It is not a state-of-the-art method, but it still works. In addition, as commented by the reviewer, it basically relies on a visual identification of CBB- or silver-stained bands. However, at least in this experiment, the selection was not random, as most of the proteins present in the AGEs-pulldown rather than the BSA-pull-down were histone proteins. Others are AGEs that were used for pull-down. Quantitative proteomic analysis may be the best approach, but it is not readily available in our laboratory.

5) The logic and flow of the manuscript is confusing. On one hand, the authors first identify that the cell-surface plasminogen receptor, H2B, is a binding target of DHA-modified BSA and AGEs. On the other hand, they next claim that AGEs could form aggregates with other histones to inhibit histone toxicity.

Response: This has also been pointed out by other reviewers. As the reviewer mentioned, the structure of this paper was confusing due to the existence of two different streams. Therefore, we decided to remove the data on cytoprotective effect of AGEs via binding to histone H4 from the manuscript and decided to focus on AGEs-dependent regulatory effect on monocytes and macrophages via binding to histone H2B in the revised manuscript.

6) The downstream biological effects of this binding activity are not clearly elucidated. More biochemical, in cellulo and in vivo experiments are required to uncover the function of the interactions between AGEs and histones. For example, the experiments performed in Fig.5 should also be performed with truncated and mutant histones.

Response: As a downstream biological effect of the interaction between AGEs and histone H2B, monocytes/macrophages cell line J774A.1 was used to assess plasminogen activation (**new Figure 4g**). Consistent with the result obtained from experiments with recombinant H2B (**Figure 4d**), plasmin formation was promoted by treatment of cells with plasminogen, which was suppressed by the treatment with AGEs. In addition, to investigate whether the inhibitory effect of AGEs on macrophage infiltration was due to a similar mechanism, we conducted a new *in vivo* experiment using a plasminogen activation

inhibitor, tranexamic acid. As shown in **new Figure 5d**, the inhibitory effect of AGEs on macrophage infiltration was completely canceled by treatment with tranexamic acid. We also added a new *in vivo* analysis for downstream factors of plasminogen activation. Plasminogen is known to play a role in the activation of matrix metalloproteinases (MMPs), and activation of MMP-9 is essential for monocytes/macrophages motility and invasion during the inflammatory response (Gong *et al.*, *J. Clin. Invest.* 118, 3012–3024 (2008)). Hence, we investigated whether treatment with AGEs could reduce the activation of MMP-9 in a model of peritonitis. Western blotting with peritoneal lavage fluid in thioglycollate-stimulated mice showed that AGEs significantly suppressed MMP-9 activation compared to untreated and unmodified BSA (**new Figure 5e**). Following the reviewer's suggestion, we further conducted the infiltration assay using mutant H2B-expressing cells. Unfortunately, however, the expression of truncated histone induced marked cell death, and therefore it was not possible to assess its impact on the infiltration.

Reviewer #4 (Remarks to the Author):

The manuscript by Itakura et al investigates how sugar-protein compounds called glycation end products (AGEs) function and attempts to discover their binding partners in cells. They use DHA modified serum albumin as an example of AGE to screen for cellular binding partners in lipid raft fractions prepared from mouse spleen cells. Ultimately, histones are shown to bind DHA (AGE). More specifically, AGE bound histone component H2B which serves as a cell surface plasminogen receptor. Although, AGE species are known to be pro-inflammatory and are implicated in inflammatory diseases the authors propose that AGE in fact can be immunosuppressive by preventing inflammatory monocyte/macrophage recruitment and activation.

Advanced glycation end products or AGEs form when reduced sugars covalently react w/ proteins. That AGEs are associated with diseases as pathogenic molecules is known, but “how” is a black box. In order to characterize the AGE-associated mechanisms, authors used a 3-step approach:

- 1) Find AGE-binders
- 2) Gain / Loss of function experiments to validate AGE binding
- 3) Evaluate the dynamic efforts of AGE during disease/models

Major concerns:

The biochemical studies related to the discovery of binding partners for DHA modified serum albumin (AGE) are by and large convincing. However, the physiological significance of this particular AGE species as a bona fide immunoregulatory complex is less convincing. This is important because the authors are proposing a relatively new function of AGE species as anti-inflammatory instead of what has been well established that AGE serve as proinflammatory entities.

Response: We agree with the reviewer's comment that the physiological significance of this particular AGE species as a bona fide immunoregulatory complex was not very convincing. We therefore conducted further experiments to establish our proposal. Especially, as a downstream biological effect of the interaction between AGEs and histone H2B, monocytes/macrophages cell line J774A.1 was used to assess plasminogen activation (**new Figure 4g**). Consistent with the result obtained from experiments with recombinant

H2B (**Figure 4d**), plasmin formation was promoted by treatment of cells with plasminogen, which was suppressed by the treatment with AGEs. In addition, to investigate whether the inhibitory effect of AGEs on macrophage infiltration was due to a similar mechanism, we conducted a new *in vivo* experiment using a plasminogen activation inhibitor, tranexamic acid. As shown in **new Figure 5d**, the inhibitory effect of AGEs on macrophage infiltration was completely canceled by treatment with tranexamic acid. This result indicates that AGEs regulate monocytes/macrophage recruitment via the inhibition of the Plg activation, but not via its cytotoxicity. This notion is also supported by our new data (**Supplementary Figure 11**), showing that AGEs do not have any cytotoxic effect on macrophages even in the concentration range up to 300 µg/ml for 72 h. We also added a new *in vivo* data on downstream factors of plasminogen activation. Western blotting with peritoneal lavage fluid in thioglycollate-stimulated mice showed that AGEs significantly suppressed MMP-9 activation compared to untreated and unmodified BSA (**new Figure 5e**). These new data further strengthen our findings that AGEs can serve as anti-inflammatory entities.

With respect to AGE binding to monocytes/macrophages can the authors use another myeloid cell type to show the specificity to monocytes? Perhaps myeloid dendritic cell?

Response: In accordance with the reviewer's comment, we tested other cell-types (dendritic cells, neutrophils, and eosinophils) in addition to monocytes and macrophages and observed that AGEs bind primarily to macrophages and monocytes (**New Figure 3b**).

AGEs bind to macrophages via histones. In order to show that histones are really required for this binding, can histones be blocked somehow to do a loss of function expt to make data more convincing?

Response: To answer the reviewer's comment, we conducted a new experiment using anti-histone and anti-AGE receptor antibodies. As shown in **new Figure 3g**, treatment with anti-H2B, H3, or H4 antibody significantly inhibited AGEs binding to macrophages to about the same extent as anti-AGE receptors (RAGE, AGER1, or CD36) antibodies. These data indicate that AGEs bind to macrophages via histones, in addition to the well-known AGE receptors.

The studies described in Fig 5 and 6 where the authors attempt to demonstrate the physiological significance of AGEs as immunosuppressive molecules is not as convincing. The *in vitro* invasion assay needs to be described better. The TG treatment shows reduction in F480+ GR1 low macrophages suddenly at 72 hrs after treatment, however, the trend at 48 hrs shows as actual increase in macrophage number in TG treated animals. Similar trend can be observed with monocytes where they are actually increased at 24 followed by a precipitous decline at 48 hrs. Considering the model of peritonitis this looks less like an issue with recruitment but it appears that monocytes and macrophages in AGE treated mice are undergoing cell death. This needs to be clarified.

Response: As mentioned above in response to the reviewer's major concern, we conducted further experiments to establish our proposal and added some new data to the revised manuscript. We believe that these new data further strengthen our findings that AGEs can serve as anti-inflammatory molecules. As for a description on the *in vitro* invasion assay, the related text was rewritten with reference to the reviewer's comments (page 8, lines 22-30). To clarify the mechanism underlying the inhibitory effect of AGEs on monocytes/macrophage recruitment into peritoneal cavity in more detail, we carried out a new *in vivo* experiment using a plasminogen activation inhibitor, tranexamic acid. As shown in **new Figure 5d**, the inhibitory effect of AGEs on macrophage recruitment was

completely canceled by the treatment with tranexamic acid. This result indicates that AGEs regulate monocytes/macrophage recruitment via the inhibition of the Plg activation, but not via its cytotoxicity. This notion is supported by our new data (**Supplementary Figure 11**) showing that AGEs do not have any cytotoxic effect on macrophages even in the concentration range up to 300 µg/ml for 72 h.

Furthermore, the authors show that in addition to the putative recruitment defect the macrophages exhibit a M2 phenotype. However, the relative gene expression differences are minor at best. The authors need to assess protein expression and also perform assays that show the physiological significance of these proposed intrinsic immunoregulatory nature AGE treated macrophages – perhaps even some in vitro assays may suffice.

Response: Based on our findings that AGEs bind to histone H2B on the surface of macrophage and regulate plasminogen activation, we speculated that predominant action of AGEs is the suppression of proinflammatory cell infiltration through binding to H2B. This means that the changes in gene expression shown in **Figure 5e** are due to a decrease in infiltrated inflammatory cells, not a promotion of macrophage polarization. This may be the reason why the differences in the gene expression were relatively small. In accordance with the reviewer's suggestion, we examined the expression of protein markers for M1 (CD80) and M2 macrophages (CD206) by flow cytometry. However, the differences were negligible, and significant increase of M2 macrophage was not observed (**new Supplementary Figure 12**).

In Fig.6 the authors perform a survival experiment where they show that histones+ BSA has a notable effect. This effect is not trivial and needs to be explored further.

The authors at the very least show a protective effect of AGE in a more robust inflammatory model. Perhaps to begin with they should use a sepsis model – LPS induced sepsis, and cecal slurry model.

Response: Based on the reviewer's suggestion, we carried out new experiments using cecal ligation and puncture (CLP)-induced sepsis model for a more pathophysiological approach. As a result, both prolonged survival and decreased levels of inflammatory cytokines in the blood were observed in mice treated with AGEs. However, other reviewers pointed out that the structure of this paper was confusing due to the existence of two different streams. Therefore, we decided to remove the data on cytoprotective effect of AGEs via binding to histone H4 from the manuscript.

The comments offered by the four reviewers have been very helpful in formulating what we believe is a stronger paper. We appreciate these thoughtful comments and hope that our responses and in particular our revisions have allowed this paper to achieve a priority sufficient for publication in Nature Communications.

Response to Reviewer #1:

Here the authors present evidence that histone H2A/B and H4 are binding partners for dehydroascorbic acid (DHA)-modified albumin. The presence of histones on cell surfaces has been observed by others, but the key finding of this paper is that such displayed histones can interact with AGEs. The histone interaction involves electrostatic mechanisms, as high salt (1-3M) disrupts such interactions, but the authors demonstrate than simply being a basic protein, such as lysozyme, does not confer binding to the AGEs.

The authors observed that AGEs bound only to macrophages and monocytes but not dendritic cells and neutrophils for example. While the evidence indicates that histones can bind AGEs it is not clear as to what approximate percentage of binding of AGEs to cells is attributable to histone interactions. According to Fig. 3g, binding is slightly, albeit statistically significantly, inhibited by antibodies to histones, implying that histone binding is contributing to only a small fraction of AGE binding. An intriguing experiment would be add histones to cells, including monocytes and even dendritic cells and neutrophils, to observe whether that would enhance surface histone expression and confer binding of AGEs to the cells.

Response: We agree with the reviewer’s comment on the effects of antibodies on the binding of AGEs to cells (**Fig. 3g**). Following the reviewer’s suggestion, we carried out an additional experiment using recombinant histone H2B to demonstrate the involvement of H2B in the binding of AGEs to cells. New data (**new Supplementary Figure 10**) show that the treatment of peritoneal cells with recombinant H2B significantly enhanced the association of H2B to the cell surface of macrophages, monocytes, and dendritic cells, resulting in remarkable increase in the binding of AGEs. Therefore, the addition of an extracellular histone to cells further enhanced surface expression of histones and conferred binding of AGEs to the cells. The data support our view that the binding of AGEs to cells is attributed to H2B. We appreciate the reviewer for suggesting this experiment. These new findings are included in the Result section (**page 7, lines 23-29**) and the Discussion section (**page 13, lines 4-12**) of the revised manuscript.

Other studies have reported that plasminogen binding to cell surfaces may be to histones. Here the authors examined that AGEs inhibited plasminogen binding to histones, implying the binding sites for both are similar. Interestingly, the authors show that AGEs (Fig. 4) can inhibit binding of plasminogen to macrophages. The consequences of such data implies that AGEs can functionally interfere with macrophage movements an involvements in inflammation, also consistent with evidence of inhibition of MMP9 activation by AGEs, independently of RAGE recognition. However, it is a bit unclear as to whether plasminogen binding can displace histones on their surface through their binding to the surface through unknown mechanisms.

Response: To answer the concerns of the reviewer, we examined changes in the cell surface and extracellular levels of

Supporting data for review

Plg treatment does not cause H2B release from cell surface. J774A.1 cells were labeled with membrane impermeable EZ-Link Sulfo-NHS-LC-Biotin for 30 min on ice and then treated with either 200 nM or 1000 nM Plg. After incubation for 15 min, cells were centrifuged, and supernatants were collected as “Extracellular fraction”. The cell lysates were subjected to pull-down with streptavidin-coupled magnetic beads, and the resulting precipitates were collected as “Membrane fraction”. Proteins in Membrane fraction (**a**) and Extracellular fraction (**b**) were transferred to PVDF membrane, stained with Ponceau S to visualize total proteins, and subjected to western blotting for H2B.

histone H2B before and after treatment with plasminogen (Plg) and observed that there was no significant effect of Plg treatment for H2B levels (Refer to **Supporting data for review** on the right). Data from several studies indicate that the interaction of Plg with cell-surface Plg receptors, including histone H2B, accelerates conversion of Plg to plasmin (Plm) (Longstaff *et al.*, *Blood*, 93, 3839-3846 (1999)) and enhances the catalytic activity of Plm (Gonzalez-Gronow *et al.*, *Arch. Biochem. Biophys.*, 286, 625-628 (1991)). In addition, Plm formed on the cell surface is retained on the cell membrane and protected from inactivation by its inhibitor (Plow *et al.*, *J. Cell Biology*, 103, 2411-20 (1986)). Considering these previous reports and our findings (Supporting data for review), the regulation of macrophage dynamics by AGEs is thought to result from the inhibition of Plg activity on cell surface. These statements related to plasminogen activation on cell surface are now included in the Result section (**page 7, line 32-page 8, line 2**) and the graphical summary of this study was added in **new Supplementary Figure 20**.

An obvious question is whether DNA might inhibit histone binding to AGEs, but this was not directly address.

Response: To clarify the effect of DNA on histone binding to AGEs, we carried out a competitive solid phase binding assay and found that DNA inhibits histone binding to AGEs. This finding is included in the Result section (**page 6, lines 2-3**) as **new Supplementary Figure 5** in the revised manuscript.

Altogether, this study provides strong evidence that cell surface histones in macrophages can function as a receptor for AGEs, with functional consequence on interactions with plasminogen and other signaling pathways previously associated with only AGE interactions with RAGEs. An unsolved riddle at present is the mechanism of histone localization in lipid rafts or membrane microdomains. This reviewer would propose the possibility that histones may interact through charge interactions with highly sulfated glycosaminoglycans, e.g. heparan sulfate, and thereby associate with the membrane.

Response: Previous reports have indicated that histone proteins can bind not only to phosphatidylserine (PS), as referenced in the original manuscript, but also to negatively charged cell surface molecules such as heparan sulfate proteoglycans (Watson *et al.*, *J. Biol. Chem.*, 274, 21707-21713 (1999)). In addition, heparan sulfate-carrying core proteins, such as syndecan-4, have been shown to localize in lipid raft compartment (Tkachenko *et al.*, *J. Biol. Chem.*, 277, 19946-19951 (2002)), and their expression levels vary among cell types; activated macrophages, monocytes, and dendritic cells display high expression of syndecan-4, while neutrophils and eosinophils express low levels of syndecan-4 (Parish *et al.*, *Nat. Rev. Immunol.*, 6, 633-643 (2006); Averbek *et al. Exp. Dermatol.*, 16, 580-589 (2007)). These previous studies suggest that, as proposed by the reviewer, histones bound to the highly sulfated glycosaminoglycans may also be involved in the binding to AGEs. Therefore, new statements related to the mechanism of histone localization have been added to the revised manuscript (**page 13, lines 4-12**).

In any case, the potential anti-inflammatory nature of histone/AGE interactions proposed here is intriguing and potentially of biological significance.

Response: We are very pleased to receive favorable reviews from the reviewer.

Response to Reviewer #3:

In this resubmission, the authors made efforts to add additional data and reorganize the

figures. There are more details in this new version in comparison to the original one. However, the overall content of the manuscript and findings did not change. The authors still fail to address the major concerns raised by the reviewers, including more mechanistic insights and physiological relevance.

I still stick to the original point of view that this paper is not suitable for publication in *Nat. Commun.* It could be transferred to another journal which requires less mechanistic insights (see attached).

Response: In response to our first submission, the reviewer raised multiple concerns regarding mechanistic insights and physiological relevance. The issues included (i) comparability between the DHA-derived AGEs and classic AGEs, (ii) binding mode of AGEs to histone, (iii) structure of AGEs involved in the binding to histone, (iv) methodology (pull-down assay), (v) structure of manuscript, and (vi) downstream biological effects of AGEs-histone binding. We thought that our experiments and explanations in the response to the reviewer were sufficient to answer all these concerns and to show mechanistic insights and physiological relevance of our study. However, the reviewer commented that we should have addressed more mechanistic insights and physiological relevance. Other reviewers, who are very positive about this paper, have also suggested doing some experiments related to these issues.

Based on the reviewers' comments and suggestions, we conducted the additional new experiments in this submission as follows: (i) To gain insight into the involvement of histones in the binding of AGEs to the cells, we investigated the effects of the addition of recombinant histone H2B and observed that the treatment of peritoneal cells with recombinant H2B significantly enhanced the association of H2B to the cell surface of macrophages, monocytes, and dendritic cells, resulting in remarkable increase in the binding of AGEs (**new Supplementary Figure 10**). Therefore, the addition of an extracellular histone to cells further enhanced surface expression of histones and conferred binding of AGEs to the cells. The data support our view that the binding of AGEs to cells is attributed, at least in part, to H2B. These new findings are included in the Result section (**page 7, lines 23-29**) and the Discussion section (**page 13, lines 4-12**) of the revised manuscript. (ii) Because histone is a highly basic protein, we speculated that electrostatic interactions might be involved in the recognition of AGEs by histone. Hence, we investigated the effect of DNA on histone binding to AGEs using a competitive solid phase binding assay and observed that histone binding to AGEs was significantly inhibited by DNA (**new Supplementary Figure 5**). (iii) To provide further evidence that AGEs directly inhibit the migration of monocytes/macrophages to the peritoneum, we examined the effects of AGEs on cell death and *in situ* proliferation of peritoneal macrophages in *in vivo* peritonitis model and revealed that the treatment with AGEs had no significant effect on cell death and proliferation (**new Supplementary Figures 16 and 17**). The new data exclude the possibility of AGEs to be involved in cell death and cell proliferation, which supports our migration hypothesis. These results are included in the Result section of the revised manuscript (**page 9, lines 8-10 and page 9, lines 28-30**). (iv) To better understand the biological impact of AGE binding, we performed RNA-Seq analysis of macrophages isolated from the BSA- or AGEs-treated mice. The results showed that several pro-inflammatory M1-associated genes, such as IL-1b, Ccr7, Igtp, and Sell, were significantly downregulated in macrophages from the mice treated with AGEs compared to those treated with BSA (**new Supplementary Figure 18**). Conversely, AGEs-treatment increased the expressions of anti-inflammatory M2-associated genes, including GATA2, Clec10a, Snn, Pcdh7, Stxbp6, and Gar1 (**new Supplementary Figure 18**). Differential expression analysis data further strengthened our conclusion that AGEs could suppress the

infiltration of proinflammatory monocytes/macrophages. These findings are included in the Result section (**page 10, lines 8-11**). The RNA-Seq data have been deposited into the Gene Expression Omnibus (accession code: GSE195558, secure token: upexsueupjylhwhf). (v) We conducted *in vivo* experiments using cecal ligation and puncture (CLP)-induced sepsis model and observed that mice treated with AGEs rather than unmodified BSA displayed a significantly prolonged survival time (**new Figure 5g**). The data suggest that AGEs have a function of regulating monocytes/macrophage recruitment to the site of inflammation through inhibition of Plg activation, which may contribute to protection against inflammatory disorders. These *in vivo* data are included in the Result section (**page 10, lines 13-16**) and the Discussion section (**page 13, lines 25-30**) of the revised manuscript. In addition, regarding physiological relevance, the sentences on the possible involvement of some specific structures of H2B in the histone binding to the AGEs, which was addressed in the previous submission, have been retained in the current version of the manuscript (**page 12, lines 26-31**). We believe that these new findings have made this paper much more compelling and solid.

Despite these new data, we also admit that direct evidence of the contribution of the interaction between AGEs and histone H2B in physiological and pathophysiological settings is still limited. This limitation may be due, at least in part, to a lack of methodology for specifically inhibiting the interaction between AGEs and H2B. In this regard, we attempted to use antibodies to block this interaction; however, an anti-H2B antibody showed a significant but only modest decrease in AGEs binding (**Fig. 3g**). We speculate that this may be due to redundancy and functional compensation between the different types of histones in binding of AGEs. In addition, as an alternative approach, we also attempted to inhibit the interaction between AGEs and H2B using both gene silencing of extrachromosomal H2B and dominant negative mutants of H2B (data not shown). However, neither method worked for unknown reasons. Further study is under way in our laboratory to elucidate specific regulatory mechanisms of AGEs-H2B interactions and to establish their physiological relevance. According to the editor's suggestion, these explanation and limitation of this study have been included in the Discussion section of the revised manuscript (**page 13, line 31-page 14, line 8**).

We sincerely hope that the reviewer finds the revised paper suitable for publication in Nature Communications.

Response to Reviewer #5:

In this present manuscript, authors identified for the first time that histones can play as a novel AGE binding cellular protein, and its manifestation in monocyte/macrophage recruitment. Authors have reasonably justified majority of comments from the first reviewer and others. The author gave the explanation that as they are using reducing sugars it is unavoidable to formation of these products and they have deleted data (which is causing the concern) of these products from the study because they were not able to detect them in the AGEs used in the study.

I think overall work is novel and thought provoking, the manuscript structure is well organized and well written and should be consider for the publication.

Response: We are very pleased to receive favorable reviews form the reviewer.

Response to Reviewer #6:

Itakura et al provide an improved version of the manuscript adding novel pieces of data to support their conclusions but some major concerns remain. Firstly, the new evidence

supporting a direct binding of AGE to histones using assays with blocking antibodies is useful, although the effects shown are mild, raising the question whether there is significant redundancy between the different types of histones bound by AGE. This limitation should be at least discussed in the manuscript.

Response: As the reviewer pointed out, the blocking antibody for H2B showed a significant but only modest decrease in AGEs binding. Given that all types of histones are localized on the cell membrane of macrophages (**Fig. S2**) and have capacities for binding to AGEs (**Figs. 1b, c, and d**), this is presumed to be due to redundancy and functional compensation between the different types of histones in binding of AGEs, as suggested by reviewer. Therefore, we added new sentences to the revised manuscript to describe this methodological limitation (**page 13, line 31-page 14, line 3**).

To demonstrate the direct binding of AGEs to H2B on the cell surface in another way, we carried out the experiments using recombinant H2B based on the suggestion by other reviewer (reviewer #1). The treatment of peritoneal cells with recombinant H2B enhanced the association of H2B to the cell surface of macrophages, monocytes, and dendritic cells, which led to the remarkable increase in the binding of AGEs (**new Supplementary Figure 10**). These new data support our claim that the binding of AGEs to cells is attributed, at least in part, to H2B.

In addition, there are two major limitations in the current version of the manuscript that should be addressed:

Major concerns:

1. The evidence that AGE directly inhibits migration of monocytes/macrophages to the peritoneum is rather weak and other possible explanations such as cell death or *in situ* proliferation differences are not properly addressed. This should be done at least with simple flow cytometric analysis of apoptotic (using ie Annexin V staining) and proliferation markers (ie Ki-67, even better using BrdU/EdU incorporation experiments). Excluding these two options would greatly strengthen the migration hypothesis. This is particularly important as the *in vitro* viability assay seems to be problematic with some values above 100%, which is impossible. I would recommend repeating these experiments using a viability dye and flow cytometry.

Response: Following the reviewer's suggestion, to provide further evidence that AGEs directly inhibit the migration of monocytes/macrophages to the peritoneum, we examined the effects of AGEs on cell death and *in situ* proliferation of peritoneal macrophages in *in vivo* peritonitis model. Flow cytometric analysis using Annexin V and anti-Ki-67 antibody revealed that the treatment with AGEs had no significant effect on cell death and proliferation (**new Supplementary Figure 17**). These findings were also confirmed by both *in vitro* Annexin V staining and *in vitro* BrdU incorporation assay (**new Supplementary Figures 16a and b**). These new supplemental data exclude the possibility of AGEs to be involved in cell death and cell proliferation, which supports our migration hypothesis. These results are included in the Result section of the revised manuscript (**page 9, lines 8-10 and page 9, lines 28-30**).

On the other hand, in the original manuscript, cell viability was determined by measuring ATP contents and expressed as relative values to control. The values reflect the combined effects of cell viability and proliferative activity, which might be the reason why some values exceeded 100%. Therefore, in addition to the data on cell viability (Annexin V assay) and cell proliferation (BrdU incorporation assay), this data has been revised to be displayed as "ATP contents" (**Supplementary Figure 16c**).

2. There is no evidence provided on where the inhibition of AGE on macrophage/monocytes might play a role in a disease or more complex in vivo setting. As previously suggested a sepsis model would be a useful model here to evaluate the relevance of the findings of the study. Indeed, assessing the functional impact of AGE binding to histones would greatly increase the value of the study.

Response: Based on the reviewer's suggestion, new data have been added for *in vivo* experiment using cecal ligation and puncture (CLP)-induced sepsis model to clarify the role of AGEs in more pathophysiological settings (**new Figure 5g**). The data showed that mice treated with AGEs rather than unmodified BSA displayed a significantly prolonged survival time, providing evidence that AGEs might play a role in the context of disease. This result is included in the Result section (**page 10, lines 13-16**) and the Discussion section (**page 13, lines 25-27**) of the revised manuscript.

Rather than the superficial analysis of the macrophage phenotype currently provided, doing RNAseq of sorted peritoneal macrophages with and without AGE treatment would be very informative and help better understand the biological impact of AGE binding.

Response: Following the reviewer's suggestion, RNA-Seq analysis of isolated macrophages from the BSA- or AGEs-treated mice was performed for more comprehensive information. Hierarchical clustering analysis revealed that peritoneal macrophages derived from BSA- and AGEs-treated mice displayed different gene expression profiles (**New Supplementary Figure 18a**). By comparing our data with previously published data on gene signatures in M1 and M2 macrophages (Orecchioni *et al.*, *Front. Immunol.*, 10, 1084 (2019)), we found that several M1-associated genes, such as IL-1b, Ccr7, Igtp, and Sell, were significantly downregulated in macrophages from the mice treated with AGEs compared to those treated with BSA (**New Supplementary Figure 18b**). Conversely, AGEs-treatment increased the expressions of M2-associated genes, including GATA2, Clec10a, Snn, Pcdh7, Stxbp6, and Gar1 (**New Supplementary Figure 18b**). These results further strengthened our conclusion that AGEs could suppress the infiltration of pro-inflammatory monocytes/macrophages. Quantitative PCR (qPCR) analysis revealed that treatment with AGEs affected the gene expression of pro-inflammatory M1 macrophage markers (IL-1 β and IL-6) and anti-inflammatory M2 macrophage markers (Arg-1, IL-10, and CD206) in infiltrated macrophages (**Fig. 5f**), whereas no significant differences in their expression except for IL-1 β between treatments were found in the RNA-Seq analysis. This discrepancy between qPCR and RNA-Seq analysis may be due to differences in methodologies between the two technologies in quantifying transcript abundances. Given that qPCR generally has higher sensitivity than RNA-Seq (Everaert *et al. Sci. Rep.*, 7, 1559 (2017)), the relatively small differences (~0.6-fold decrease and ~1.5-fold increase) in gene expression analysis by qPCR shown in **Fig. 5f** might not be detectable by RNA-Seq analysis. The results of these findings are described in detail in the Result section (**page 10, lines 8-11**). The RNA-Seq data have been deposited into the Gene Expression Omnibus (accession code: GSE195558, secure token: upexsueupjylhwf).

Minor concerns:

- In figure 3a, the dot plots for BSA vs AGE condition should be shown, not just the histograms

Response: Following the reviewer's comment, we have added the dot plots of Figure 3a to the **new Supplementary Figure 7**.

- In Figure 3b histograms for each cell type including gating strategy should be shown, not

just an MFI quantification

Response: Following the reviewer's comment, we have added the gating strategy and histograms of Figure 3b to the **new Supplementary Figure 8**.

- In figure 3h dot plots should be shown in addition

Response: We have added contour plots of **Figure 3h** instead of dot plots to the **new Supplementary Figure 9**. This is because the numbers of cells in GFP(-) and GFP(+) were quite different and we need to display these data as the relative frequency of the populations, regardless of the number of events collected. We are very grateful to the reviewer for pointing this out.

Response to Reviewer #1:

The authors have responded robustly, in the opinion of this reviewer, to the many suggestions and criticisms raised in the prior reviews. The new results here, of which there are many, further support the discovery and interpretation of results that the authors report here a novel discovery that histones, and especially cell surface histones, play a role as novel receptors for AGE. These interactions may be particularly important to the recruitment of monocyte/macrophage recruitment. This reviewer has no more specific suggestions for improvement beyond those already mentioned. A key limitation is the physiological significance of this discovery, specifically in regard to cell surface histone involvement in binding AGE. The data largely relies on in vitro models, but overall the results are well presented and the conclusions are strongly supported by these data. Thus, while the concept presented in this manuscript is provocative, it is very interesting and could spur the field to consider the relationship of histones as AGE receptors in comparison to the many other AGE receptors known.

Response: We are very pleased to receive favorable reviews from the reviewer. We agree with the reviewer's comment on the limitation of this discovery, specifically with respect to the involvement of cell surface histone in the binding AGEs. This is partially due to a lack of methodology for specifically inhibiting the interaction between AGEs and histone as addressed in the last submission of the revised manuscript (**page 13, last paragraph**). With that in mind, further study is under way in our laboratory to elucidate specific regulatory mechanisms of AGEs-histone interactions and to establish their physiological relevance.

But to this reviewer, the big picture of how such histone-specific AGE interactions function in the context of the many other AGE receptors, at least in a physiological setting, is completely lacking.

Response: The reviewer's comment reminded us that we should have addressed how histone-specific AGE interactions function in the context of the many other AGE receptors. In this regard, binding of AGEs to the many other AGE receptors has been shown to trigger downstream signaling pathways and mediate the cellular activation or proliferation leading to inflammation and tissue destruction. On the other hand, the discovery of histone as a cell-surface receptor for AGEs in this study suggests that AGEs may also be involved in the homeostatic response via binding to histone. Therefore, there should be some functional interaction between histone and other AGE receptors. To answer the reviewer's concern, the following statement has been added to the Discussion section (**page 14, lines 6-18**).

“Another concern is the relationship between cell-surface histones and other AGE receptors. In this regard, due to the high affinity between AGEs and histones, AGEs are expected to bind immediately to histones on the cell surface in addition to other AGE receptors. Therefore, it can be expected that cell-surface histones may compete with other AGE receptors for binding to AGEs, resulting in the downregulation of AGEs-induced inflammatory responses via other AGE receptors. On the other hand, there may be functional interactions between histones and AGE receptors. An attractive hypothesis is that there may be an intracellular crosstalk of downstream signaling mediated by histone and other AGE receptors. However, the presence of such intracellular signaling mechanisms induced by AGEs remains unexplored. Further study is under way in our laboratory to elucidate specific regulatory mechanisms of AGEs-histone interactions and to investigate functional interactions between histone and other AGE receptors.”

So overall, one has to weigh the content of the paper in terms of its discovery and its potential implications, and on this score the paper is sound and credible.

Response: We sincerely thank the reviewer for a positive evaluation.

Response to Reviewer #6:

The authors have thoroughly addressed all concerns in their revised manuscript, providing both new evidence strengthening their original conclusions as well as improving several minor aspects of the article. The article is in my view fit for publication in its current form.

Response: We sincerely thank the reviewer for a positive evaluation.